# Achieve Performatively Optimal Policy for Performative Reinforcement Learning

## Abstract

Performative reinforcement learning is an emerging dynamical decision making framework, which extends reinforcement learning to the common applications where the agent's policy can change the environmental dynamics. Existing works on performative reinforcement learning only aim at a performatively stable (PS) policy that maximizes an approximate value function. However, there is a provably positive constant gap between the PS policy and the desired performatively optimal (PO) policy that maximizes the original value function. In contrast, this work proposes a zeroth-order performative policy gradient (0-PPG) algorithm that **for the first time converges to the desired PO policy with polynomial computation complexity under mild conditions**. For the convergence analysis, we prove two important properties of the nonconvex value function. First, when the policy regularizer dominates the environmental shift, the value function satisfies a certain gradient dominance property, so that any stationary point of the value function is a desired PO. Second, though the value function has unbounded gradient, we prove that all the sufficiently stationary points lie in a convex and compact policy subspace $\Pi_\Delta$, where the policy value has a constant lower bound $\Delta > 0$ and thus the gradient becomes bounded and Lipschitz continuous.

## 1. Introduction

Reinforcement learning is a powerful dynamic decision making framework with many successes in AI, such as AlphaGo (Silver et al., 2017), AlphaStar (Vinyals et al., 2019), Pluribus (Brown and Sandholm, 2019), large language model alignment (Bai et al., 2022) and reasoning (Havrilla et al., 2024). However, most reinforcement learning works ignore the effect of the deployed policy on the environmental dynamics, including transition kernel and reward function. This effect is significant in some applications. For example, the behavior of the autonomous vehicles can affect the behavior of the pedestrians and the other vehicles, so the environment may become very different from the designers' imagination (Nikolaidis et al., 2017). Also, a recommender system formulated as a contextual Markov decision process not only affects the user demographics (context distribution) but also how users interact with the platforms (Chaney et al., 2018; Mansoury et al., 2020).

To account for such effect of deployed policy on environmental dynamics, performative reinforcement learning has been proposed by (Mandal et al., 2023) where the transition kernel $p_\pi$ and reward function $r_\pi$ are modeled as functions of the deployed policy $\pi$. Similar to conventional reinforcement learning, the ultimate goal is to find the *performatively optimal (PO)* policy that maximizes the *performative value function*, defined as the accumulated discounted reward when deploying a policy $\pi$ to its corresponding environment $(p_\pi, r_\pi)$. However, the policy-dependent environmental dynamics pose significant challenge to achieve PO. Hence, (Mandal et al., 2023) pursues a suboptimal *performatively stable (PS)* policy using repeated retraining method with environmental dynamics fixed for the current policy at each policy optimization step. However, (Mandal et al., 2023) shows that PS can have a positive constant distance to PO.

Two extensions of the basic performative reinforcement learning problem (Mandal et al., 2023) have been proposed and studied. (Rank et al., 2024) extends to the setting where the environmental dynamics gradually adjust to the currently deployed policy, and proposes a mixed delayed repeated retraining algorithm with accelerated convergence to a PS policy. (Mandal and Radanovic, 2024) extends (Mandal et al., 2023) from tabular setting to linear Markov decision processes with large number of states, and also obtains the convergence rate of the repeated retraining algorithm to a PS policy.

In sum, all these existing performative reinforcement learning works pursue a suboptimal PS policy. Therefore, we want to ask the following fundamental research question.

[1]Anonymous Institution, Anonymous City, Anonymous Region, Anonymous Country. Correspondence to: Anonymous Author <anon.email@domain.com>.

Preliminary work. Under review by the International Conference on Machine Learning (ICML). Do not distribute.

> **Q:** *Can we design an algorithm that converges to the desired performatively optimal (PO) policy?*

### 1.1. Our Contributions

We will answer affirmatively to the research question above in the following steps. Each step yields a novel contribution.

• We study an entropy regularized performative reinforcement learning problem, compatible with the basic performative reinforcement learning problem in (Mandal et al., 2023). We prove that the objective function satisfies a certain gradient dominance condition, which implies that an approximate stationary point (not the suboptimal PS) is the desired approximate PO policy, under a mild regularizer dominance condition similar to that used by (Mandal et al., 2023; Rank et al., 2024; Mandal and Radanovic, 2024) to ensure convergence to a suboptimal PS policy. The proof adopts novel techniques such as recursion for $p_\pi$-related error term and frequent switch among various necessary and sufficient conditions of smoothness and strong concavity like properties for various variables (see Section 3.2).

• We obtain a policy lower bound as a decreasing function of a stationary measure. This bound not only implies the unbounded *performative policy gradient* (a challenge to obtain a stationary policy and thus PO), but also inspires us to find a stationary policy in the policy subspace $\Pi_\Delta$ with a constant policy lower bound $\Delta > 0$ where we prove the objective function to be Lipschitz continuous and Lipschitz smooth (a solution to this challenge). The policy lower bound is obtained using a novel technique which simplifies a complicated inequality of the minimum policy value $\pi[a_{\min}(s)|s]$ in two cases (see Section 3.3).

• We construct a zeroth-order estimation of the *performative policy gradient* and obtains its estimation error. This is more challenging than the existing zero-th order estimation methods since our objective function is only well-defined on the policy space, a compact subset of a linear subspace of the Euclidean space $\mathbb{R}^{|\mathcal{S}||\mathcal{A}|}$. To solve this puzzle, we adjust a two-point estimation to the linear subspace $\mathcal{L}_0$ of policy difference, and simplify the estimation error analysis by mapping policies onto the Euclidean space $\mathbb{R}^{|\mathcal{S}|(|\mathcal{A}|-1)}$ via orthogonal transformation (see Section 4.1).

• We propose a zeroth-order performative policy gradient (0-PPG) algorithm (see Algorithm 1) by combining the *performative policy gradient* estimation above with the Frank-Wolfe algorithm. Then we obtain a polynomial computation complexity of our 0-PPG algorithm to converge to a stationary policy, which is also the desired PO policy under the regularizer dominance condition above. The convergence analysis uses a policy averaging technique to show that an approximate stationary policy on $\Pi_\Delta$ is also approximately stationary on the whole policy space $\Pi$ (see Section 4.2).

Finally, we briefly show that the results above, including gradient dominance, Lipschitz properties and the finite-time convergence of 0-PPG algorithm to the desired PO, can be adjusted to the performative reinforcement learning problem with the quadratic regularizer used by (Mandal et al., 2023; Rank et al., 2024) (see Appendix K).

## 2. Preliminary: Performative Reinforcement Learning

### 2.1. Problem Formulation

Performative reinforcement learning is characterized by a Markov decision process (MDP) $\mathcal{M}_\pi = (\mathcal{S}, \mathcal{A}, p_\pi, r_\pi, \rho)$ that depends on a certain policy $\pi$. Here, $\mathcal{S}$ and $\mathcal{A}$ denote the finite state space with cardinality $|\mathcal{S}|$ and finite action space with cardinality $|\mathcal{A}|$ respectively. The policy $\pi \in [0,1]^{|\mathcal{S}||\mathcal{A}|}$, with entries $\pi(a|s)$ for any state $s \in \mathcal{S}$ and action $a \in \mathcal{A}$, lies in the following policy space, such that $\pi(\cdot|s)$ for any state $s$ can be seen as a distribution over $\mathcal{A}$.

$$\Pi \overset{\text{def}}{=} \Big\{ \pi \in [0,1]^{|\mathcal{S}||\mathcal{A}|} : \sum_{a \in \mathcal{A}} \pi(a|s) = 1, \forall s \in \mathcal{S} \Big\}.$$

The transition kernel $p_\pi \in [0,1]^{|\mathcal{S}|^2|\mathcal{A}|}$ dependent on policy $\pi \in \Pi$, with entries $p_\pi(s'|s,a)$ for any $s, s' \in \mathcal{S}$ and $a \in \mathcal{A}$, lies in the following transition kernel space such that $p_\pi(\cdot|s,a)$ can be seen as a state distribution on $\mathcal{S}$.

$$\mathcal{P} \overset{\text{def}}{=} \Big\{ p \in [0,1]^{|\mathcal{S}|^2|\mathcal{A}|} : \sum_{s \in \mathcal{S}} p(s'|s,a) = 1, \forall s \in \mathcal{S}, a \in \mathcal{A} \Big\}.$$

$r_\pi \in \mathcal{R} \overset{\text{def}}{=} [0,1]^{|\mathcal{S}||\mathcal{A}|}$ is the reward function with entries $r_\pi(s,a) \in [0,1]$ for any $s \in \mathcal{S}$ and $a \in \mathcal{A}$. $\rho \in [0,1]^{|\mathcal{S}|}$ is the initial state distribution such that $\sum_{s \in \mathcal{S}} \rho(s) = 1$. Note that we consider $p_\pi, r_\pi, \rho, \pi$ as Euclidean vectors, so that we can conveniently define their Euclidean norm. For example, we define $\|p_\pi\|_q = \big[ \sum_{s,a,s'} |p_\pi(s'|s,a)|^q \big]^{1/q}$ for any $q > 1$ and $\|p_\pi\|_\infty = \max_{s,a,s'} |p_\pi(s'|s,a)|$. Such norms can be similarly defined over $r_\pi, \rho, \pi$ by summing or maximizing over all the entries. Specifically, denote $\|\cdot\| = \|\cdot\|_2$ by convention.

When an agent applies its policy $\pi \in \Pi$ to MDP $\mathcal{M}_{\pi'} = (\mathcal{S}, \mathcal{A}, p_{\pi'}, r_{\pi'}, \rho)$, the initial environmental state $s_0 \in \mathcal{S}$ is generated from the distribution $\rho$. Then at each time $t = 0, 1, 2, \ldots$, the agent takes a random action $a_t \sim \pi(\cdot|s_t)$ based on the current state $s_t \in \mathcal{S}$, the environment transitions to the next state $s_{t+1} \sim p_{\pi'}(\cdot|s_t, a_t)$ and provides reward $r_t = r_{\pi'}(s_t, a_t) \in [0,1]$ to the agent. The value of applying policy $\pi$ to $\mathcal{M}_{\pi'}$ can be characterized by the following *value function*.

$$V_{\lambda, \pi'}^\pi \overset{\text{def}}{=} \mathbb{E}_{\pi, p_{\pi'}, \rho} \Big[ \sum_{t=0}^\infty \gamma^t r_{\pi'}(s_t, a_t) \Big] - \lambda \mathcal{H}_{\pi'}(\pi). \quad (1)$$

Here, $\mathbb{E}_{\pi, p_{\pi'}, \rho}$ is the expectation under policy $\pi$, transition kernel $p_{\pi'}$ and initial state distribution $\rho$. $\gamma \in (0, 1)$ is the discount factor. $\mathcal{H}_{\pi'}(\pi)$ is a regularizer with coefficient $\lambda \geq 0$ to ensure or accelerate algorithm convergence. Existing works use the quadratic regularizers such as $\mathcal{H}_{\pi'}(\pi) = \frac{1}{2}\|d_{\pi, p_{\pi'}}\|^2$ (Mandal et al., 2023; Rank et al., 2024) and $\mathcal{H}_{\pi'}(\pi) = \frac{1}{2}\|\Phi^\top d_{\pi, p_{\pi'}}\|^2$ (Mandal and Radanovic, 2024) with a feature matrix $\Phi$, where the occupancy measure $d_{\pi, p} \in [0, 1]^{|\mathcal{S}||\mathcal{A}|}$ for any policy $\pi$ and transition kernel $p$ is defined as the following distribution on $\mathcal{S} \times \mathcal{A}$.

$$d_{\pi, p}(s, a) \overset{\text{def}}{=} (1 - \gamma) \sum_{t=0}^{\infty} \gamma^t \mathbb{P}_{\pi, p, \rho}\{s_t = s, a_t = a\}. \quad (2)$$

Then the state occupancy measure defined as $d_{\pi, p}(s) \overset{\text{def}}{=} \sum_a d_{\pi, p}(s, a)$ satisfies the following well-known Bellman equation for any state $s' \in \mathcal{S}$.

$$d_{\pi, p}(s') = (1 - \gamma)\rho(s') + \gamma \sum_{s, a} d_{\pi, p}(s)\pi(a|s)p(s'|s, a). \quad (3)$$

The ultimate goal of performative reinforcement learning is to find the *performatively optimal (PO)* policy $\pi$ that maximizes the *performative value function* $V_{\lambda, \pi}^\pi$ (with $\pi' = \pi$ in Eq. (1)), as formally defined below.

**Definition 1** (Ultimate Goal: PO). *For any $\epsilon \geq 0$, a policy $\pi \in \Pi$ is defined as $\epsilon$-performatively optimal ($\epsilon$-PO) if $\max_{\pi' \in \Pi} V_{\lambda, \pi'}^{\pi'} - V_{\lambda, \pi}^\pi \leq \epsilon$. Specifically, we call a 0-PO policy as a PO policy.*

Conventional reinforcement learning can be seen as a special case of performative reinforcement learning with fixed environmental dynamics, namely, constant transition kernel $p_\pi \equiv p$ and constant reward function $r_\pi \equiv r$. However, this may fail on applications with policy-dependent environmental dynamics, such as recommender system and autonomous driving (Mandal et al., 2023) as explained in Section 1.

## 2.2. Performatively Stable (PS) Policy in Existing Works

Achieving an $\epsilon$-PO policy (defined by Definition 1) is challenging, due to the policy-dependent environmental dynamics $p_\pi$ and $r_\pi$. To alleviate the challenge, all the existing works (Mandal et al., 2023; Rank et al., 2024; Mandal and Radanovic, 2024) aim at a *performatively stable (PS)* policy $\pi_{\text{PS}}$ defined as follows, as an approximation of a *PO policy*.

$$\pi_{\text{PS}} \in \arg\max_{\pi \in \Pi} V_{\lambda, \pi_{\text{PS}}}^\pi. \quad (4)$$

In other words, a PS policy $\pi_{\text{PS}}$ has the optimal value on the fixed environment $\mathcal{M}_{\pi_{\text{PS}}}$. However, (Mandal et al., 2023) shows that a PS policy can be suboptimal, so these existing algorithms cannot converge to a PO policy. Nevertheless, we will briefly introduce these algorithms, to later

partially inspire and compare with our method for achieving a PO policy. Note that an occupancy measure $d$ (a distribution on $\mathcal{S} \times \mathcal{A}$) corresponds to the policy $\pi^d$ defined as $\pi^d(a|s) = \frac{d(s, a)}{d(s)}$ ($\pi^d(a|s) = 1/|\mathcal{A}|$ if $d(s) = 0$), where $d(s) = \sum_{a' \in \mathcal{A}} d(s, a')$. Hence, (Mandal et al., 2023; Rank et al., 2024; Mandal and Radanovic, 2024) transform the policy optimization problem (4) into a problem of solving $d$. The basic performative reinforcement learning (Mandal et al., 2023) considers the following dual optimization problem of $d$ in the environment $p_{d'} = p_{\pi^{d'}}$, $r_{d'} = r_{\pi^{d'}}$ corresponding to another occupancy measure $d'$.

$$
\begin{cases}
\displaystyle \max_{d:\text{distribution on } \mathcal{S} \times \mathcal{A}} \sum_{s, a} d(s, a)r_{d'}(s, a) - \frac{\lambda}{2}\|d\|^2 \\
\text{s.t.} \sum_a d(s, a) = \rho(s) + \gamma \sum_{s', a} d(s', a)p_{d'}(s|s', a)
\end{cases} \quad (5)
$$

The objective function above corresponds to the value function $V_{\lambda, \pi'}^\pi$ defined in Eq. (1) with quadratic regularizer $\mathcal{H}_{\pi'}(\pi) = \frac{1}{2}\|d_{\pi, p_{\pi'}}\|^2$. The equality constraint above comes from the Bellman equation (3). Denote $\phi(d')$ as the optimal solution to the problem (5) above. Then the target becomes a performatively stable occupancy measure $d_{\text{PS}}$ defined as a fixed point $d_{\text{PS}} = \phi(d_{\text{PS}})$, which corresponds to a PS policy $\pi_{\text{PS}} = \pi^{d_{\text{PS}}}$. Suppose the transition kernel and reward function are sensitive with parameters $\epsilon_p', \epsilon_r' > 0$ respectively, that is, for any occupancy measures $d, d'$.

$$\|p_{d'} - p_d\| \leq \epsilon_p'\|d' - d\|, \quad \|r_{d'} - r_d\| \leq \epsilon_r'\|d' - d\|. \quad (6)$$

It has been proved by (Mandal et al., 2023) that $\phi$ is a contraction mapping under a regularizer dominance condition that $\lambda > \mathcal{O}(\epsilon_p' + \epsilon_r')$. In this case, any repeated retraining method characterized by $d_{t+1} \approx \phi(d_t)$ with sufficient precision can converge to the PS policy.

Similarly, (Rank et al., 2024; Mandal and Radanovic, 2024) also apply repeated retraining to optimization problems of occupancy measure, which converges to a PS policy for extensions of the basic performative reinforcement learning (Mandal et al., 2023). Next, we will propose our significantly different strategies to achieve the desired PO policy.

# 3. Entropy Regularized Performative Reinforcement Learning

In this section, we obtain critical properties of an entropy regularized performative reinforcement learning problem for achieving the desired PO policy.

## 3.1. Negative Entropy Regularizer

To achieve the PO policy, one might attempt to solve the problem $(\mathcal{P}_d)$, adjusted from the dual problem (5) above with fixed $d'$ replaced by the decision variable $d$. The solution $d_{\text{PO}}$ will yield the PO policy $\pi^{d_{\text{PO}}}$. However, such

replacement will make the convex quadratic optimization problem (5) much more complicated, due to the unknown and possibly complicated functions $p_d$ and $r_d$. Therefore, we will instead focus on the primal problem $\max_\pi V_{\lambda,\pi}^\pi$.

We consider the following negative entropy regularizer of the policy $\pi$, which is widely used in reinforcement learning to encourage environment exploration and accelerate convergence (Mnih et al., 2016; Mankowitz et al., 2019; Cen et al., 2022; Chen and Huang, 2024).

$$\mathcal{H}_{\pi'}(\pi) = \mathbb{E}_{\pi, p_{\pi'}, \rho}\Big[ \sum_{t=0}^{\infty} \gamma^t \log \pi(a_t|s_t) \Big]. \quad (7)$$

In addition, this negative entropy regularizer can be seen as a strongly convex function of the occupancy measure $d_{\pi, p_{\pi'}}$ (proved in Appendix B), which is critical to develop algorithms convergent to a PO (see Theorem 1 later) or PS policy (Mandal et al., 2023). For optimization problem on a probability simplex variable (policy $\pi$ or occupancy measure $d$), negative entropy regularizer is more natural and yields faster theoretical convergence than the quadratic regularizers used in the existing performative reinforcement learning works (Mandal et al., 2023; Rank et al., 2024) (see pages 43-45 of (Chen, 2020) for explanation).

Therefore, we will mainly focus on the following entropy-regularized value function, which is obtained by substituting the negative entropy regularizer (7) into the general value function (1).

$$V_{\lambda,\pi'}^\pi \stackrel{\text{def}}{=} \mathbb{E}_{\pi, p_{\pi'}, \rho}\Big[\sum_{t=0}^{\infty} \gamma^t [r_{\pi'}(s_t, a_t) - \lambda \log \pi(a_t|s_t)]\Big]. \quad (8)$$

Specifically, we will study the critical properties of the entropy-regularized value function (8) (Section 4) to develop algorithm that converges to PO (Sections 4.1-4.2). Then we will briefly discuss about how to adjust these results to the existing quadratic regularizers (Appendix K).

We make the following standard assumptions to study the properties of the entropy-regularized value function (8).

**Assumption 1** (Sensitivity). *There exist constants $\epsilon_p, \epsilon_r > 0$ such that for any $\pi, \pi' \in \Pi$,*

$$\|p_{\pi'} - p_\pi\| \le \epsilon_p \|\pi' - \pi\|, \quad \|r_{\pi'} - r_\pi\| \le \epsilon_r \|\pi' - \pi\| \quad (9)$$

**Assumption 2** (Smoothness). *$p_\pi$ and $r_\pi$ are Lipschitz smooth with modulus $S_p, S_r > 0$ respectively, that is, for any $\pi \in \Pi$, $s, s' \in \mathcal{S}$, $a \in \mathcal{A}$, we have*

$$\|\nabla_\pi p_{\pi'}(s'|s, a) - \nabla_\pi p_\pi(s'|s, a)\| \le S_p \|\pi' - \pi\|, \quad (10)$$

$$\|\nabla_\pi r_{\pi'}(s, a) - \nabla_\pi r_\pi(s, a)\| \le S_r \|\pi' - \pi\|. \quad (11)$$

**Assumption 3.** *There exists a constant $D > 0$ such that $\inf_{\pi \in \Pi, p \in \mathcal{P}, s \in \mathcal{S}} d_{\pi, p}(s) \ge D$.*

Assumptions 1-2 ensure that the environmental dynamics $p_\pi$ and $r_\pi$ adjust continuously and smoothly to policy $\pi$, and thus the *performative value function* $V_{\lambda,\pi}^\pi$ is differentiable with *performative policy gradient* $\nabla_\pi V_{\lambda,\pi}^\pi$. Similar versions of Assumption 1 on environmental sensitivity have been used in the performative reinforcement learning literature (e.g. Eq. (6) in (Mandal et al., 2023)). Assumption 3 has been used (Zhang et al., 2021) or implied by stronger assumptions (Wei et al., 2021; Chen et al., 2022; Agarwal et al., 2021; Leonardos et al., 2022; Wang et al., 2023; Chen and Huang, 2024; Bhandari and Russo, 2024) in conventional reinforcement learning (see Appendix C for the proof), which guarantees that each state is visited sufficiently often.

### 3.2. Gradient Dominance

For the nonconvex policy optimization problem $\max_{\pi \in \Pi} V_{\lambda,\pi}^\pi$ with the entropy regularized value function (8) on the convex policy space $\Pi$, it is natural to consider its approximate stationary solution as defined below.

**Definition 2** (Stationary Policy). *For any $\epsilon \ge 0$, a policy $\pi \in \Pi$ is $\epsilon$-stationary if $\max_{\tilde{\pi} \in \Pi} \langle \nabla_\pi V_{\lambda,\pi}^\pi, \tilde{\pi} - \pi \rangle \le \epsilon$. We call a 0-stationary policy as a stationary policy.*

Note that for a policy to be the desired PO, it is necessary to be stationary, while the PS policy targeted by existing works is neither necessary nor sufficient. Furthermore, we will show that stationary policy can also be a sufficient condition of the desired PO under mild conditions. As a preliminary step, we show the important gradient dominance property of the objective function as follows.

**Theorem 1** (Gradient Dominance). *Under Assumptions 1-3, the entropy regularized value function (8) satisfies the following gradient dominance property for any $\pi_0, \pi_1 \in \Pi$.*

$$V_{\lambda,\pi_1}^{\pi_1} \le V_{\lambda,\pi_0}^{\pi_0} + D^{-1} \max_{\pi \in \Pi} \langle \nabla_{\pi_0} V_{\lambda,\pi_0}^{\pi_0}, \pi - \pi_0 \rangle - \frac{\mu}{2} \|\pi_1 - \pi_0\|^2, \quad (12)$$

*where the constant $\mu \in \mathbb{R}$ is defined as follows.*

$$\mu = \frac{D\lambda}{1-\gamma} - \frac{6\gamma|\mathcal{S}|(1 + \lambda \log |\mathcal{A}|)}{D(1-\gamma)^3}$$
$$\big[\epsilon_p\big(\sqrt{|\mathcal{A}|} + \gamma \epsilon_p \sqrt{|\mathcal{S}|}\big) + S_p(1-\gamma)\big]$$
$$- \frac{S_r(1-\gamma) + 4\epsilon_r(\sqrt{|\mathcal{A}|} + \gamma \epsilon_p \sqrt{|\mathcal{S}|})}{D^2(1-\gamma)^2}. \quad (13)$$

**Remark:** With sufficiently large regularizer strength $\lambda$ and small environmental shift strength $\epsilon_p, \epsilon_r, S_p, S_r$ (i.e., when the regularizer dominates the environmental shift), we have $\mu \ge 0$, which implies the gradient dominance form (Eq. (12) with $\mu = 0$) that holds for conventional unregularized reinforcement learning (see Lemma 4 of (Agarwal

et al., 2021)). In this case, stationary policy becomes a sufficient condition of the desired PO, as shown in the following Corollary 1. Note that the existing performative reinforcement learning works (Mandal et al., 2023; Rank et al., 2024; Perdomo et al., 2020) also require a regularizer dominance condition similar to our $\mu \geq 0$ (e.g. $\lambda > \mathcal{O}(\epsilon'_p + \epsilon'_r)$ in (Mandal et al., 2023)) to ensure convergence to a PS policy.

**Corollary 1.** *Under Assumptions 1-3, if $\mu \geq 0$ for $\mu$ defined in Eq. (13), then any $D\epsilon$-stationary policy is also the desired $\epsilon$-PO policy. Furthermore, if $\mu > 0$, the PO policy is unique.*

**Intuition and Novelty for Proving Theorem 1:** Define the following more refined value function

$$J_\lambda(\pi, \pi', p, r)$$
$$\overset{\text{def}}{=} \mathbb{E}_{\pi,p}\Big[\sum_{t=0}^{\infty}\gamma^t[r(s_t,a_t) - \lambda\log\pi'(a_t|s_t)]\Big|s_0 \sim \rho\Big]. \quad (14)$$

To get the intuition, we consider the following three cases from the simplest conventional reinforcement learning to the hardest performative reinforcement learning.

(Case I): For conventional reinforcement learning with fixed dynamics $p_\pi \equiv p$ and $r_\pi \equiv r$, denote $d_\alpha = \alpha d_{\pi_1,p} + (1-\alpha)d_{\pi_0,p}$ ($\alpha \in [0,1]$). Based on the Bellman equation (3), $d_\alpha = d_{\pi_\alpha,p}$ is the occupancy measure of the policy $\pi_\alpha(a|s) = \frac{d_\alpha(s,a)}{d_\alpha(s)}$. Therefore, $V_{\lambda,\pi_\alpha}^{\pi_\alpha}$ can be rewritten as $J_\lambda(\pi_\alpha, \pi_\alpha, p, r) = \sum_{s,a} d_\alpha(s,a)[r(s,a) - \lambda\log\pi_\alpha(a|s)]$, which has the following strong concavity like property by Pinsker's inequality (see Eq. (91) for detail).

$$J_\lambda(\pi_\alpha,\pi_\alpha,p,r) - \alpha J_\lambda(\pi_1,\pi_1,p,r)$$
$$- (1-\alpha)J_\lambda(\pi_0,\pi_0,p,r)$$
$$= \frac{1}{1-\gamma}\sum_s\Big[\alpha d_1(s)\text{KL}[\pi_1(\cdot|s)\|\pi_\alpha(a|s)]$$
$$+ (1-\alpha)d_0(s)\text{KL}[\pi_0(\cdot|s)\|\pi_\alpha(a|s)]\Big]$$
$$\geq \frac{D\lambda\alpha(1-\alpha)}{2(1-\gamma)}\|\pi_1 - \pi_0\|^2. \quad (15)$$

(Case II): Consider a harder case with varying $p_\pi$ and constant reward $r_\pi \equiv r$. Similarly, we denote $d_\alpha = \alpha d_{\pi_1,p_{\pi_1}} + (1-\alpha)d_{\pi_0,p_{\pi_0}}$ and $\pi_\alpha(a|s) = \frac{d_\alpha(s,a)}{d_\alpha(s)}$. The varying $p_\pi$ brings a major challenge that $d_\alpha = d_{\pi_\alpha,p_{\pi_\alpha}}$ required by Case I no longer holds. To solve this challenge, we prove that the error term $e_\alpha(s) = d_{\pi_\alpha,p_\alpha}(s) - d_\alpha(s)$ of interest satisfies the following novel recursion (see Eq. (89) for the derivation based on the Bellman equation (3)).

$$e_\alpha(s') = \gamma\sum_{s,a}\Big[e_\alpha(s)\pi_\alpha(a|s)p_{\pi_\alpha}(s'|s,a) + h_\alpha(s,a,s')\Big],$$

where $h_\alpha(s,a,s') = d_\alpha(s,a)p_{\pi_\alpha}(s'|s,a) - \alpha d_1(s,a)p_{\pi_1}(s'|s,a) - (1-\alpha)d_0(s,a)p_{\pi_0}(s'|s,a)$. Since

$d_\alpha(s,a)p_{\pi_\alpha}(s'|s,a)$ is a Lipschitz smooth function of $\alpha$ with Lipschitz constant $\ell_{dp}(s,a)$ defined by Eq. (87), we have $|h_\alpha(s,a,s')| \leq \frac{\alpha(1-\alpha)}{2}\ell_{dp}(s,a)$, which can be substituted into the recursion above and yields the following novel error bound (see Eq. (90) for detail).

$$\sum_s |e_\alpha(s)| \leq \alpha(1-\alpha)\mathcal{O}(\epsilon_p + S_p)\|\pi_1 - \pi_0\|^2,$$

which implies the desired strong concavity like property as follows.

$$J_\lambda(\pi_\alpha,\pi_\alpha,p_\alpha,r) - \alpha J_\lambda(\pi_1,\pi_1,p_1,r)$$
$$- (1-\alpha)J_\lambda(\pi_0,\pi_0,p_0,r)$$
$$\geq \text{Eq. (15)} - \alpha(1-\alpha)(1+\lambda)\mathcal{O}(\epsilon_p + S_p)\|\pi_1 - \pi_0\|^2$$
$$\geq \frac{\alpha(1-\alpha)\mu_1}{2}\|\pi_1 - \pi_0\|^2 \quad (16)$$

where $\mu_1 = \frac{D\lambda}{2(1-\gamma)} - (1+\lambda)\mathcal{O}(\epsilon_p + S_p)$ defined by Eq. (92) equals $\mu$ defined by Eq. (13) when $\epsilon_r = S_r = 0$.

(Case III): Now we consider performative reinforcement learning with varying $p_\pi$ and $r_\pi$. The policy $\pi_\alpha$ and its occupancy measure $d_\alpha$ are the same as in Case II above. Then the function $w(\alpha) = \alpha J_\lambda(\pi_1,\pi_1,p_1,r_\alpha) + (1-\alpha)J_\lambda(\pi_0,\pi_0,p_0,r_\alpha)$ can be proved Lipschitz smooth with parameter $\mu_2 = \mathcal{O}(S_r + \epsilon_r)$ defined by Eq. (94), so using $r = r_\alpha$ in Eq. (16) we obtain the following strong concavity like property.

$$J_\lambda(\pi_\alpha,\pi_\alpha,p_\alpha,r_\alpha) - \alpha J_\lambda(\pi_1,\pi_1,p_1,r_1)$$
$$- (1-\alpha)J_\lambda(\pi_0,\pi_0,p_0,r_0)$$
$$\geq \frac{\alpha(1-\alpha)\mu_1}{2}\|\pi_1 - \pi_0\|^2 + w(\alpha) - \alpha w(1) - (1-\alpha)w(0)$$
$$\geq \frac{\alpha(1-\alpha)(\mu_1 - \mu_2)}{2}\|\pi_1 - \pi_0\|^2.$$

Rearranging the inequality above, we obtain the following inequality of $V_{\lambda,\pi_\alpha}^{\pi_\alpha} = J_\lambda(\pi_\alpha,\pi_\alpha,p_\alpha,r_\alpha)$.

$$\frac{V_{\lambda,\pi_\alpha}^{\pi_\alpha} - V_{\lambda,\pi_0}^{\pi_0}}{\alpha} \geq V_{\lambda,\pi_1}^{\pi_1} - V_{\lambda,\pi_0}^{\pi_0} + \frac{\mu(1-\alpha)}{2}\|\pi_1 - \pi_0\|^2,$$

where $\mu = \mu_1 - \mu_2$ is exactly defined by Eq. (13). Letting $\alpha \to +0$ above, we have

$$V_{\lambda,\pi_1}^{\pi_1} \leq V_{\lambda,\pi_0}^{\pi_0} + \Big[\frac{d}{d\alpha}V_{\lambda,\pi_\alpha}^{\pi_\alpha}\Big]\Big|_{\alpha=0} - \frac{\mu}{2}\|\pi_1 - \pi_0\|^2.$$

Using the chain rule, we can find a policy $\pi_0^*$ such that $\big[\frac{d}{d\alpha}V_{\lambda,\pi_\alpha}^{\pi_\alpha}\big]\big|_{\alpha=0} \leq D\langle\nabla_{\pi_0}V_{\lambda,\pi_0}^{\pi_0}, \pi_0^* - \pi_0\rangle$, which along with the bound above proves the gradient dominance property (12).

### 3.3. Policy Lower Bound and Lipschitz Properties

**Policy Lower Bound:** Based on Section 3.2, we can focus on achieving an $\epsilon$-stationary policy. A major challenge is

the unbounded *performative policy gradient* $\nabla_\pi V_{\lambda,\pi}^\pi$ on $\Pi$. Specifically, we will show that as $\pi(a|s) \to 0$ for any state $s$ and action $a$, $\|\nabla_\pi V_{\lambda,\pi}^\pi\| \to +\infty$. To tackle this challenge, we prove the following policy lower bound.

**Theorem 2.** *If Assumptions 1 and 3 hold, and $p_\pi$, $r_\pi$ are differentiable functions of $\pi$, then the following policy lower bound holds for any $\pi \in \Pi$, $s \in \mathcal{S}$, $a \in \mathcal{A}$.*

$$\pi(a|s) \geq \pi_{\min} \exp\left[ -\frac{2|\mathcal{A}|}{\lambda}(1-\gamma)\langle \nabla_\pi V_{\lambda,\pi}^\pi, \pi'-\pi \rangle \right]. \quad (17)$$

*Here, we define the following constant $\pi_{\min}$ and policy $\pi'$.*

$$\pi_{\min} \overset{\text{def}}{=} \frac{1}{2|\mathcal{A}|^{1/(1-\gamma)}} \exp\Bigg\{ -\frac{1}{\lambda(1-\gamma)}$$
$$-\frac{2|\mathcal{A}|\sqrt{2|\mathcal{S}|}}{\lambda}\left[ \frac{\epsilon_p\sqrt{|\mathcal{S}|}(1+\lambda\log|\mathcal{A}|)}{1-\gamma} + \epsilon_r \right] \Bigg\}, \quad (18)$$

$$\pi'(a|s) = \begin{cases} \pi[a_{\min}(s)|s], & a = a_{\max}(s) \\ \pi[a_{\max}(s)|s], & a = a_{\min}(s) , \\ \pi(a|s), & \text{Otherwise} \end{cases} \quad (19)$$

*where $a_{\max}(s) \in \arg\max_a \pi(a|s)$ and $a_{\min}(s) \in \arg\min_a \pi(a|s)$.*

**Implications of Theorem 2:** First, as $\pi(a|s) \to 0$, we have $\langle \nabla_\pi V_{\lambda,\pi}^\pi, \pi'-\pi \rangle \to +\infty$, so $\|\nabla_\pi V_{\lambda,\pi}^\pi\| \to +\infty$ as aforementioned. Second, any stationary policy $\pi$ satisfies $\langle \nabla_\pi V_{\lambda,\pi}^\pi, \pi'-\pi \rangle \leq 0$, so $\pi(a|s) \geq \pi_{\min}$. Therefore, we can search $\epsilon$-stationary policy on the convex and compact policy subspace $\Pi_\Delta \overset{\text{def}}{=} \{\pi \in \Pi : \pi(a|s) \geq \Delta\}$ with lower bound $\Delta \in (0, \pi_{\min}]$.

**Intuition and Novelty for Proving Theorem 2:** As a preliminary step, consider a conventional reinforcement learning problem with fixed environmental dynamics $p_\pi \equiv p$ and $r_\pi \equiv r$. In this case, $\nabla_\pi V_{\lambda,\pi}^\pi$ has analytical form (see Eq. (98)) based on policy gradient theorem, so by direct computation we obtain the following bound (see Eq. (99) for detail)

$$\langle \nabla_\pi V_{\lambda,\pi}^\pi, \pi'-\pi \rangle \geq \frac{1}{1-\gamma} \max_s \Big\{ \big(\pi[a_{\max}(s)|s] - \pi[a_{\min}(s)|s]\big)$$
$$\left[ \lambda \log \frac{\pi[a_{\max}(s)|s]}{\pi[a_{\min}(s)|s]} - 1 - \frac{\gamma(1+\lambda\log|\mathcal{A}|)}{1-\gamma} \right] \Big\}.$$

To directly solve the inequality above of $\pi[a_{\min}(s)|s]$ is not easy. To simplify this inequality, we consider two cases, either $\pi[a_{\min}(s)|s] \geq \frac{1}{2}\pi[a_{\max}(s)|s] \geq \frac{1}{2|\mathcal{A}|}$ or $\pi[a_{\min}(s)|s] < \frac{1}{2}\pi[a_{\max}(s)|s]$. In the second case, we can replace $\pi[a_{\max}(s)|s]$ and $\pi[a_{\max}(s)|s] - \pi[a_{\min}(s)|s]$ above with their lower bounds $\frac{1}{|\mathcal{A}|}$ and $\frac{1}{2|\mathcal{A}|}$ respectively. Then it becomes straightforward to obtain the policy lower bound.

$$\pi[a_{\min}(s)|s] \geq \pi'_{\min}\exp\left[ -\frac{2|\mathcal{A}|}{\lambda}(1-\gamma)\langle \nabla_\pi V_{\lambda,\pi}^\pi, \pi'-\pi \rangle \right],$$

where $\pi'_{\min}$ is defined by Eq. (18) with $\epsilon_p = \epsilon_r = 0$.

Then by extending conventional reinforcement learning to performative reinforcement learning, $\nabla_\pi V_{\lambda,\pi}^\pi$ is perturbed by a magnitude of at most $\frac{\epsilon_p\sqrt{|\mathcal{S}|}(1+\lambda\log|\mathcal{A}|)}{1-\gamma} + \epsilon_r$ (see Eq. (102) for detail) based on the chain rule. This perturbation bound along with $\|\pi'-\pi\| \leq \sqrt{2|\mathcal{S}|}$ yields the second line of Eq. (18) and proves Theorem 2.

**Lipschitz Properties:** Furthermore, in the policy subspace $\Pi_\Delta$, the *performative value function* $V_{\lambda,\pi}^\pi$ is actually Lipschitz continuous and Lipschitz smooth as shown below, which facilitates finding an $\epsilon$-stationary policy in $\Pi_\Delta$.

**Theorem 3.** *Under Assumptions 1-2, $V_{\lambda,\pi}^\pi$ satisfies the following Lipschitz propreties for any $\Delta > 0$ and $\pi, \pi' \in \Pi_\Delta$.*

$$|V_{\lambda,\pi'}^{\pi'} - V_{\lambda,\pi}^\pi| \leq \frac{L_\lambda}{\Delta}\|\pi'-\pi\|, \quad (20)$$

$$\|\nabla_{\pi'} V_{\lambda,\pi'}^{\pi'} - \nabla_\pi V_{\lambda,\pi}^\pi\| \leq \frac{\ell_\lambda}{\Delta}\|\pi'-\pi\|. \quad (21)$$

*where*

$$L_\lambda \overset{\text{def}}{=} \frac{\sqrt{|\mathcal{A}|}(2-\gamma+\gamma\lambda\log|\mathcal{A}|) + \epsilon_p\sqrt{|\mathcal{S}|}(1+\lambda\log|\mathcal{A}|)}{(1-\gamma)^2}$$
$$+ \frac{\epsilon_r}{1-\gamma} \quad (22)$$

$$\ell_\lambda \overset{\text{def}}{=} \frac{3|\mathcal{A}|(1+\lambda\log|\mathcal{A}|)}{(1-\gamma)^2} + \frac{\epsilon_p\sqrt{|\mathcal{S}||\mathcal{A}|}(5+6\lambda\log|\mathcal{A}|)}{(1-\gamma)^3}$$
$$+ \frac{\epsilon_r\left[\sqrt{|\mathcal{A}|}(1-\gamma) + \sqrt{|\mathcal{S}|}(\gamma+2\epsilon_p)\right]}{|\mathcal{A}|(1-\gamma)^2}$$
$$+ \frac{S_p\sqrt{|\mathcal{S}|}(1+\lambda\log|\mathcal{A}|) + S_r(1-\gamma)}{|\mathcal{A}|(1-\gamma)^2}. \quad (23)$$

## 4. Zeroth-Order Performative Policy Gradient (0-PPG) Algorithm

### 4.1. Performative Policy Gradient Estimation

In Section 3, we have obtained important properties of the entropy regularized *performative value function* $V_{\lambda,\pi}^\pi$ (defined by Eq. (8)), which indicates that it suffices to find an $\epsilon$-stationary policy in the subspace $\Pi_\Delta$ for $\Delta \in (0, \pi_{\min}]$. To achieve this goal, an accurate estimation of the *performative policy gradient* $\nabla_\pi V_{\lambda,\pi}^\pi$ is important, which has two challenges. First, unlike conventional reinforcement learning where policy gradient has analytical form, such analytical form does not exist in performative reinforcement learning due to the arbitrary forms of $p_\pi$ and $r_\pi$. Second, in practice, we cannot access the values of $p_\pi(s'|s,a)$ and $r_\pi(s,a)$ but can only obtain stochastic samples from them (Mandal et al., 2023).

Despite these challenges in estimating $\nabla_\pi V_{\lambda,\pi}^\pi$, note that $V_{\lambda,\pi}^\pi$ for any policy $\pi$ can be evaluated, since it is actually

the policy evaluation problem in conventional reinforcement learning under fixed environment $p_\pi$ and $r_\pi$ (for fixed $\pi$). Furthermore, for any $\epsilon_V > 0$ and $\eta \in (0, 1)$, many existing policy evaluation algorithms such as temporal difference (Bhandari et al., 2018; Li et al., 2023; Samsonov et al., 2023), can obtain $\hat{V}^\pi_{\lambda,\pi} \approx V^\pi_{\lambda,\pi}$ with the following $\epsilon_V$ error with probability at least $1 - \eta$.

$$|\hat{V}^\pi_{\lambda,\pi} - V^\pi_{\lambda,\pi}| \le \epsilon_V. \tag{24}$$

As a result, we will consider a zeroth-order estimation of $\nabla_\pi V^\pi_{\lambda,\pi}$ using policy evaluation. However, this has another challenge that $V^\pi_{\lambda,\pi}$ is only well-defined on $\pi \in \Pi$, so we cannot directly apply the existing zeroth-order estimation methods (Agarwal et al., 2010; Shamir, 2017; Malik et al., 2020) which require the objective function to be well-defined on a sphere. Fortunately, for any $\pi, \pi' \in \Pi$, the policy difference $\pi' - \pi$ lies in the following linear subspace of dimensionality $|\mathcal{S}|(|\mathcal{A}| - 1)$.

$$\mathcal{L}_0 \stackrel{\text{def}}{=} \left\{ u \in \mathbb{R}^{|\mathcal{S}||\mathcal{A}|} : \sum_a u(a|s) = 0, \forall s \in \mathcal{S} \right\}. \tag{25}$$

Therefore, inspired by the popular two-point zeroth-order estimations, we obtain the following estimation of $\nabla_\pi V^\pi_{\lambda,\pi}$.

$$\hat{g}_{\lambda,\delta}(\pi) = \frac{|\mathcal{S}|(|\mathcal{A}| - 1)}{2N\delta} \sum_{i=1}^N \left( \hat{V}^{\pi+\delta u_i}_{\lambda,\pi+\delta u_i} - \hat{V}^{\pi-\delta u_i}_{\lambda,\pi-\delta u_i} \right) u_i, \tag{26}$$

where $\{u_i\}_{i=1}^N$ are i.i.d. samples from the uniform distribution on $U_1 \cap \mathcal{L}_0$ with

$$U_1 \stackrel{\text{def}}{=} \{ u \in \mathbb{R}^{|\mathcal{S}||\mathcal{A}|} : \|u\| = 1 \}. \tag{27}$$

Our estimation (26) above is more tricky than the existing two-point zeroth-order estimations (Agarwal et al., 2010; Shamir, 2017; Malik et al., 2020) where $u_i$ is uniformly distributed on $U_1$. To elaborate, we replace their $U_1$ with $U_1 \cap \mathcal{L}_0$, a complete unit sphere on the linear subspace $\mathcal{L}_0$, and further require $\pi \in \Pi_\Delta$ and $\delta < \Delta$, to guarantee that $\pi + \delta u_i, \pi - \delta u_i \in \Pi$ for any $u_i \in U_1 \cap \mathcal{L}_0$ and thus the stochastic gradient estimation (26) is valid (see Appendix H for the proof of validity). Moreover, we use the following three steps to obtain $u_i$ uniformly from $U_1 \cap \mathcal{L}_0$: (1) Obtain $v_i$ from the uniform distribution on $U_1$; (2) Project $v_i$ onto $\mathcal{L}_0$ as Eq. (28) below; (3) Normalize this projection as Eq. (29) below.

$$\text{proj}_{\mathcal{L}_0}(v_i)(a|s) = v_i(a|s) - \frac{1}{|\mathcal{A}|} \sum_{a'} v_i(a'|s), \tag{28}$$

$$u_i = \frac{\text{proj}_{\mathcal{L}_0}(v_i)}{\|\text{proj}_{\mathcal{L}_0}(v_i)\|}. \tag{29}$$

The gradient estimation (26) has the following provable error bound.

**Proposition 1.** *For any $\Delta > \delta > 0$, $\eta \in (0, 1)$ and $\pi \in \Pi_\Delta$, then the stochastic gradient $\hat{g}_{\lambda,\delta}(\pi)$ defined by Eq. (26) is valid and approximates the projected performative policy gradient $\text{proj}_{\mathcal{L}_0}(\nabla_\pi V^\pi_{\lambda,\pi})$ with the following error bound with probability at least $1 - \eta$.*

$$\begin{aligned} &\|\hat{g}_{\lambda,\delta}(\pi) - \text{proj}_{\mathcal{L}_0}(\nabla_\pi V^\pi_{\lambda,\pi})\| \\ \le& \frac{2|\mathcal{S}||\mathcal{A}|\epsilon_V}{\delta} + \frac{4L_\lambda |\mathcal{S}||\mathcal{A}|}{3N(\Delta - \delta)} \log\left( \frac{3N|\mathcal{S}||\mathcal{A}|}{\eta} \right) \\ &+ \frac{L_\lambda |\mathcal{S}||\mathcal{A}|}{\Delta - \delta} \sqrt{\frac{2}{N} \log\left( \frac{3N|\mathcal{S}||\mathcal{A}|}{\eta} \right)} + \frac{\delta \ell_\lambda}{\Delta - \delta}. \end{aligned} \tag{30}$$

**Remark:** Proposition 1 above aims to approximate $\text{proj}_{\mathcal{L}_0}(\nabla_\pi V^\pi_{\lambda,\pi})$ instead of $\nabla_\pi V^\pi_{\lambda,\pi}$. This is sufficient to obtain an $\epsilon$-stationary policy, because for any policies $\pi, \pi'$, the stationarity measure only involves $\langle \nabla_\pi V^\pi_{\lambda,\pi}, \pi' - \pi \rangle$ which equals to $\langle \text{proj}_{\mathcal{L}_0}(\nabla_\pi V^\pi_{\lambda,\pi}), \pi' - \pi \rangle$ as $\pi' - \pi \in \mathcal{L}_0$. Therefore, we only care about $\text{proj}_{\mathcal{L}_0}(\nabla_\pi V^\pi_{\lambda,\pi})$.

The approximation error (30) has the order of $\mathcal{O}\left( \frac{\epsilon_V}{\delta} + \frac{\log(N/\eta)}{\sqrt{N}} + \delta \right)$, which can be arbitrarily small with sufficiently large batchsize $N$ (for reducing the variance), small $\delta$ (for reducing the bias), and smaller policy evaluation error $\epsilon_V$.

**Intuition and Novelty for Proving Proposition 1:** Unlike existing zeroth-order estimations on the whole Euclidean space, our estimation (30) is made on the policy space $\Pi$, which lies in the linear manifold $\mathcal{L}_0 + |\mathcal{A}|^{-1} \subset \mathbb{R}^{|\mathcal{S}||\mathcal{A}|}$. The key to our proof is to find an orthogonal transformation $T : \mathbb{R}^{|\mathcal{S}|(|\mathcal{A}|-1)} \to \mathcal{L}_0$, so that the goal is simplified to analyze the gradient estimation of $f_\lambda(x) \stackrel{\text{def}}{=} V^{T(x)+|\mathcal{A}|^{-1}}_{\lambda,T(x)+|\mathcal{A}|^{-1}}$ on any $x \in \mathbb{R}^{|\mathcal{S}|(|\mathcal{A}|-1)}$. In particular, the true gradient can be rewritten as $\nabla f_\lambda(x) = T^{-1}\left( \text{proj}_{\mathcal{L}_0} \nabla_\pi V^\pi_{\lambda,\pi} \big|_{\pi=T(x)+|\mathcal{A}|^{-1}} \right)$ using differentiability, and when $\epsilon_V = 0$ (i.e., $\hat{V}^\pi_{\lambda,\pi} = V^\pi_{\lambda,\pi}$ for any $\pi \in \Pi$), our estimated gradient (30) on the policy space $\Pi$ can be rewritten as the following two-point estimator on $\mathbb{R}^{|\mathcal{S}|(|\mathcal{A}|-1)}$ (see Eq. (112) for details).

$$\hat{g}_{\lambda,\delta}(\pi) = \frac{|\mathcal{S}|(|\mathcal{A}| - 1)}{2N\delta} \sum_{i=1}^N \left[ f_\lambda(x + \delta \tilde{u}_i) - f_\lambda(x - \delta \tilde{u}_i) \right] \tilde{u}_i,$$

where $\tilde{u}_i = T^{-1}(u_i)$ is uniformly distributed on a unit sphere in $\mathbb{R}^{|\mathcal{S}|(|\mathcal{A}|-1)}$ and $x = T^{-1}(\pi - |\mathcal{A}|^{-1})$. Therefore, we can apply estimation analysis to the Euclidean space $\mathbb{R}^{|\mathcal{S}|(|\mathcal{A}|-1)}$. Finally, it is straightforward extend the conclusion from $\epsilon_V = 0$ to $\epsilon_V > 0$ by adding the policy evaluation error terms (see Eq. (116)).

**Algorithm 1** Zeroth-Order Performative Policy Gradient (0-PPG) Algorithm

---

1: **Inputs:** $T, N, \Delta > \delta > 0, \epsilon_V \geq 0, \beta > 0$.
2: **Initialize:** policy $\pi_0 \in \Pi_\Delta$.
3: **for** Iterations $t = 0, 1, \ldots, T-1$ **do**
4:     Obtain i.i.d. vectors $\{v_i\}_{i=1}^N$ uniformly from the unit sphere $U_1 \overset{\text{def}}{=} \{u \in \mathbb{R}^{|\mathcal{S}||\mathcal{A}|} : \|u\| = 1\}$.
5:     Obtain $\{\text{proj}_{\mathcal{L}_0}(v_i)\}_{i=1}^N$ from Eq. (28).
6:     Obtain $\{u_i\}_{i=1}^N$ by Eq. (29).
7:     Obtain stochastic policy evaluation $\hat{V}_{\lambda,\pi}^\pi \approx V_{\lambda,\pi}^\pi$ for $\pi \in \{\pi_t \pm \delta u_i\}_{i=1}^N$ with error bound (24).
8:     Obtain stochastic performative policy gradient estimation $\hat{g}_{\lambda,\delta}(\pi_t)$ using Eq. (26).
9:     Obtain $\tilde{\pi}_t$ by Eq. (33).
10:    Update $\pi_{t+1}$ by Eq. (32).
11: **end for**
12: **Output:** $\pi_{\tilde{T}}$ where
$$\widetilde{T} \in \arg\min_{0 \leq t \leq T-1} \langle \hat{g}_{\lambda,\delta}(\pi_t), \tilde{\pi}_t - \pi_t \rangle.$$

---

## 4.2. Zeroth-Order Performative Policy Gradient (0-PPG) Algorithm

With the estimated gradient $\hat{g}_{\lambda,\delta}(\pi_t)$ defined by Eq. (26), we can consider the following Frank-Wolfe algorithm to find an $\epsilon$-stationary policy.

$$\tilde{\pi}_t = \arg\max_{\pi \in \Pi_\Delta} \langle \pi, \hat{g}_{\lambda,\delta}(\pi_t) \rangle, \tag{31}$$

$$\pi_{t+1} = \pi_t + \beta(\tilde{\pi}_t - \pi_t). \tag{32}$$

**Lemma 1.** *The step* (31) *has the analytical solution below.*

$$\tilde{\pi}_t(a|s) = \begin{cases} \Delta; a \neq \tilde{a}_t(s) \\ 1 - \Delta(|\mathcal{A}| - 1); a = \tilde{a}_t(s) \end{cases}, \tag{33}$$

*where* $\tilde{a}_t(s) \in \arg\max_a \hat{g}_{\lambda,\delta}(\pi_t)(a|s)$.

See the proof of Lemma 1 in Section A.1. Then combining the *performative policy gradient* estimation (see Section 3.1) with the Frank-Wolfe algorithm, we propose our zeroth-order performative policy gradient (0-PPG) algorithm (see Algorithm 1). We obtain the following convergence result of Algorithm 1 in Theorem 4, the main theoretical result of this work, as follows.

**Theorem 4.** *Suppose Assumptions 1-3 hold. For any* $0 < \epsilon \leq \min\left[24\sqrt{2|\mathcal{S}|}\frac{\ell_\lambda}{D}, \frac{2\lambda}{5|\mathcal{A}|D^2(1-\gamma)}, \frac{288L_\lambda|\mathcal{S}|^{1.5}|\mathcal{A}|}{D\pi_{\min}}\right]$ *and* $\eta \in (0, 1)$, *select the following hyperparameters for Algorithm 1:* $\Delta = \frac{\pi_{\min}}{3}$, $\beta = \frac{D\pi_{\min}\epsilon}{36\ell_\lambda|\mathcal{S}|}$, $\delta = \mathcal{O}(\epsilon)$, $\epsilon_V = \mathcal{O}(\epsilon^2)$, $N = \mathcal{O}[\epsilon^{-2}\log(\eta^{-1}\epsilon^{-1})]$, *and the number of iterations* $T = \mathcal{O}(\epsilon^{-2})$ *(see Eqs.* (122)-(127) *in Appendix J for detailed expression of these hyperparameters). Then with probability at least* $1 - \eta$, *the output policy* $\tilde{\pi}_{\tilde{T}}$ *of Algorithm*

1 *is an* $D\epsilon$-*stationary policy. Furthermore, if* $\mu \geq 0$, $\tilde{\pi}_{\tilde{T}}$ *is also an* $\epsilon$-*PO policy. The total number of policy evaluations is* $2NT = \mathcal{O}[\epsilon^{-4}\log(\eta^{-1}\epsilon^{-1})]$.

**Comparison with Existing Works:** Theorem 4 indicates that our 0-PPG algorithm for the first time converges to the desire PO policy with arbitrarily small precision $\epsilon$ in polynomial computation complexity, under the mild regularizer dominance condition that $\mu \geq 0$. In contrast, existing works only converge to a suboptimal PS policy under a similar regularizer dominance condition (Mandal et al., 2023; Rank et al., 2024; Mandal and Radanovic, 2024). Our preferable convergence result is due to the major algorithmic difference that existing works adopt repeated retraining algorithms with iteration $\pi_{t+1} \approx \arg\max_{\pi \in \Pi} V_{\lambda,\pi}^{\pi_t}$ where the policy $\pi$ is deployed in a fixed environment $\mathcal{M}_{\pi_t}$ with $\pi \neq \pi_t$, whereas our 0-PPG algorithm evaluates $V_{\lambda,\pi}^\pi$ where each policy $\pi$ is always deployed at its corresponding environment $\mathcal{M}_\pi$.

**Intuition and Novelty for Proving Theorem 4:** Standard convergence analysis of Frank-Wolfe algorithm yields that $\max_{\tilde{\pi} \in \Pi_\Delta} \langle \nabla_\pi V_{\lambda,\pi_{\tilde{T}}}^{\pi_{\tilde{T}}}, \tilde{\pi} - \pi_{\tilde{T}} \rangle \leq \frac{D\epsilon}{2}$ on $\Pi_\Delta$. However, it requires a trick to prove the following Proposition 2 which implies that $\pi_{\tilde{T}}$ is $D\epsilon$-stationary on $\Pi$.

**Proposition 2.** *If* $\Delta \leq \pi_{\min}/3$ *and a policy* $\pi$ *satisfies* $\max_{\tilde{\pi} \in \Pi_\Delta} \langle \nabla_\pi V_{\lambda,\pi}^\pi, \tilde{\pi} - \pi \rangle \leq \frac{D\lambda}{5|\mathcal{A}|(1-\gamma)}$, *then the stationary measures on* $\Pi_\Delta$ *and* $\Pi$ *bound each other as follows.*

$$\max_{\tilde{\pi} \in \Pi} \langle \nabla_\pi V_{\lambda,\pi}^\pi, \tilde{\pi} - \pi \rangle \leq 2 \max_{\tilde{\pi} \in \Pi_\Delta} \langle \nabla_\pi V_{\lambda,\pi}^\pi, \tilde{\pi} - \pi \rangle \tag{34}$$

To prove Proposition 2, note that $\pi'$ defined by Eq. (19) also belongs to $\Pi_\Delta$, so Theorem 2 implies $\pi(a|s) \geq 2\Delta$. Then for any $\pi_2 \in \Pi$, we have $\frac{\pi_2 + \pi}{2} \in \Pi_\Delta$ and thus

$$\max_{\pi_2 \in \Pi} \langle \nabla_\pi V_{\lambda,\pi}^\pi, \pi_2 - \pi \rangle = 2 \max_{\pi_2 \in \Pi} \left\langle \nabla_\pi V_{\lambda,\pi}^\pi, \frac{\pi_2 + \pi}{2} - \pi \right\rangle$$
$$\leq 2 \max_{\tilde{\pi} \in \Pi_\Delta} \langle \nabla_\pi V_{\lambda,\pi}^\pi, \tilde{\pi} - \pi \rangle.$$

## 5. Conclusion

We have studied an entropy-regularized performative reinforcement learning problem, obtained its important properties including gradient dominance, policy lower bound, Lipschitz continuity and smoothness. Based on these properties, we have proposed a zeroth-order performative policy gradient (0-PPG) algorithm only using sample-based policy evaluation, which for the first time converges to a *performatively optimal (PO)* policy with polynomial number of policy evaluations under the regularizer dominance condition. These theoretical results also holds for the quadratice regularizers used in the existing works on performative reinforcement learning (see Appendix K for discussion). A future direction is to extend the algorithm and results to more practical environments of large state and action spaces.

## Impact Statement

This paper presents work whose goal is to advance the field of Machine Learning. There are many potential societal consequences of our work, none which we feel must be specifically highlighted here.

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

# Appendix

## Table of Contents

## A. Supporting Lemmas

### A.1. Frank-Wolfe Step

We repeat Lemma 1 as follows.

**Lemma 2.** *The step* (31) *has the following analytical solution.*

$$\tilde{\pi}_t(a|s) = \begin{cases} \Delta; a \neq \tilde{a}_t(s) \\ 1 - \Delta(|\mathcal{A}| - 1); a = \tilde{a}_t(s) \end{cases}, \tag{35}$$

*where* $\tilde{a}_t(s) \in \arg\max_a \hat{g}_{\lambda,\delta}(\pi_t)(a|s)$.

*Proof.* For $\tilde{\pi}_t$ defined by Eq. (35) and for any $\pi \in \Pi_\Delta$, we have

$$\langle \tilde{\pi}_t - \pi, \hat{g}_{\lambda,\delta}(\pi_t) \rangle$$
$$= \sum_{s,a} \hat{g}_{\lambda,\delta}(\pi_t)(a|s)[\tilde{\pi}_t(a|s) - \pi(a|s)]$$

$$= \sum_s \left\{ \hat{g}_{\lambda,\delta}(\pi_t)[\tilde{a}_t(s)|s]\big[1 - \Delta(|\mathcal{A}| - 1) - \pi[\tilde{a}_t(s)|s]\big] - \sum_{a \neq \tilde{a}_t(s)} \hat{g}_{\lambda,\delta}(\pi_t)(a|s)[\pi(a|s) - \Delta] \right\}$$

$$\overset{(a)}{\geq} \sum_s \left\{ \hat{g}_{\lambda,\delta}(\pi_t)[\tilde{a}_t(s)|s]\big[1 - \Delta(|\mathcal{A}| - 1) - \pi[\tilde{a}_t(s)|s]\big] - \sum_{a \neq \tilde{a}_t(s)} \hat{g}_{\lambda,\delta}(\pi_t)[\tilde{a}_t(s)|s][\pi(a|s) - \Delta] \right\}$$

$$= \sum_s \left\{ \hat{g}_{\lambda,\delta}(\pi_t)[\tilde{a}_t(s)|s]\big[1 - \Delta(|\mathcal{A}| - 1) - \pi[\tilde{a}_t(s)|s]\big] - \hat{g}_{\lambda,\delta}(\pi_t)[\tilde{a}_t(s)|s]\big[1 - \pi[\tilde{a}_t(s)|s] - \Delta(|\mathcal{A}| - 1)\big] \right\}$$

$$= 0,$$

where (a) uses $\pi(a|s) - \Delta \geq 0$ and $\hat{g}_{\lambda,\delta}(\pi_t)(a|s) \leq \hat{g}_{\lambda,\delta}(\pi_t)[\tilde{a}_t(s)|s]$. Therefore, Eq. (31) holds, that is, $\tilde{\pi}_t = \arg\max_{\pi \in \Pi_\Delta} \langle \pi, \hat{g}_{\lambda,\delta}(\pi_t) \rangle$. $\square$

### A.2. Lipschitz Property of Occupany Measure

**Lemma 3.** *The occupancy measure $d_{\pi,p}$ defined by Eq. (2) has the following Lipschitz properties for any $\pi, \pi' \in \Pi$, $p, p' \in \mathcal{P}$ and $\tilde{s} \in \mathcal{S}$.*

$$\sum_s |d_{\pi',p}(s) - d_{\pi,p}(s)| \leq \frac{\gamma}{1 - \gamma} \max_s \|\pi'(\cdot|s) - \pi(\cdot|s)\|_1 \leq \frac{\gamma\sqrt{|\mathcal{A}|}}{1 - \gamma} \|\pi' - \pi\| \tag{36}$$

$$\sum_s |d_{\pi,p'}(s) - d_{\pi,p}(s)| \leq \frac{\gamma}{1 - \gamma} \max_{s,a} \|p'(\cdot|s,a) - p(\cdot|s,a)\|_1 \leq \frac{\gamma\sqrt{|\mathcal{S}|}}{1 - \gamma} \|p' - p\| \tag{37}$$

$$\sum_{s,a} |d_{\pi',p'}(s,a) - d_{\pi,p}(s,a)| \leq \frac{1}{1 - \gamma} \max_s \|\pi'(\cdot|s) - \pi(\cdot|s)\|_1 + \frac{\gamma}{1 - \gamma} \max_{s,a} \|p'(\cdot|s,a) - p(\cdot|s,a)\|_1$$

$$\leq \frac{\sqrt{|\mathcal{A}|}}{1 - \gamma} \|\pi' - \pi\| + \frac{\gamma\sqrt{|\mathcal{S}|}}{1 - \gamma} \|p' - p\| \tag{38}$$

*Proof.* The first $\leq$ of Eqs. (36) and (37) follows from Lemma 5 of (Chen and Huang, 2024). The second $\leq$ of Eqs. (36) and (37) uses $\|x\|_1 \leq \sqrt{d}\|x\|$ for any $x \in \mathbb{R}^d$.

Eq. (38) can be proved as follows.

$$\sum_{s,a} |d_{\pi',p'}(s,a) - d_{\pi,p}(s,a)|$$

$$= \sum_{s,a} |d_{\pi',p'}(s)\pi'(a|s) - d_{\pi,p}(s)\pi(a|s)|$$

$$\leq \sum_{s,a} d_{\pi',p'}(s)|\pi'(a|s) - \pi(a|s)| + \pi(a|s)|d_{\pi',p'}(s) - d_{\pi,p}(s)|$$

$$\leq \sum_s [d_{\pi',p'}(s) \max_{s'} \|\pi'(\cdot|s') - \pi(\cdot|s')\|_1] + \sum_s |d_{\pi',p'}(s) - d_{\pi,p}(s)|$$

$$\overset{(a)}{\leq} \max_{s'} \|\pi'(\cdot|s') - \pi(\cdot|s')\|_1 + \frac{\gamma}{1 - \gamma} \max_s \|\pi'(\cdot|s) - \pi(\cdot|s)\|_1 + \frac{\gamma}{1 - \gamma} \max_{s,a} \|p'(\cdot|s,a) - p(\cdot|s,a)\|_1$$

$$\leq \frac{1}{1 - \gamma} \max_s \|\pi'(\cdot|s) - \pi(\cdot|s)\|_1 + \frac{\gamma}{1 - \gamma} \max_{s,a} \|p'(\cdot|s,a) - p(\cdot|s,a)\|_1$$

$$\leq \frac{\sqrt{|\mathcal{A}|}}{1 - \gamma} \|\pi' - \pi\| + \frac{\gamma\sqrt{|\mathcal{S}|}}{1 - \gamma} \|p' - p\|,$$

where (a) uses Eqs. (36) and (37).

$\square$

### A.3. Various Value Functions

Define the following value functions.

$$J_\lambda(\pi, \pi', p, r) \stackrel{\text{def}}{=} \mathbb{E}_{\pi,p}\Big[ \sum_{t=0}^\infty \gamma^t [r(s_t, a_t) - \lambda \log \pi'(a_t|s_t)] \Big| s_0 \sim \rho \Big]$$

$$= \frac{1}{1-\gamma} \sum_{s,a} d_{\pi,p}(s,a)[r(s,a) - \lambda \log \pi'(a|s)], \tag{39}$$

$$V_\lambda(\pi, \pi', p, r; s) \stackrel{\text{def}}{=} \mathbb{E}_{\pi,p}\Big[ \sum_{t=0}^\infty \gamma^t [r(s_t, a_t) - \lambda \log \pi'(a_t|s_t)] \Big| s_0 = s \Big], \tag{40}$$

$$Q_\lambda(\pi, \pi', p, r; s, a) \stackrel{\text{def}}{=} \mathbb{E}_{\pi,p}\Big[ \sum_{t=0}^\infty \gamma^t [r(s_t, a_t) - \lambda \log \pi'(a_t|s_t)] \Big| s_0 = s, a_0 = a \Big]$$

$$= r(s,a) - \lambda \log \pi'(a|s) + \gamma \sum_{s'} p(s'|s,a) V_\lambda(\pi, \pi', p, r; s'). \tag{41}$$

Note that the value function (8) of interest can be rewritten into the above functions as follows.

$$V_{\lambda,\pi'}^\pi = J_\lambda(\pi, \pi, p_{\pi'}, r_{\pi'}) = \sum_s \rho(s) V_\lambda(\pi, \pi, p_{\pi'}, r_{\pi'}; s) = \sum_{s,a} \rho(s)\pi(a|s) Q_\lambda(\pi, \pi, p_{\pi'}, r_{\pi'}; s, a). \tag{42}$$

Hence, we will investigate the properties of the value functions (39)-(41) as follows.

**Lemma 4.** *For any* $\pi \in \Pi$, $p \in \mathcal{P}$, $r \in \mathcal{R}$, *we have* $V_{\lambda,\pi}^\pi$, $J_\lambda(\pi, \pi, p, r)$, $V_\lambda(\pi, \pi, p, r; s)$, $Q_\lambda(\pi, \pi, p, r; s, a) \in \left[0, \frac{1+\lambda \log |\mathcal{A}|}{1-\gamma}\right]$.

*Proof.* We will prove the range of $J_\lambda(\pi, \pi, p, r)$ as follows using $r(s,a) \in [0, 1]$. The proof for the other value functions follow the same way.

$$0 \le J_\lambda(\pi, \pi, p, r) = \mathbb{E}_{\pi,p,\rho}\Big[ \sum_{t=0}^\infty \gamma^t [r(s_t, a_t) - \lambda \log \pi(a_t|s_t)] \Big]$$

$$\le \sum_{t=0}^\infty \gamma^t + \lambda \mathbb{E}_{\pi,p,\rho}\Big[ \sum_{t=0}^\infty \gamma^t \sum_a [-\pi(a|s_t) \log \pi(a|s_t)] \Big]$$

$$\le \frac{1}{1-\gamma} + \lambda \sum_{t=0}^\infty \gamma^t \log |\mathcal{A}|$$

$$\le \frac{1+\lambda \log |\mathcal{A}|}{1-\gamma}.$$

$\square$

**Lemma 5.** *The gradients of* $J_\lambda(\pi, \pi', p, r)$ *defined by Eq.* (39) *have the following expressions.*

$$\frac{\partial J_\lambda(\pi, \pi', p, r)}{\partial \pi(a|s)} = \frac{d_{\pi,p}(s) Q_\lambda(\pi, \pi', p, r; s, a)}{1-\gamma}, \tag{43}$$

$$\frac{\partial J_\lambda(\pi, \pi', p, r)}{\partial \pi'(a|s)} = -\frac{\lambda d_{\pi,p}(s,a)}{(1-\gamma)\pi'(a|s)}, \tag{44}$$

$$\frac{\partial J_\lambda(\pi, \pi', p, r)}{\partial p(s'|s,a)} = \frac{d_{\pi,p}(s,a)}{1-\gamma}\big[r(s,a) - \lambda \log \pi'(a|s) + \gamma V_\lambda(\pi, \pi', p, r; s')\big], \tag{45}$$

$$\frac{\partial J_\lambda(\pi, \pi', p, r)}{\partial r(s,a)} = \frac{d_{\pi,p}(s,a)}{1-\gamma}, \tag{46}$$

$$\frac{\partial J_\lambda(\pi, \pi, p, r)}{\partial \pi(a|s)} = \frac{d_{\pi,p}(s)[Q_\lambda(\pi, \pi, p, r; s, a) - \lambda]}{1-\gamma}. \tag{47}$$

*Proof.* Eq. (43) follows from the policy gradient expression in Eq. (7) of (Agarwal et al., 2021), with reward function $r(s,a)$ replaced by $r(s,a) - \lambda \log \pi'(a|s)$.

Eq. (45) can be proved as follows.

$$p(s'|s,a) \overset{(a)}{=} \frac{d_{\pi,p}(s)\pi(a|s)}{1-\gamma}\big[r(s,a) - \lambda \log \pi(a|s) + \gamma V_\lambda(\pi, \pi', p, r; s')\big]$$

$$= \frac{d_{\pi,p}(s,a)}{1-\gamma}\big[r(s,a) - \lambda \log \pi(a|s) + \gamma V_\lambda(\pi, \pi', p, r; s')\big],$$

where (a) uses Eq. (9) in (Chen and Huang, 2024).

Eqs. (44) and (46) can be proved by taking derivatives of Eq. (39).

Based on the chain rule, Eq. (47) can be proved as follows by adding Eqs. (43) and (44) with $\pi' = \pi$.

$$\frac{\partial J_\lambda(\pi, \pi, p, r)}{\partial \pi(a|s)} = \Big[\frac{\partial J_\lambda(\pi, \pi', p, r)}{\partial \pi(a|s)} + \frac{\partial J_\lambda(\pi, \pi', p, r)}{\partial \pi'(a|s)}\Big]\Big|_{\pi'=\pi}$$

$$= \frac{d_{\pi,p}(s)Q_\lambda(\pi, \pi, p, r; s, a)}{1-\gamma} - \frac{\lambda d_{\pi,p}(s,a)}{(1-\gamma)\pi(a|s)}$$

$$= \frac{d_{\pi,p}(s)[Q_\lambda(\pi, \pi, p, r; s, a) - \lambda]}{1-\gamma},$$

where the final $=$ uses $d_{\pi,p}(s,a) = d_{\pi,p}(s)\pi(a|s)$. $\qquad\square$

**Lemma 6.** *The function $J_\lambda$ defined by eq. (39) has the following Lipschitz properties for any $\pi, \pi' \in \Pi$, $p, p' \in \mathcal{P}$ and $r, r' \in \mathcal{R}$.*

$$|J_\lambda(\pi', \pi', p, r) - J_\lambda(\pi, \pi, p, r)| \le L_\pi \max_s \|\log \pi'(\cdot|s) - \log \pi(\cdot|s)\| \tag{48}$$

$$|J_\lambda(\pi, \pi, p', r) - J_\lambda(\pi, \pi, p, r)| \le L_p \|p' - p\| \tag{49}$$

$$|J_\lambda(\pi, \pi, p, r') - J_\lambda(\pi, \pi, p, r)| \le \frac{\|r' - r\|_\infty}{1-\gamma} \le \frac{\|r' - r\|}{1-\gamma} \tag{50}$$

$$\|\nabla_p J_\lambda(\pi', \pi', p, r) - \nabla_p J_\lambda(\pi, \pi, p, r)\| \le \ell_\pi \max_s \|\log \pi'(\cdot|s) - \log \pi(\cdot|s)\| \tag{51}$$

$$\|\nabla_p J_\lambda(\pi, \pi, p', r) - \nabla_p J_\lambda(\pi, \pi, p, r)\| \le \ell_p \|p' - p\| \tag{52}$$

$$\|\nabla_p J_\lambda(\pi', \pi', p', r') - \nabla_p J_\lambda(\pi, \pi, p, r)\| \le \ell_\pi \max_s \|\log \pi'(\cdot|s) - \log \pi(\cdot|s)\| + \ell_p \|p' - p\| + \frac{\sqrt{|\mathcal{S}|}}{(1-\gamma)^2}\|r' - r\|_\infty \tag{53}$$

$$\|\nabla_r J_\lambda(\pi', \pi', p', r') - \nabla_r J_\lambda(\pi, \pi, p, r)\| \le \frac{\max_s \|\pi'(\cdot|s) - \pi(\cdot|s)\|_1 + \gamma \max_{s,a} \|p'(\cdot|s,a) - p(\cdot|s,a)\|_1}{(1-\gamma)^2} \tag{54}$$

$$\|\nabla_\pi J_\lambda(\pi', \pi', p', r') - \nabla_\pi J_\lambda(\pi, \pi, p, r)\| \le \Big(\frac{|\mathcal{A}|(1 + 2\lambda \log|\mathcal{A}|)}{(1-\gamma)^2} + \gamma L_\pi\Big) \max_s \|\log \pi'(\cdot|s) - \log \pi(\cdot|s)\|$$

$$+ \gamma\sqrt{|\mathcal{A}|}\Big[\frac{2\sqrt{|\mathcal{S}|}(1 + \lambda \log|\mathcal{A}|)}{(1-\gamma)^2} + L_p\Big]\|p' - p\| + \frac{\sqrt{|\mathcal{A}|}\|r' - r\|_\infty}{1-\gamma}, \tag{55}$$

*where $L_\pi := \frac{\sqrt{|\mathcal{A}|}(2 - \gamma + \gamma\lambda \log|\mathcal{A}|)}{(1-\gamma)^2}$, $L_p := \frac{\sqrt{|\mathcal{S}|}(1 + \lambda \log|\mathcal{A}|)}{(1-\gamma)^2}$, $\ell_\pi := \frac{\sqrt{|\mathcal{S}||\mathcal{A}|}(2 + 3\gamma\lambda \log|\mathcal{A}|)}{(1-\gamma)^3}$ and $\ell_p := \frac{2\gamma|\mathcal{S}|(1 + \lambda \log|\mathcal{A}|)}{(1-\gamma)^3}$.*

*Proof.* Eqs. (48), (49), (51) and (52) directly follow from Lemma 6 of (Chen and Huang, 2024). Eq. (50) can be proved as follows.

$$|J_\lambda(\pi, p, r') - J_\lambda(\pi, p, r)| = \Big|\frac{1}{1-\gamma}\sum_{s,a} d_{\pi,p}(s,a)[r'(s,a) - r(s,a)]\Big|$$

$$\le \frac{1}{1-\gamma}\sum_{s,a} d_{\pi,p}(s,a)|r'(s,a) - r(s,a)|$$

$$= \frac{1}{1-\gamma} \sum_{s,a} d_{\pi,p}(s,a) \|r' - r\|_\infty$$

$$= \frac{1}{1-\gamma} \|r' - r\|_\infty \leq \frac{1}{1-\gamma} \|r' - r\|.$$

To prove Eq. (53), note that

$$\left| \frac{\partial J_\lambda(\pi, \pi, p, r')}{\partial p(s'|s, a)} - \frac{\partial J_\lambda(\pi, \pi, p, r)}{\partial p(s'|s, a)} \right|$$

$$\overset{(a)}{=} \frac{d_{\pi,p}(s,a)}{1-\gamma} \left| r'(s,a) - r(s,a) + \gamma[V_\lambda(\pi, \pi', p, r'; s') - V_\lambda(\pi, \pi', p, r; s')] \right|$$

$$\overset{(b)}{\leq} \frac{d_{\pi,p}(s,a)}{1-\gamma} \left[ \|r' - r\|_\infty + \gamma \sum_{t=0}^{\infty} \gamma^t \|r' - r\|_\infty \right]$$

$$\leq \frac{d_{\pi,p}(s,a)}{(1-\gamma)^2} \|r' - r\|_\infty \tag{56}$$

where (a) uses Eq. (45) and (b) uses Eq. (40). Therefore, we can prove Eq. (53) as follows.

$$\|\nabla_p J_\lambda(\pi', \pi', p', r') - \nabla_p J_\lambda(\pi, \pi, p, r)\|$$

$$\leq \|\nabla_p J_\lambda(\pi', \pi', p', r') - \nabla_p J_\lambda(\pi, \pi, p', r')\| + \|\nabla_p J_\lambda(\pi, \pi, p', r') - \nabla_p J_\lambda(\pi, \pi, p, r')\|$$

$$+ \|\nabla_p J_\lambda(\pi, \pi, p, r') - \nabla_p J_\lambda(\pi, \pi, p, r)\|$$

$$\overset{(a)}{\leq} \ell_\pi \max_s \|\log \pi'(\cdot|s) - \log \pi(\cdot|s)\| + \ell_p \|p' - p\| + \sqrt{\sum_{s,a,s'} \left| \frac{\partial J_\lambda(\pi, \pi, p, r')}{\partial p(s'|s, a)} - \frac{\partial J_\lambda(\pi, \pi, p, r)}{\partial p(s'|s, a)} \right|^2}$$

$$\overset{(b)}{\leq} \ell_\pi \max_s \|\log \pi'(\cdot|s) - \log \pi(\cdot|s)\| + \ell_p \|p' - p\| + \sqrt{\frac{\|r' - r\|_\infty^2}{(1-\gamma)^4} \sum_{s,a,s'} d_{\pi,p}^2(s,a)}$$

$$\leq \ell_\pi \max_s \|\log \pi'(\cdot|s) - \log \pi(\cdot|s)\| + \ell_p \|p' - p\| + \frac{\sqrt{|\mathcal{S}|}}{(1-\gamma)^2} \|r' - r\|_\infty,$$

where (a) uses Eqs. (51)-(52) and (b) uses Eq. (56).

Then, we prove Eq. (54) as follows.

$$\|\nabla_r J_\lambda(\pi', \pi', p', r') - \nabla_r J_\lambda(\pi, \pi, p, r)\|$$

$$\overset{(a)}{=} \frac{\|d_{\pi',p'} - d_{\pi,p}\|}{1-\gamma}$$

$$\leq \frac{\|d_{\pi',p'} - d_{\pi,p}\|_1}{1-\gamma}$$

$$\overset{(b)}{\leq} \frac{1}{(1-\gamma)^2} \max_s \|\pi'(\cdot|s) - \pi(\cdot|s)\|_1 + \frac{\gamma}{(1-\gamma)^2} \max_{s,a} \|p'(\cdot|s,a) - p(\cdot|s,a)\|_1,$$

where (a) uses Eq. (46), (b) uses Eq. (38).

To prove Eq. (55), we will first prove the following auxiliary bounds.

$$Q_\lambda(\pi, \pi, p, r; s, a) - \lambda \overset{(a)}{\in} \left[ -\lambda, \frac{1 + \lambda \log |\mathcal{A}|}{1-\gamma} - \lambda \right] \Rightarrow |Q_\lambda(\pi, \pi, p, r; s, a) - \lambda| \leq \frac{1 + \lambda \log |\mathcal{A}|}{1-\gamma}, \tag{57}$$

where (a) uses Lemma 4.

$$|V_\lambda(\pi', \pi', p', r'; s) - V_\lambda(\pi, \pi, p, r; s)|$$

$$\leq |V_\lambda(\pi', \pi', p', r'; s) - V_\lambda(\pi, \pi, p', r'; s)| + |V_\lambda(\pi, \pi, p', r'; s) - V_\lambda(\pi, \pi, p, r'; s)| + |V_\lambda(\pi, \pi, p, r'; s) - V_\lambda(\pi, \pi, p, r; s)|$$

$$\overset{(a)}{\le} L_\pi \max_s \| \log \pi'(\cdot|s) - \log \pi(\cdot|s) \| + L_p \| p' - p \| + \frac{\| r' - r \|_\infty}{1 - \gamma}, \tag{58}$$

where (a) applies Eqs. (48)-(50) to the case where the initial state distribution $\rho$ is probability 1 at $s$ (so $J_\lambda(\pi, \pi, p, r)$ becomes $V_\lambda(\pi, \pi, p, r; s)$).

$$|Q_\lambda(\pi, \pi, p, r'; s, a) - Q_\lambda(\pi, \pi, p, r; s, a)|$$

$$\overset{(a)}{=} \left| \mathbb{E}_{\pi, p} \Big[ \sum_{t=0}^\infty \gamma^t [r'(s_t, a_t) - r(s_t, a_t)] \Big| s_0 = s, a_0 = a \Big] \right|$$

$$\le \mathbb{E}_{\pi, p} \Big[ \sum_{t=0}^\infty \gamma^t [r'(s_t, a_t) - r(s_t, a_t)] \Big| \Big| s_0 = s, a_0 = a \Big]$$

$$\le \mathbb{E}_{\pi, p} \Big[ \sum_{t=0}^\infty \gamma^t \| r' - r \|_\infty \Big| s_0 = s, a_0 = a \Big]$$

$$\le \frac{\| r' - r \|_\infty}{1 - \gamma}, \tag{59}$$

where (a) uses Eq. (41).

$$|Q_\lambda(\pi', \pi', p', r; s, a) - Q_\lambda(\pi, \pi, p, r; s, a)|$$

$$\overset{(a)}{\le} \lambda |\log \pi'(a|s) - \log \pi(a|s)| + \gamma \Big| \sum_{s'} [p'(s'|s, a) V_\lambda(\pi', \pi', p', r; s) - p(s'|s, a) V_\lambda(\pi, \pi, p, r; s)] \Big|$$

$$\le \lambda |\log \pi'(a|s) - \log \pi(a|s)| + \gamma \sum_{s'} p'(s'|s, a) |V_\lambda(\pi', \pi', p', r; s) - V_\lambda(\pi, \pi, p, r; s)|$$

$$+ \gamma \sum_{s'} |p'(s'|s, a) - p(s'|s, a)| |V_\lambda(\pi, \pi, p, r; s)|$$

$$\overset{(b)}{\le} \lambda |\log \pi'(a|s) - \log \pi(a|s)| + \gamma L_\pi \max_{s'} \| \log \pi'(\cdot|s') - \log \pi(\cdot|s') \| + \gamma L_p \| p' - p \|$$

$$+ \frac{\gamma (1 + \lambda \log |\mathcal{A}|)}{1 - \gamma} \| p'(\cdot|s, a) - p(\cdot|s, a) \|_1, \tag{60}$$

where (a) uses Eq. (41), and (b) uses Eq. (58) and Lemma 4.

Note that

$$(1 - \gamma) \Big| \frac{\partial J_\lambda(\pi', \pi', p', r')}{\partial \pi'(a|s)} - \frac{\partial J_\lambda(\pi, \pi, p, r)}{\partial \pi(a|s)} \Big|$$

$$\overset{(a)}{=} \big| d_{\pi', p'}(s) [Q_\lambda(\pi', \pi', p', r'; s, a) - \lambda] - d_{\pi, p}(s) [Q_\lambda(\pi, \pi, p, r; s, a) - \lambda] \big|$$

$$\le \big| [d_{\pi', p'}(s) - d_{\pi, p}(s)] [Q_\lambda(\pi', \pi', p', r'; s, a) - \lambda] + d_{\pi, p}(s) [Q_\lambda(\pi', \pi', p', r'; s, a) - Q_\lambda(\pi', \pi', p', r; s, a)]$$

$$+ d_{\pi, p}(s) [Q_\lambda(\pi', \pi', p', r; s, a) - Q_\lambda(\pi, \pi, p, r; s, a)] \big|$$

$$\le \big| d_{\pi', p'}(s) - d_{\pi, p}(s) \big| \cdot \big| Q_\lambda(\pi', \pi', p', r'; s, a) - \lambda \big| + d_{\pi, p}(s) \big| Q_\lambda(\pi', \pi', p', r'; s, a) - Q_\lambda(\pi', \pi', p', r; s, a) \big|$$

$$+ d_{\pi, p}(s) \big| Q_\lambda(\pi', \pi', p', r; s, a) - Q_\lambda(\pi, \pi, p, r; s, a) \big|$$

$$\overset{(b)}{\le} \frac{1 + \lambda \log |\mathcal{A}|}{1 - \gamma} \big| d_{\pi', p'}(s) - d_{\pi, p}(s) \big| + \frac{d_{\pi, p}(s) \| r' - r \|_\infty}{1 - \gamma} + d_{\pi, p}(s) \Big[ \lambda |\log \pi'(a|s) - \log \pi(a|s)|$$

$$+ \gamma L_\pi \max_{s'} \| \log \pi'(\cdot|s') - \log \pi(\cdot|s') \| + \gamma L_p \| p' - p \| + \frac{\gamma (1 + \lambda \log |\mathcal{A}|)}{1 - \gamma} \| p'(\cdot|s, a) - p(\cdot|s, a) \|_1 \Big],$$

where (a) uses Eq. (47), (b) uses Eqs. (57), (59) and (60). Applying triangular inequality to the bound above, we can prove Eq. (55) as follows.

$$(1 - \gamma) \big\| \nabla_{\pi'} J_\lambda(\pi', \pi', p', r') - \nabla_\pi J_\lambda(\pi, \pi, p, r) \big\|$$

$$\leq \frac{1 + \lambda \log |\mathcal{A}|}{1 - \gamma} \sqrt{\sum_{s,a} |d_{\pi',p'}(s) - d_{\pi,p}(s)|^2} + \frac{\|r' - r\|_\infty}{1 - \gamma} \sqrt{\sum_{s,a} d_{\pi,p}(s)^2} + \lambda \sqrt{\sum_{s,a} d_{\pi,p}(s)^2 |\log \pi'(a|s) - \log \pi(a|s)|^2}$$

$$+ \left[ \gamma L_\pi \max_{s'} \|\log \pi'(\cdot|s') - \log \pi(\cdot|s')\| + \gamma L_p \|p' - p\| \right] \sqrt{\sum_{s,a} d_{\pi,p}(s)^2}$$

$$+ \frac{\gamma(1 + \lambda \log |\mathcal{A}|)}{1 - \gamma} \sqrt{\sum_{s,a} d_{\pi,p}(s)^2 \|p'(\cdot|s,a) - p(\cdot|s,a)\|_1^2}$$

$$\leq \frac{\sqrt{|\mathcal{A}|}(1 + \lambda \log |\mathcal{A}|)}{1 - \gamma} \sum_s |d_{\pi',p'}(s) - d_{\pi,p}(s)| + \frac{\sqrt{|\mathcal{A}|}\|r' - r\|_\infty}{1 - \gamma} + \lambda \sqrt{\sum_s d_{\pi,p}(s) \|\log \pi'(\cdot|s) - \log \pi(\cdot|s)\|^2}$$

$$+ \left[ \gamma L_\pi \max_{s'} \|\log \pi'(\cdot|s') - \log \pi(\cdot|s')\| + \gamma L_p \|p' - p\| \right] \sqrt{|\mathcal{A}|} + \frac{\gamma(1 + \lambda \log |\mathcal{A}|)}{1 - \gamma} \sqrt{|\mathcal{S}| \sum_{s,a} \|p'(\cdot|s,a) - p(\cdot|s,a)\|^2}$$

$$\overset{(a)}{\leq} \frac{\gamma \sqrt{|\mathcal{A}|}(1 + \lambda \log |\mathcal{A}|)}{(1 - \gamma)^2} \left[ \max_s \|\pi'(\cdot|s) - \pi(\cdot|s)\|_1 + \max_{s,a} \|p'(\cdot|s,a) - p(\cdot|s,a)\|_1 \right] + \frac{\sqrt{|\mathcal{A}|}\|r' - r\|_\infty}{1 - \gamma}$$

$$+ \lambda \max_{s'} \|\log \pi'(\cdot|s') - \log \pi(\cdot|s')\| + \left[ \gamma L_\pi \max_{s'} \|\log \pi'(\cdot|s') - \log \pi(\cdot|s')\| + \gamma L_p \|p' - p\| \right] \sqrt{|\mathcal{A}|}$$

$$+ \frac{\gamma \sqrt{|\mathcal{S}|}(1 + \lambda \log |\mathcal{A}|)}{1 - \gamma} \|p' - p\|$$

$$\overset{(b)}{\leq} \left[ \frac{|\mathcal{A}|(\gamma + 2\lambda \log |\mathcal{A}|)}{(1 - \gamma)^2} + \gamma L_\pi \right] \max_{s'} \|\log \pi'(\cdot|s') - \log \pi(\cdot|s')\| + \gamma \sqrt{|\mathcal{A}|} \left[ \frac{2\sqrt{|\mathcal{S}|}(1 + \lambda \log |\mathcal{A}|)}{(1 - \gamma)^2} + L_p \right] \|p' - p\|$$

$$+ \frac{\sqrt{|\mathcal{A}|}\|r' - r\|_\infty}{1 - \gamma},$$

where (a) uses Lemma 3, (b) uses $\|\pi'(\cdot|s) - \pi(\cdot|s)\|_1 \leq \|\log \pi'(\cdot|s) - \log \pi(\cdot|s)\|_1$, $\|p'(\cdot|s,a) - p(\cdot|s,a)\|_1 \leq \sqrt{|\mathcal{S}|}\|p'(\cdot|s,a) - p(\cdot|s,a)\| \leq \sqrt{|\mathcal{S}|}\|p' - p\|$, $\frac{\gamma \sqrt{|\mathcal{S}|}(1 + \lambda \log |\mathcal{A}|)}{1 - \gamma} \leq \frac{\sqrt{|\mathcal{S}||\mathcal{A}|}(1 + \lambda \log |\mathcal{A}|)}{(1 - \gamma)^2}$ and $\lambda \leq \frac{\lambda |\mathcal{A}| \log |\mathcal{A}|}{(1 - \gamma)^2}$. $\qquad \square$

### A.4. Zeroth-order Gradient Estimation Error

We import Theorem 1.6.2 of (Tropp et al., 2015) as follows.

**Lemma 7** (Matrix Bernstein Inequality). *Suppose complex-valued matrices $S_1, \ldots, S_N \in \mathbb{C}^{d_1 \times d_2}$ are independently distributed with $\mathbb{E}S_k = 0$ and $\|S_k\| \leq C$ for each $k = 1, \ldots, N$. Denote the sum $Z_N = \sum_{k=1}^N S_k$ its variance statistic as follows*

$$v(Z_N) = \max \left[ \left\| \sum_{k=1}^N \mathbb{E}(S_k S_k^*) \right\|, \left\| \sum_{k=1}^N \mathbb{E}(S_k^* S_k) \right\| \right], \tag{61}$$

*where $S_k^*$ denotes the conjugate transpose of $S_k$. Then for any $\epsilon \geq 0$, we have*

$$\mathbb{P}\{\|Z_N\| \geq \epsilon\} \leq (d_1 + d_2) \exp \left[ \frac{-\epsilon^2/2}{v(Z_N) + C\epsilon/3} \right]. \tag{62}$$

Applying the above lemma to vectors, we obtain the following vector Bernstein inequality.

**Lemma 8** (Vector Bernstein Inequality). *Suppose independently distributed vectors $x_1, \ldots, x_N \in \mathbb{C}^d$ satisfies $\|x_k\| \leq c$ for each $k = 1, \ldots, N$. Then for any $\eta \in (0, 1)$, with probability at least $1 - \eta$, we have*

$$\left\| \frac{1}{N} \sum_{k=1}^N (x_k - \mathbb{E}x_k) \right\| < \frac{4c}{3N} \log \left( \frac{d+1}{\eta} \right) + 2c \sqrt{\frac{2}{N} \log \left( \frac{d+1}{\eta} \right)}. \tag{63}$$

*Proof.* Note that $S_k = x_k - \mathbb{E}x_k$ satisfies the conditions of Lemma 7 with $d_1 = d$, $d_2 = 1$ and $C$ replaced by $2c$. In addition, $v(Z_N)$ defined by Eq. (61) satisfies $v(Z_N) \le 4Nc^2$ since

$$\max[\|S_k S_k^*\|, \|S_k^* S_k\|^2] \le \|S_k^*\|^2 \|S_k\|^2 \le 4c^2.$$

For any $\eta \in (0, 1)$, let

$$\epsilon = \frac{4c}{3} \log\left(\frac{d+1}{\eta}\right) + c\sqrt{2N \log\left(\frac{d+1}{\eta}\right)}.$$

Therefore, Lemma 7 implies that

$$\mathbb{P}\left\{\frac{1}{N}\Big\|\sum_{k=1}^{N}(x_k - \mathbb{E}x_k)\Big\| \ge \frac{\epsilon}{N}\right\} \le (d+1)\exp\left[\frac{-\epsilon^2/2}{4Nc^2 + 2c\epsilon/3}\right] \le \eta,$$

which implies that with probability at least $1 - \eta$, we have

$$\frac{1}{N}\Big\|\sum_{k=1}^{N}(x_k - \mathbb{E}x_k)\Big\| < \frac{\epsilon}{N} = \frac{4c}{3N}\log\left(\frac{d+1}{\eta}\right) + 2c\sqrt{\frac{2}{N}\log\left(\frac{d+1}{\eta}\right)}.$$

$\square$

For any function $f : \mathbb{R}^d \to \mathbb{R}$, obtain the following zeroth-order stochastic estimator of the gradient $\nabla f$.

$$g_\delta(x) = \frac{d}{2N\delta}\sum_{i=1}^{N}[f(x + \delta u_i) - f(x - \delta u_i)]u_i \approx \nabla f(x) \tag{64}$$

where $\delta > 0$ and $\{u_i\}_{i=1}^{N}$ are i.i.d. samples of the uniform distribution on the sphere $\mathbb{S}_d = \{u \in \mathbb{R}^d : \|u\| = 1\}$.

**Lemma 9.** *Suppose $f : \mathbb{R}^d \to \mathbb{R}$ is an $L_f$-Lipschitz continuous and $\ell_f$-smooth function. Then for any $\eta \in (0, 1)$, with probability at least $1 - \eta$, the gradient estimator $g_\delta$ defined by Eq. (64) has the following error bound.*

$$\|g_\delta(x) - \nabla f(x)\| \le \frac{4L_f d}{3N}\log\left(\frac{d+1}{\eta}\right) + 2L_f d\sqrt{\frac{2}{N}\log\left(\frac{d+1}{\eta}\right)} + \delta\ell_f. \tag{65}$$

*Proof.* Note that $g_{\delta,i}(x) \overset{\text{def}}{=} \frac{d}{2\delta}[f(x + \delta u_i) - f(x - \delta u_i)]u_i$ has the following norm bound

$$\|g_{\delta,i}(x)\| \le \frac{d}{2\delta}\big|f(x + \delta u_i) - f(x - \delta u_i)\big| \cdot \|u_i\| \le \frac{d}{2\delta} \cdot L_f \|2\delta u_i\| = L_f d. \tag{66}$$

Define the following smoothed approximation of $f$ as follows.

$$f_\delta(x) \overset{\text{def}}{=} \mathbb{E}_{v \sim \text{Unif}(\mathbb{B}_d)}[f(x + \delta v)], \tag{67}$$

where $\text{Unif}(\mathbb{B}_d)$ denotes the uniform distribution on the ball $\mathbb{B}_d \overset{\text{def}}{=} \{u \in \mathbb{R}^d : \|u\| \le 1\}$. Then based on Lemma 1 of (Flaxman et al., 2005), we have

$$\mathbb{E}[g_{\delta,i}(x)] = \nabla f_\delta(x) = \mathbb{E}_{v \sim \text{Unif}(\mathbb{B}_d)}[\nabla f(x + \delta v)]. \tag{68}$$

Therefore, applying Lemma 8 to $g_{\delta,i}(x)$, the following bound holds with probability at least $1 - \eta$.

$$\frac{1}{N}\Big\|\sum_{i=1}^{N}[g_{\delta,i}(x) - \nabla f_\delta(x)]\Big\| < \frac{4L_f d}{3N}\log\left(\frac{d+1}{\eta}\right) + 2L_f d\sqrt{\frac{2}{N}\log\left(\frac{d+1}{\eta}\right)}. \tag{69}$$

Note that

$$\|\nabla f_\delta(x) - \nabla f(x)\| = \left\|\mathbb{E}_{v \sim \text{Unif}(\mathbb{B}_d)}[\nabla f(x + \delta v) - \nabla f(x)]\right\| \le \delta \ell_f. \tag{70}$$

As a result, we can prove the conclusion as follows by using Eqs. (69) and (70) above.

$$
\begin{aligned}
\|g_\delta(x) - \nabla f(x)\| =& \left\|\left[\frac{1}{N}\sum_{i=1}^{N} g_{\delta,i}(x)\right] - \nabla f(x)\right\| \\
\le& \left\|\left[\frac{1}{N}\sum_{i=1}^{N} g_{\delta,i}(x)\right] - \nabla f_\delta(x)\right\| + \|\nabla f_\delta(x) - \nabla f(x)\| \\
<& \frac{4L_f d}{3N}\log\left(\frac{d+1}{\eta}\right) + 2L_f d\sqrt{\frac{2}{N}\log\left(\frac{d+1}{\eta}\right)} + \delta\ell_f.
\end{aligned}
$$

$\square$

### A.5. Orthogonal Transformation

**Lemma 10.** *There exists an orthogonal transformation $\mathcal{T}$ from the space $\mathbb{R}^{d-1}$ to $\mathcal{Z}_d = \{z = [z_1, \ldots, z_d] \in \mathbb{R}^d : \sum_i z_i = 0\}$, that is, $\mathcal{T}$ is invertible and satisfies the following properties for any $x, y \in \mathcal{Z}_d$ and $\alpha, \beta \in \mathbb{R}$.*

$$\mathcal{T}(\alpha x + \beta y) = \alpha\mathcal{T}(x) + \beta\mathcal{T}(y), \tag{71}$$
$$\langle\mathcal{T}(x), \mathcal{T}(y)\rangle = \langle x, y\rangle. \tag{72}$$

*Proof.* It can be verified that $\mathbb{R}^d$ admits the following orthonormal basis with $\langle e_i, e_j\rangle = 0$ for any $i \ne j$ and $\|e_i\| = 1$.

$$e_k = \frac{1}{\sqrt{k(k+1)}}[\underbrace{1, 1, \ldots, 1}_{k\ 1's}, -k, \underbrace{0, 0, \ldots, 0}_{(d-k-1)\ 0's}] \in \mathbb{R}^d; k = 1, 2, \ldots, d-1.$$

$$e_d = \frac{1}{\sqrt{d}}[\underbrace{1, 1, \ldots, 1}_{d\ 1's}] \in \mathbb{R}^d.$$

Define the transformation $\mathcal{T}$ at $x = [x_1, x_2, \ldots, x_{d-1}] \in \mathbb{R}^{d-1}$ as follows.

$$\mathcal{T}(x) = \sum_{i=1}^{d-1} x_i e_i. \tag{73}$$

Since $\mathcal{Z}_d$ is a linear subspace of $\mathbb{R}^d$ orthogonal to $e_d$, $\mathcal{Z}_d$ admits the orthonormal basis $\{e_i\}_{i=1}^{d-1}$. Hence, $\mathcal{T}(x) \in \mathcal{Z}_d$. Conversely, for any $y \in \mathcal{Z}_d$, there exists unique $x \in \mathbb{R}^{d-1}$ such that $y = \sum_{i=1}^{d-1} x_i e_i$. Hence, $\mathcal{T} : \mathbb{R}^{d-1} \to \mathcal{Z}^d$ is invertible.

For any $x = [x_1, \ldots, x_{d-1}], y = [y_1, \ldots, y_{d-1}] \in \mathbb{R}^{d-1}$ and $\alpha, \beta \in \mathbb{R}$, we can prove Eqs. (71) and (72) respectively as follows.

$$
\begin{aligned}
\mathcal{T}(\alpha x + \beta y) =& \sum_{i=1}^{d-1}(\alpha x_i + \beta y_i)e_i \\
=& \alpha\sum_{i=1}^{d-1} x_i e_i + \beta\sum_{i=1}^{d-1} y_i e_i \\
=& \alpha\mathcal{T}(x) + \beta\mathcal{T}(y).
\end{aligned}
$$

$$\langle\mathcal{T}(x), \mathcal{T}(y)\rangle = \left\langle\sum_{i=1}^{d-1} x_i e_i, \sum_{j=1}^{d-1} y_j e_j\right\rangle$$

$$= \sum_{i=1}^{d-1} \sum_{j=1}^{d-1} x_i y_j \langle e_i, e_j \rangle$$

$$= \sum_{i=1}^{d-1} x_i y_i = \langle x, y \rangle.$$

$\square$

### A.6. Basic Inequalities

**Lemma 11.** *For any $\epsilon \in (0, 0.5]$ and $x \geq 4\epsilon^{-1} \log(\epsilon^{-1})$, the following inequality holds.*

$$0 < \frac{\log x}{x} \leq \epsilon \tag{74}$$

*Specifically, any $x \geq 3$ satisfies $\frac{\log x}{x} \leq \frac{1}{2}$.*

*Proof.* As $\epsilon^{-1} \geq 2$, we have $x \geq 4\epsilon^{-1} \log(\epsilon^{-1}) \geq (4)(2) \log(2) > 5.54$, so $\log x > \log 5.54 > 1.71$, which proves the first $<$ of Eq. (74).

Note that the function $f(x) = \frac{\log x}{x}$ has the following derivative

$$f'(x) = \frac{1 - \log x}{x^2} < 0,$$

where $<$ uses $\log x > 1.71$. Hence, $f$ is monotonic decreasing in $x \geq 4\epsilon^{-1} \log(\epsilon^{-1}) > 5.54$, Therefore, we prove the second $\leq$ of Eq. (74) as follows.

$$\frac{\log x}{x\epsilon} \leq \frac{\log[4\epsilon^{-1} \log(\epsilon^{-1})]}{\epsilon[4\epsilon^{-1} \log(\epsilon^{-1})]} = \frac{\log 4 + \log(\epsilon^{-1}) + \log[\log(\epsilon^{-1})]}{4 \log(\epsilon^{-1})} \overset{(a)}{\leq} \frac{\log 4}{4 \log(2)} + \frac{\log(\epsilon^{-1}) + \log(\epsilon^{-1})}{4 \log(\epsilon^{-1})} = 1, \tag{75}$$

where (a) uses $\epsilon^{-1} \geq 2$ and $\log u \leq u$ for $u = \log(\epsilon^{-1})$.

When $x \geq 3$, $f'(x) = \frac{1 - \log x}{x^2} < 0$, so $f(x) \leq f(3) = \frac{\log 3}{3} < \frac{1}{2}$. $\square$

**Lemma 12.** *For any $\pi, \pi' \in \Pi$, we have $\|\pi' - \pi\| \leq \sqrt{2|\mathcal{S}|}$.*

*Proof.*

$$\|\pi' - \pi\|^2 = \sum_{s,a} |\pi'(a|s) - \pi(a|s)|^2 \leq \sum_{s,a} [\pi'^2(a|s) + \pi^2(a|s)] \leq \sum_{s,a} [\pi'(a|s) + \pi(a|s)] = 2|\mathcal{S}|.$$

$\square$

## B. Negative Entropy Regularizer as a Strongly Convex Function of Occupancy Measure

The negative entropy regularizer (7) can be rewritten as follows

$$\mathcal{H}_{\pi'}(\pi) = \mathbb{E}_{\pi, p_{\pi'}, \rho} \Big[ \sum_{t=0}^{\infty} \gamma^t \log \pi(a_t|s_t) \Big] = \frac{1}{1-\gamma} \sum_{s,a} d_{\pi, p_{\pi'}}(s, a) \log \frac{d_{\pi, p_{\pi'}}(s, a)}{d_{\pi, p_{\pi'}}(s)}, \tag{76}$$

where $d_{\pi, p_{\pi'}}(s) = \sum_{a'} d_{\pi, p_{\pi'}}(s, a')$. Hence, it suffices to prove that the following function of occupancy measure $d$ is strongly convex.

$$H(d) = \sum_{s,a} d(s, a) \log \frac{d(s, a)}{d(s)}, \tag{77}$$

where $d(s) = \sum_{a'} d(s, a')$. For any $\alpha \in [0, 1]$ and occupancy measures $d_1, d_0$, denote $d_\alpha = \alpha d_1 + (1 - \alpha)d_0$ and the corresponding policy as $\pi_\alpha(a|s) = \frac{d_\alpha(s,a)}{d_\alpha(s)}$. Then we have

$$
\begin{aligned}
&\alpha H(d_1) + (1 - \alpha)H(d_0) - H(d_\alpha) \\
&= \sum_{s,a} \Big[ \alpha d_1(s, a) \log \pi_1(a|s) + (1 - \alpha)d_0(s, a) \log \pi_0(a|s) - [\alpha d_1(s, a) + (1 - \alpha)d_0(s, a)] \log \pi_\alpha(a|s) \Big] \\
&= \sum_{s,a} \Big[ \alpha d_1(s, a) \log \frac{\pi_1(a|s)}{\pi_\alpha(a|s)} + (1 - \alpha)d_0(s, a) \log \frac{\pi_0(a|s)}{\pi_\alpha(a|s)} \Big] \\
&= \sum_{s,a} \Big[ \alpha d_1(s)\pi_1(a|s) \log \frac{\pi_1(a|s)}{\pi_\alpha(a|s)} + (1 - \alpha)d_0(s)\pi_0(a|s) \log \frac{\pi_0(a|s)}{\pi_\alpha(a|s)} \Big] \\
&= \sum_{s} \Big[ \alpha d_1(s)\mathrm{KL}[\pi_1(\cdot|s)\|\pi_\alpha(a|s)] + (1 - \alpha)d_0(s)\mathrm{KL}[\pi_0(\cdot|s)\|\pi_\alpha(a|s)] \Big] \\
&\overset{(a)}{\geq} \frac{1}{2} \sum_{s} \Big[ \alpha d_1(s)\|\pi_1(\cdot|s) - \pi_\alpha(\cdot|s)\|_1^2 + (1 - \alpha)d_0(s)\|\pi_0(\cdot|s) - \pi_\alpha(\cdot|s)\|_1^2 \Big] \\
&\overset{(b)}{\geq} \frac{D}{2} \sum_{s} \Big[ \alpha\|\pi_1(\cdot|s) - \pi_\alpha(\cdot|s)\|_1^2 + (1 - \alpha)\|\pi_0(\cdot|s) - \pi_\alpha(\cdot|s)\|_1^2 \Big] \\
&\geq \frac{D}{2} \Big[ \alpha \max_s \|\pi_1(\cdot|s) - \pi_\alpha(\cdot|s)\|_1^2 + (1 - \alpha) \max_s \|\pi_0(\cdot|s) - \pi_\alpha(\cdot|s)\|_1^2 \Big] \\
&\overset{(c)}{\geq} \frac{D(1 - \gamma)}{2} \Big[ \alpha\|d_1 - d_\alpha\|_1^2 + (1 - \alpha) \max_s \|d_0 - d_\alpha\|_1^2 \Big] \\
&= \frac{D(1 - \gamma)}{2} \Big[ \alpha(1 - \alpha)^2\|d_1 - d_0\|_1^2 + (1 - \alpha)\alpha^2\|d_1 - d_0\|_1^2 \Big] \\
&= \frac{\alpha(1 - \alpha)}{2} \cdot D(1 - \gamma)\|d_1 - d_0\|_1^2.
\end{aligned}
\tag{78}
$$

where (a) uses Pinsker's inequality, (b) uses Assumption 3, (c) uses Eq. (38) with $p' = p$. The inequality above implies that $H(d)$ is $D(1 - \gamma)$-strongly convex, so the negative entropy regularizer (76) can be seen as a $D$-strongly convex function of the occupancy measure $d_{\pi, p_{\pi'}}$.

## C. Existing Assumptions That Implies Assumption 3

The following assumptions have been used in the reinforcement learning literature. We will show that each of these assumptions implies Assumption 3.

**Assumption 4.** *(Bhandari and Russo, 2024)* $\rho(s) > 0$ for any $s \in \mathcal{S}$.

**Assumption 5.** *(Agarwal et al., 2021; Leonardos et al., 2022; Wang et al., 2023; Chen and Huang, 2024)* $D_\rho := \sup_{\pi \in \Pi, p \in \mathcal{P}} \|d_{\pi,p}/\rho\|_\infty < \infty$.

**Assumption 6.** *(Wei et al., 2021; Chen et al., 2022) There exists a constant $\mu_{\min} > 0$ and mixing time $t_{mix} \in \mathbb{N}$ such that under any policy $\pi \in \Pi$ and transition kernel $p \in \mathcal{P}$, the stationary state distribution $\mu_{\pi,p}(s)$ has uniform lower bound $\min_{s \in \mathcal{S}} \mu_{\pi,p}(s) \geq \mu_{\min}$, and*

$$
d_{\mathrm{TV}}\big[\mathbb{P}_{\pi,p,\rho}(s_{t_{mix}} = \cdot), \mu_{\pi,p}\big] \leq \frac{1}{4},
\tag{79}
$$

*where $\mathbb{P}_{\pi,p,\rho}(s_{t_{mix}} = \cdot)$ denotes the state distribution at time $t_{mix}$, under the policy $\pi$, transition kernel $p$ and initial state distribution $\rho$, and $d_{\mathrm{TV}}$ denotes the total variation distance between two probability distributions.*

**Proof of Assumption 4$\Rightarrow$Assumption 3:** For any policy $\pi \in \Pi$, transition kernel $p \in \mathcal{P}$ and state $s \in \mathcal{S}$, we have

$$
d_{\pi,p}(s) = \sum_a d_{\pi,p}(s, a)
$$

$$\overset{(a)}{=} \sum_a (1-\gamma) \sum_{t=0}^{\infty} \gamma^t \mathbb{P}_{\pi,p,\rho}\{s_t = s, a_t = a\}$$

$$= (1-\gamma) \sum_{t=0}^{\infty} \gamma^t \mathbb{P}_{\pi,p,\rho}\{s_t = s\}$$

$$\geq (1-\gamma)\mathbb{P}_{\pi,p,\rho}\{s_0 = s\}$$

$$= (1-\gamma)\rho(s)$$

$$\geq (1-\gamma)\min_{s\in\mathcal{S}}\rho(s).$$

As $\mathcal{S}$ is a finite state space, $\rho(s) > 0, \forall s \in \mathcal{S}$ implies that $\min_{s\in\mathcal{S}} \rho(s) > 0$. Hence, Assumption 3 holds with $D = (1-\gamma)\min_{s\in\mathcal{S}} \rho(s) > 0$.

**Proof of Assumption 5$\Rightarrow$Assumption 3:** If $\rho(s) = 0$ for a state $s$, then Assumption 5 implies that $d_{\pi,p}(s) = (1-\gamma)\sum_{t=0}^{\infty} \gamma^t \mathbb{P}_{\pi,p,\rho}\{s_t = s\} = 0$ for any $\pi \in \Pi$ and $p \in \mathcal{P}$, which means the state $s$ will never be visited. Therefore, we can exclude all such states $s$ from $\mathcal{S}$ such that Assumption 4 holds, which implies Assumption 3 as proved above.

**Proof of Assumption 6$\Rightarrow$Assumption 3:** Eq. (79) implies that for any $n \in \mathbb{N}_+$, we have

$$d_{\mathrm{TV}}\big[\mathbb{P}_{\pi,p,\rho}(s_{nt_{\mathrm{mix}}} = \cdot), \mu_{\pi,p}\big] = \frac{1}{2}\sum_s |\mathbb{P}_{\pi,p,\rho}\{s_{nt_{\mathrm{mix}}} = s\} - \mu_{\pi,p}(s)| \leq \frac{1}{4^n}.$$

Select $n = \lceil \log(\mu_{\min}^{-1})/\log 4 \rceil$. Then the bound above implies $|\mathbb{P}_{\pi,p,\rho}\{s_{nt_{\mathrm{mix}}} = s\} - \mu_{\pi,p}(s)| \leq \mu_{\min}/2$ for any state $s$, which along with $\mu_{\pi,p}(s) \geq \mu_{\min}$ implies that $\mathbb{P}_{\pi,p,\rho}\{s_{nt_{\mathrm{mix}}} = s\} \geq \mu_{\min}/2$. Therefore, we can prove Assumption 3 as follows.

$$d_{\pi,p}(s) = (1-\gamma)\sum_{t=0}^{\infty} \gamma^t \mathbb{P}_{\pi,p,\rho}\{s_t = s\} \geq (1-\gamma)\gamma^{nt_{\mathrm{mix}}}\mathbb{P}_{\pi,p,\rho}\{s_{nt_{\mathrm{mix}}} = s\} \geq \frac{\mu_{\min}}{2}\gamma^{nt_{\mathrm{mix}}}(1-\gamma).$$

## D. Proof of Theorem 1

Fix any $\pi_0, \pi_1 \in \Pi$. For any $\alpha \in [0,1]$, denote $d_\alpha = \alpha d_{\pi_1,p_{\pi_1}} + (1-\alpha)d_{\pi_0,p_{\pi_0}}$, $\pi_\alpha(a|s) = \frac{d_\alpha(s,a)}{d_\alpha(s)}$ where $d_\alpha(s) = \sum_{a'} d_\alpha(s,a')$, and $p_\alpha = p_{\pi_\alpha}$. It can be easily verified that $d_0 = d_{\pi_0,p_0}$, $d_1 = d_{\pi_1,p_1}$ and $d_\alpha = \alpha d_0 + (1-\alpha)d_1$. Then we can obtain the following derivatives and their bounds about $\pi_\alpha, d_\alpha$ in Eqs. (80)-(86).

$$\frac{d}{d\alpha}\pi_\alpha(a|s) = \frac{d_\alpha(s)[d_1(s,a) - d_0(s,a)] - d_\alpha(s,a)[d_1(s) - d_0(s)]}{d_\alpha^2(s)}$$

$$= \frac{[\alpha d_1(s) + (1-\alpha)d_0(s)][d_1(s,a) - d_0(s,a)] - [\alpha d_1(s,a) + (1-\alpha)d_0(s,a)][d_1(s) - d_0(s)]}{d_\alpha^2(s)}$$

$$= \frac{d_0(s)d_1(s,a) - d_0(s,a)d_1(s)}{d_\alpha^2(s)}$$

$$= \frac{d_0(s)d_1(s)[\pi_1(a|s) - \pi_0(a|s)]}{d_\alpha^2(s)}. \tag{80}$$

Hence,

$$\left\|\frac{d\pi_\alpha}{d\alpha}\right\|^2 = \sum_{s,a} \left|\frac{d_0(s)d_1(s)[\pi_1(a|s) - \pi_0(a|s)]}{d_\alpha^2(s)}\right|^2$$

$$\overset{(a)}{\leq} \sum_{s,a}\left[\frac{\max[d_0(s),d_1(s)]\min[d_0(s),d_1(s)]}{\min^2[d_0(s),d_1(s)]}\right]^2 [\pi_1(a|s) - \pi_0(a|s)]^2$$

$$\overset{(b)}{\leq} D^{-2}\sum_{s,a}[\pi_1(a|s) - \pi_0(a|s)]^2 \leq D^{-2}\|\pi_1 - \pi_0\|^2, \tag{81}$$

where (a) uses $d_\alpha(s) = \alpha d_1(s) + (1-\alpha)d_0(s) \geq \min[d_0(s), d_1(s)]$ and (b) uses Assumption 3. Then by taking derivative of Eq. (80), we have

$$\frac{d^2}{d\alpha^2}\pi_\alpha(a|s) = -\frac{2d_0(s)d_1(s)[\pi_1(a|s) - \pi_0(a|s)][d_1(s) - d_0(s)]}{d_\alpha^3(s)}. \tag{82}$$

Hence,

$$\left\|\frac{d^2\pi_\alpha}{d\alpha^2}\right\|^2 = \sum_{s,a}\left|\frac{2d_0(s)d_1(s)[\pi_1(a|s) - \pi_0(a|s)][d_1(s) - d_0(s)]}{[\alpha d_1(s) + (1-\alpha)d_0(s)]^3}\right|^2$$

$$\overset{(a)}{\leq} \sum_{s,a}\left[\frac{2\max[d_0(s), d_1(s)]\min[d_0(s), d_1(s)]\big|d_1(s) - d_0(s)\big|}{D^2\min[d_0(s), d_1(s)]}\right]^2[\pi_1(a|s) - \pi_0(a|s)]^2$$

$$\leq (2D^{-2})^2\max_s\left[|d_1(s) - d_0(s)|^2\right]\sum_{s,a}[\pi_1(a|s) - \pi_0(a|s)]^2$$

$$\leq (2D^{-2})^2\|\pi_1 - \pi_0\|^2\left[\sum_s|d_1(s) - d_0(s)|\right]^2$$

$$\overset{(b)}{\leq} (2D^{-2})^2\|\pi_1 - \pi_0\|^2\left[\frac{\gamma\sqrt{|\mathcal{A}|}}{1-\gamma}\|\pi_1 - \pi_0\| + \frac{\gamma\sqrt{|\mathcal{S}|}}{1-\gamma}\|p_{\pi_1} - p_{\pi_0}\|\right]^2$$

$$\overset{(c)}{\leq} (2D^{-2})^2\|\pi_1 - \pi_0\|^2\left[\frac{\gamma\sqrt{|\mathcal{A}|}}{1-\gamma}\|\pi_1 - \pi_0\| + \frac{\gamma\epsilon_p\sqrt{|\mathcal{S}|}}{1-\gamma}\|\pi_1 - \pi_0\|\right]^2$$

$$\leq (2D^{-2})^2\|\pi_1 - \pi_0\|^4\left[\frac{\gamma(\epsilon_p\sqrt{|\mathcal{S}|} + \sqrt{|\mathcal{A}|})}{1-\gamma}\right]^2, \tag{83}$$

where (a) uses $d_\alpha(s) = \alpha d_1(s) + (1-\alpha)d_0(s) \geq \min[d_0(s), d_1(s)] \geq D$, (b) uses Lemma 3, and (c) uses Assumption 1.

$$d_0(s)d_1(s)\left|\frac{d}{d\alpha}\left[\frac{d_\alpha(s,a)}{d_\alpha^2(s)}\right]\right|$$

$$= \left|\frac{d_0(s)d_1(s)}{d_\alpha^2(s)}[d_1(s,a) - d_0(s,a)] - \frac{2d_0(s)d_1(s)d_\alpha(s,a)}{d_\alpha^3(s)}[d_1(s) - d_0(s)]\right|$$

$$\leq \frac{d_0(s)d_1(s)}{d_\alpha^2(s)}\left[|d_1(s,a) - d_0(s,a)| + \frac{2d_\alpha(s,a)}{d_\alpha(s)}|d_1(s) - d_0(s)|\right]$$

$$\leq \frac{\max[d_0(s), d_1(s)]\min[d_0(s), d_1(s)]}{\min^2[d_0(s), d_1(s)]}\left[|d_1(s,a) - d_0(s,a)| + 2\pi_\alpha(a|s)|d_1(s) - d_0(s)|\right]$$

$$\leq D^{-1}\left[|d_1(s,a) - d_0(s,a)| + 2\pi_\alpha(a|s)|d_1(s) - d_0(s)|\right]. \tag{84}$$

$$\frac{d}{d\alpha}[d_\alpha(s,a)p_\alpha(s'|s,a)]$$

$$= p_\alpha(s'|s,a)[d_1(s,a) - d_0(s,a)] + d_\alpha(s,a)\cdot\frac{d}{d\alpha}\pi_\alpha(a|s)\cdot\nabla_\pi p_{\pi_\alpha}(s'|s,a)$$

$$= p_\alpha(s'|s,a)[d_1(s,a) - d_0(s,a)] + \frac{d_\alpha(s,a)d_0(s)d_1(s)[\pi_1(a|s) - \pi_0(a|s)]}{d_\alpha^2(s)}\cdot\nabla_\pi p_{\pi_\alpha}(s'|s,a) \tag{85}$$

Then for any $\alpha, \alpha' \in [0,1]$, we have

$$\left|\frac{d}{d\alpha}[d_{\alpha'}(s,a)p_{\alpha'}(s'|s,a)] - \frac{d}{d\alpha}[d_\alpha(s,a)p_\alpha(s'|s,a)]\right|$$

$$\overset{(a)}{\leq} |p_{\alpha'}(s'|s,a) - p_\alpha(s'|s,a)|\cdot|d_1(s,a) - d_0(s,a)| + d_0(s)d_1(s)|\pi_1(a|s) - \pi_0(a|s)|\cdot$$

$$\left[\left|\frac{d_{\alpha'}(s,a)}{d_{\alpha'}^2(s)}\right|\|\nabla_\pi p_{\pi_{\alpha'}}(s'|s,a) - \nabla_\pi p_{\pi_\alpha}(s'|s,a)\| + \left|\frac{d_{\alpha'}(s,a)}{d_{\alpha'}^2(s)} - \frac{d_\alpha(s,a)}{d_\alpha^2(s)}\right|\|\nabla_\pi p_{\pi_\alpha}(s'|s,a)\|\right]$$

$$\overset{(b)}{\leq} \epsilon_p \|\pi_{\alpha'} - \pi_\alpha\| |d_1(s,a) - d_0(s,a)|$$

$$+ \pi_{\alpha'}(a|s) |\pi_1(a|s) - \pi_0(a|s)| \cdot \frac{\max[d_0(s), d_1(s)] \min[d_0(s), d_1(s)]}{\min[d_0(s), d_1(s)]} \cdot S_p \|\pi_{\alpha'} - \pi_\alpha\|$$

$$+ D^{-1} \epsilon_p |\pi_1(a|s) - \pi_0(a|s)| \cdot \left[ |d_1(s,a) - d_0(s,a)| + 2\pi_\alpha(a|s) |d_1(s) - d_0(s)| \right] \cdot |\alpha' - \alpha|$$

$$\overset{(c)}{\leq} \epsilon_p D^{-1} \|\pi_1 - \pi_0\| \cdot |\alpha' - \alpha| \cdot |d_1(s,a) - d_0(s,a)|$$

$$+ S_p \pi_{\alpha'}(a|s) \cdot |\pi_1(a|s) - \pi_0(a|s)| \cdot [d_0(s) + d_1(s)] \cdot D^{-1} \|\pi_1 - \pi_0\| \cdot |\alpha' - \alpha|$$

$$+ D^{-1} \epsilon_p |\pi_1(a|s) - \pi_0(a|s)| \cdot \left[ |d_1(s,a) - d_0(s,a)| + 2\pi_\alpha(a|s)|d_1(s) - d_0(s)| \right] \cdot |\alpha' - \alpha|$$

$$\overset{(d)}{\leq} \ell_{dp}(s,a)|\alpha' - \alpha|, \tag{86}$$

where (a) uses Eq. (85), (b) uses Assumptions 1-2, $d_{\alpha'}(s,a) = d_{\alpha'}(s)\pi_{\alpha'}(a|s)$, $d_{\alpha'}(s) = \alpha' d_1(s) + (1 - \alpha')d_0(s) \geq \min[d_0(s), d_1(s)]$ and Eq. (84), (c) uses Assumption 3 as well as Eq. (81), (d) defines $\ell_{dp}(s,a)$ as the following Eq. (87) and uses $\pi_\alpha(a|s) = \frac{\alpha d_1(s)\pi_1(a|s) + (1-\alpha)d_0(s)\pi_0(a|s)}{\alpha d_1(s) + (1-\alpha)d_0(s)} \leq \pi_0(a|s) + \pi_1(a|s)$.

$$\ell_{dp}(s,a) = 2D^{-1}\epsilon_p \|\pi_1 - \pi_0\| |d_1(s,a) - d_0(s,a)| + 2D^{-1}\epsilon_p [\pi_1(a|s) + \pi_0(a|s)] \cdot |\pi_1(a|s) - \pi_0(a|s)| \cdot |d_1(s) - d_0(s)|$$

$$+ D^{-1} S_p [\pi_1(a|s) + \pi_0(a|s)] \cdot |\pi_1(a|s) - \pi_0(a|s)| \cdot \|\pi_1 - \pi_0\| \cdot [d_0(s) + d_1(s)]. \tag{87}$$

Denote $e_\alpha(s) = d_{\pi_\alpha, p_\alpha}(s) - d_\alpha(s)$ as the error term due to the policy-dependent transition kernel $p_\alpha = p_{\pi_\alpha}$[1]. Note that the occupancy measure (2) satisfies that the Bellman equation (3) repeated as follows.

$$d_{\pi,p}(s') = (1 - \gamma)\rho(s') + \gamma \sum_{s,a} d_{\pi,p}(s)\pi(a|s)p(s'|s,a), \quad s' \in \mathcal{S}. \tag{88}$$

Therefore, the error term $e_\alpha(s)$ satisfies the following recursion.

$$e_\alpha(s')$$
$$= d_{\pi_\alpha, p_\alpha}(s') - \alpha d_1(s') - (1 - \alpha)d_0(s')$$
$$= \gamma \sum_{s,a} [d_{\pi_\alpha, p_\alpha}(s)\pi_\alpha(a|s)p_\alpha(s'|s,a) - \alpha d_{\pi_1, p_1}(s)\pi_1(a|s)p_1(s'|s,a) - (1 - \alpha)d_{\pi_0, p_0}(s)\pi_0(a|s)p_0(s'|s,a)]$$
$$= \gamma \sum_{s,a} [e_\alpha(s)\pi_\alpha(a|s)p_\alpha(s'|s,a) + d_\alpha(s,a)p_\alpha(s'|s,a) - \alpha d_1(s,a)p_1(s'|s,a) - (1 - \alpha)d_0(s,a)p_0(s'|s,a)]. \tag{89}$$

The above inequality implies that

$$\sum_{s'} |e_\alpha(s')|$$

$$\leq \gamma \sum_{s,a,s'} \left[ |e_\alpha(s)|\pi_\alpha(a|s)p_\alpha(s'|s,a) + |d_\alpha(s,a)p_\alpha(s'|s,a) - \alpha d_1(s,a)p_1(s'|s,a) - (1 - \alpha)d_0(s,a)p_0(s'|s,a)| \right]$$

$$\overset{(a)}{\leq} \gamma \sum_s |e_\alpha(s)| + \frac{\gamma\alpha(1 - \alpha)}{2} \sum_{s,a,s'} \ell_{dp}(s,a)$$

$$\overset{(b)}{\leq} \gamma \sum_s |e_\alpha(s)| + \frac{\gamma|\mathcal{S}|\alpha(1 - \alpha)}{2} \left[ 2D^{-1}\epsilon_p \|\pi_1 - \pi_0\| \sum_{s,a} |d_1(s,a) - d_0(s,a)| + 4D^{-1}\epsilon_p \|\pi_1 - \pi_0\|_\infty \sum_s |d_1(s) - d_0(s)| \right.$$

$$\left. + 4D^{-1}S_p \|\pi_1 - \pi_0\|_\infty \cdot \|\pi_1 - \pi_0\| \right]$$

$$\overset{(c)}{\leq} \gamma \sum_s |e_\alpha(s)| + \frac{\gamma|\mathcal{S}|\alpha(1 - \alpha)}{2} \left[ 6D^{-1}\epsilon_p \|\pi_1 - \pi_0\| \cdot \frac{1}{1 - \gamma} \left( \sqrt{|\mathcal{A}|}\|\pi_1 - \pi_0\| + \gamma\sqrt{|\mathcal{S}|}\|p_{\pi_1} - p_{\pi_0}\| \right) + 4D^{-1}S_p \|\pi_1 - \pi_0\|^2 \right]$$

$$\overset{(d)}{\leq} \gamma \sum_s |e_\alpha(s)| + 3D^{-1}\gamma|\mathcal{S}|\alpha(1 - \alpha)\|\pi_1 - \pi_0\|^2 \left[ \frac{\epsilon_p}{1 - \gamma}(\sqrt{|\mathcal{A}|} + \gamma\epsilon_p\sqrt{|\mathcal{S}|}) + S_p \right],$$

---

[1] If $p_{\pi_\alpha} \equiv p$ does not depend on the policy $\pi_\alpha$, it can be easily verified that $e_\alpha(s) = 0$ for all $s \in \mathcal{S}$.

where (a) uses Eq. (86) which implies that $d_\alpha(s,a)p_\alpha(s'|s,a)$ is a Lipschitz smooth function with Lipschitz constant $\ell_{dp}(s,a)$ defined by Eq. (87), (b) uses Eq. (87), (c) uses $\|\pi_1 - \pi_0\|_\infty \le \|\pi_1 - \pi_0\|$ and Lemma 3, and (d) uses Assumption 1. Rearranging the above inequality, we get

$$\sum_s |e_\alpha(s)| \le \frac{3\gamma|\mathcal{S}|\alpha(1-\alpha)}{D(1-\gamma)^2}\|\pi_1 - \pi_0\|^2 \big[\epsilon_p\big(\sqrt{|\mathcal{A}|} + \gamma\epsilon_p\sqrt{|\mathcal{S}|}\big) + S_p(1-\gamma)\big]. \tag{90}$$

Therefore, for any reward function $r$, we have

$$J_\lambda(\pi_\alpha, \pi_\alpha, p_\alpha, r) - \alpha J_\lambda(\pi_1, \pi_1, p_1, r) - (1-\alpha)J_\lambda(\pi_0, \pi_0, p_0, r)$$

$$\stackrel{(a)}{=} \frac{1}{1-\gamma}\sum_{s,a}\Big[d_{\pi_\alpha, p_\alpha}(s,a)[r(s,a) - \lambda\log\pi_\alpha(a|s)] - \alpha d_1(s,a)[r(s,a) - \lambda\log\pi_1(a|s)]$$

$$- (1-\alpha)d_0(s,a)[r(s,a) - \lambda\log\pi_0(a|s)]\Big]$$

$$= \frac{1}{1-\gamma}\sum_{s,a}\Big[[d_{\pi_\alpha, p_\alpha}(s,a) - d_\alpha(s,a)][r(s,a) - \lambda\log\pi_\alpha(a|s)]$$

$$+ d_\alpha(s,a)[r(s,a) - \lambda\log\pi_\alpha(a|s)] - \alpha d_1(s,a)[r(s,a) - \lambda\log\pi_1(a|s)] - (1-\alpha)d_0(s,a)[r(s,a) - \lambda\log\pi_0(a|s)]\Big]$$

$$\stackrel{(b)}{=} \frac{1}{1-\gamma}\sum_{s,a}[d_{\pi_\alpha, p_\alpha}(s) - d_\alpha(s)]\pi_\alpha(a|s)[r(s,a) - \lambda\log\pi_\alpha(a|s)]$$

$$+ \frac{\lambda}{1-\gamma}\sum_{s,a}\Big[\alpha d_1(s,a)\log\frac{\pi_1(a|s)}{\pi_\alpha(a|s)} + (1-\alpha)d_0(s,a)\log\frac{\pi_0(a|s)}{\pi_\alpha(a|s)}\Big]$$

$$\stackrel{(c)}{\ge} -\frac{1+\lambda\log|\mathcal{A}|}{1-\gamma}\sum_s|e_\alpha(s)| + \frac{\lambda}{1-\gamma}\sum_s\Big[\alpha d_1(s)\sum_a\Big(\pi_1(a|s)\log\frac{\pi_1(a|s)}{\pi_\alpha(a|s)}\Big) + (1-\alpha)d_0(s)\sum_a\Big(\pi_0(a|s)\log\frac{\pi_0(a|s)}{\pi_\alpha(a|s)}\Big)\Big]$$

$$\stackrel{(d)}{\ge} -\frac{1+\lambda\log|\mathcal{A}|}{1-\gamma}\frac{3\gamma|\mathcal{S}|\alpha(1-\alpha)}{D(1-\gamma)^2}\|\pi_1 - \pi_0\|^2\big[\epsilon_p\big(\sqrt{|\mathcal{A}|} + \gamma\epsilon_p\sqrt{|\mathcal{S}|}\big) + S_p(1-\gamma)\big]$$

$$+ \frac{\lambda}{1-\gamma}\sum_s\Big[\alpha d_1(s)\mathrm{KL}[\pi_1(\cdot|s)\|\pi_\alpha(\cdot|s)] + (1-\alpha)d_0(s)\mathrm{KL}[\pi_0(\cdot|s)\|\pi_\alpha(\cdot|s)]\Big]$$

$$\stackrel{(e)}{\ge} -\frac{3\gamma|\mathcal{S}|\alpha(1-\alpha)(1+\lambda\log|\mathcal{A}|)}{D(1-\gamma)^3}\|\pi_1 - \pi_0\|^2\big[\epsilon_p\big(\sqrt{|\mathcal{A}|} + \gamma\epsilon_p\sqrt{|\mathcal{S}|}\big) + S_p(1-\gamma)\big]$$

$$+ \frac{\lambda}{2(1-\gamma)}\sum_s\Big[\alpha d_1(s)\|\pi_1(\cdot|s) - \pi_\alpha(\cdot|s)\|_1^2 + (1-\alpha)d_0(s)\|\pi_0(\cdot|s) - \pi_\alpha(\cdot|s)\|_1^2\Big]$$

$$\stackrel{(f)}{=} -\frac{3\gamma|\mathcal{S}|\alpha(1-\alpha)(1+\lambda\log|\mathcal{A}|)}{D(1-\gamma)^3}\|\pi_1 - \pi_0\|^2\big[\epsilon_p\big(\sqrt{|\mathcal{A}|} + \gamma\epsilon_p\sqrt{|\mathcal{S}|}\big) + S_p(1-\gamma)\big]$$

$$+ \frac{\lambda}{2(1-\gamma)}\sum_s\Big[\alpha d_1(s)\Big\|\frac{(1-\alpha)d_0(s)}{d_\alpha(s)}[\pi_1(\cdot|s) - \pi_0(\cdot|s)]\Big\|_1^2 + (1-\alpha)d_0(s)\Big\|\frac{\alpha d_1(s)}{d_\alpha(s)}[\pi_1(\cdot|s) - \pi_0(\cdot|s)]\Big\|_1^2\Big]$$

$$\stackrel{(g)}{=} \frac{\lambda\alpha(1-\alpha)}{2(1-\gamma)}\sum_s\frac{d_0(s)d_1(s)}{d_\alpha(s)}\|\pi_1(\cdot|s) - \pi_0(\cdot|s)\|_1^2$$

$$- \frac{3\gamma|\mathcal{S}|\alpha(1-\alpha)(1+\lambda\log|\mathcal{A}|)}{D(1-\gamma)^3}\|\pi_1 - \pi_0\|^2\big[\epsilon_p\big(\sqrt{|\mathcal{A}|} + \gamma\epsilon_p\sqrt{|\mathcal{S}|}\big) + S_p(1-\gamma)\big]$$

$$\stackrel{(h)}{\ge} \frac{D\lambda\alpha(1-\alpha)}{2(1-\gamma)}\|\pi_1 - \pi_0\|^2 - \frac{3\gamma|\mathcal{S}|\alpha(1-\alpha)(1+\lambda\log|\mathcal{A}|)}{D(1-\gamma)^3}\|\pi_1 - \pi_0\|^2\big[\epsilon_p\big(\sqrt{|\mathcal{A}|} + \gamma\epsilon_p\sqrt{|\mathcal{S}|}\big) + S_p(1-\gamma)\big]$$

$$\stackrel{(i)}{=} \frac{\mu_1\alpha(1-\alpha)}{2}\|\pi_1 - \pi_0\|^2, \tag{91}$$

where (a) uses Eq. (39), (b) uses $d_{\pi_\alpha, p_\alpha}(s,a) = d_{\pi_\alpha, p_\alpha}(s)\pi_\alpha(a|s)$, $d_\alpha(s,a) = d_\alpha(s)\pi_\alpha(a|s)$ and $d_\alpha = \alpha d_1 + (1-\alpha)d_0$, (c) uses $r(s,a) \in [0,1]$, $-\sum_a \pi_\alpha(a|s)\log\pi_\alpha(a|s) \in [0, \log|\mathcal{A}|]$ and $e_\alpha(s) = d_{\pi_\alpha, p_\alpha}(s) - d_\alpha(s)$, (d) uses Eq. (90), (e)

uses Pinsker's inequality, (f) uses $\pi_\alpha(a|s) = \frac{d_\alpha(s,a)}{d_\alpha(s)} = \frac{\alpha d_1(s)}{d_\alpha(s)}\pi_1(a|s) + \frac{(1-\alpha)d_0(s)}{d_\alpha(s)}\pi_0(a|s)$, (g) uses $d_\alpha(s) = \alpha d_1(s) + (1-\alpha)d_0(s)$, (h) uses Assumption 3 and $d_\alpha(s) \le \max[d_0(s), d_1(s)]$, and (i) defines the constant $\mu_1$ below.

$$\mu_1 \stackrel{\text{def}}{=} \frac{D\lambda}{1-\gamma} - \frac{6\gamma|\mathcal{S}|(1+\lambda\log|\mathcal{A}|)}{D(1-\gamma)^3}\big[\epsilon_p\big(\sqrt{|\mathcal{A}|} + \gamma\epsilon_p\sqrt{|\mathcal{S}|}\big) + S_p(1-\gamma)\big]. \tag{92}$$

Next, we begin to consider the policy-dependent reward $r_\alpha = r_{\pi_\alpha}$. Define the function $w(\alpha) = \alpha J_\lambda(\pi_1, \pi_1, p_1, r_\alpha) + (1-\alpha)J_\lambda(\pi_0, \pi_0, p_0, r_\alpha)$, which has the following derivative

$$\begin{aligned}
w'(\alpha) =& J_\lambda(\pi_1, \pi_1, p_1, r_\alpha) - J_\lambda(\pi_0, \pi_0, p_0, r_\alpha) \\
&+ [\alpha\nabla_r J_\lambda(\pi_1, \pi_1, p_1, r_\alpha) + (1-\alpha)\nabla_r J_\lambda(\pi_0, \pi_0, p_0, r_\alpha)](\nabla_\pi r_{\pi_\alpha})\frac{d\pi_\alpha}{d\alpha}
\end{aligned} \tag{93}$$

For any $0 \le \alpha \le \alpha' \le 1$, we prove the smoothness of $w(\alpha)$ as follows.

$$\begin{aligned}
&|w'(\alpha') - w'(\alpha)| \\
=& \Big| \int_\alpha^{\alpha'} \nabla_r[J_\lambda(\pi_1, \pi_1, p_1, r_{\tilde\alpha}) - J_\lambda(\pi_0, \pi_0, p_0, r_{\tilde\alpha})](\nabla_\pi r_{\pi_{\tilde\alpha}})\frac{d\pi_{\tilde\alpha}}{d\tilde\alpha}d\tilde\alpha \\
&+ [\alpha'\nabla_r J_\lambda(\pi_1, \pi_1, p_1, r_{\alpha'}) + (1-\alpha')\nabla_r J_\lambda(\pi_0, \pi_0, p_0, r_{\alpha'})](\nabla_\pi r_{\pi_{\alpha'}})\Big(\frac{d\pi_{\alpha'}}{d\alpha'} - \frac{d\pi_\alpha}{d\alpha}\Big) \\
&+ [\alpha'\nabla_r J_\lambda(\pi_1, \pi_1, p_1, r_{\alpha'}) + (1-\alpha')\nabla_r J_\lambda(\pi_0, \pi_0, p_0, r_{\alpha'})](\nabla_\pi r_{\pi_{\alpha'}} - \nabla_\pi r_{\pi_\alpha})\frac{d\pi_\alpha}{d\alpha} \\
&+ \{\alpha'[\nabla_r J_\lambda(\pi_1, \pi_1, p_1, r_{\alpha'}) - \nabla_r J_\lambda(\pi_1, \pi_1, p_1, r_\alpha)] \\
&+ (1-\alpha')[\nabla_r J_\lambda(\pi_0, \pi_0, p_0, r_{\alpha'}) - \nabla_r J_\lambda(\pi_0, \pi_0, p_0, r_\alpha)]\}(\nabla_\pi r_{\pi_\alpha})\frac{d\pi_\alpha}{d\alpha} \\
&+ (\alpha' - \alpha)[\nabla_r J_\lambda(\pi_1, \pi_1, p_1, r_\alpha) - \nabla_r J_\lambda(\pi_0, \pi_0, p_0, r_\alpha)](\nabla_\pi r_{\pi_\alpha})\frac{d\pi_\alpha}{d\alpha}\Big| \\
\stackrel{(a)}{\le}& \int_\alpha^{\alpha'} \frac{\epsilon_r\|\pi_1 - \pi_0\|}{D(1-\gamma)^2}\big(\max_s\|\pi_1(\cdot|s) - \pi_0(\cdot|s)\|_1 + \gamma\max_{s,a}\|p_1(\cdot|s,a) - p_0(\cdot|s,a)\|_1)d\tilde\alpha \\
&+ \frac{\epsilon_r}{1-\gamma}\cdot 2D^{-2}\|\pi_1 - \pi_0\|^2\Big[\frac{\gamma(\epsilon_p\sqrt{|\mathcal{S}|} + \sqrt{|\mathcal{A}|})}{1-\gamma}\Big]|\alpha' - \alpha| + \frac{S_r\|\pi_{\alpha'} - \pi_\alpha\|}{1-\gamma}\cdot D^{-1}\|\pi_1 - \pi_0\| + 0 \\
&+ |\alpha' - \alpha|\cdot\frac{\epsilon_r\|\pi_1 - \pi_0\|}{D(1-\gamma)^2}\big(\max_s\|\pi_1(\cdot|s) - \pi_0(\cdot|s)\|_1 + \gamma\max_{s,a}\|p_1(\cdot|s,a) - p_0(\cdot|s,a)\|_1) \\
\stackrel{(b)}{\le}& 2|\alpha' - \alpha|\cdot\frac{\epsilon_r\|\pi_1 - \pi_0\|}{D(1-\gamma)^2}\big(\sqrt{|\mathcal{A}|}\|\pi_1 - \pi_0\| + \gamma\sqrt{|\mathcal{S}|}\|p_1 - p_0\|\big) \\
&+ \frac{2\epsilon_r\|\pi_1 - \pi_0\|^2}{D^2(1-\gamma)}\Big[\frac{\gamma(\epsilon_p\sqrt{|\mathcal{S}|} + \sqrt{|\mathcal{A}|})}{1-\gamma}\Big]|\alpha' - \alpha| + \frac{S_r\|\pi_1 - \pi_0\|^2}{D^2(1-\gamma)}|\alpha' - \alpha| \\
\stackrel{(c)}{\le}& \frac{2\epsilon_r\|\pi_1 - \pi_0\|}{D(1-\gamma)^2}\big(\sqrt{|\mathcal{A}|}\|\pi_1 - \pi_0\| + \gamma\epsilon_p\sqrt{|\mathcal{S}|}\|\pi_1 - \pi_0\|\big)|\alpha' - \alpha| \\
&+ \frac{2\gamma\epsilon_r\|\pi_1 - \pi_0\|^2}{D^2(1-\gamma)^2}\big(\sqrt{|\mathcal{A}|} + \epsilon_p\sqrt{|\mathcal{S}|}\big)|\alpha' - \alpha| + \frac{S_r(1-\gamma)\|\pi_1 - \pi_0\|^2}{D^2(1-\gamma)^2}|\alpha' - \alpha| \\
\stackrel{(d)}{\le}& \frac{4\epsilon_r(\sqrt{|\mathcal{A}|} + \gamma\epsilon_p\sqrt{|\mathcal{S}|}) + S_r(1-\gamma)}{D^2(1-\gamma)^2}\|\pi_1 - \pi_0\|^2|\alpha' - \alpha|,
\end{aligned}$$

where (a) uses Assumptions 1-2, $\|\nabla_r J_\lambda(\cdot, \cdot, \cdot, \cdot)\| \le \frac{1}{1-\gamma}$ (implied by Eq. (50)) as well as Eqs. (54), (81) and (83), (b) uses Eq. (81) and $\|x\|_1 \le \sqrt{d}\|x\|$ for any $x \in \mathbb{R}^d$, (c) uses Assumption 1, and (d) uses $D, \gamma \in [0, 1]$. The inequality above implies that $w(\alpha)$ is $\mu_2\|\pi_1 - \pi_0\|^2$-Lipschitz smooth with the constant $\mu_2$ defined as follows.

$$\mu_2 = \frac{4\epsilon_r(\sqrt{|\mathcal{A}|} + \epsilon_p\sqrt{|\mathcal{S}|}) + S_r(1-\gamma)}{D^2(1-\gamma)^2} \tag{94}$$

Therefore,

$$V_{\lambda,\pi_\alpha}^{\pi_\alpha} - \alpha V_{\lambda,\pi_1}^{\pi_1} - (1-\alpha)V_{\lambda,\pi_0}^{\pi_0}$$

$$= J_\lambda(\pi_\alpha, \pi_\alpha, p_\alpha, r_\alpha) - \alpha J_\lambda(\pi_1, \pi_1, p_1, r_1) - (1-\alpha)J_\lambda(\pi_0, \pi_0, p_0, r_0)$$

$$\overset{(a)}{\geq} \alpha J_\lambda(\pi_1, \pi_1, p_1, r_\alpha) + (1-\alpha)J_\lambda(\pi_0, \pi_0, p_0, r_\alpha) + \frac{\mu_1\alpha(1-\alpha)}{2}\|\pi_1 - \pi_0\|^2$$

$$\quad - \alpha J_\lambda(\pi_1, \pi_1, p_1, r_1) - (1-\alpha)J_\lambda(\pi_0, \pi_0, p_0, r_0)$$

$$= w(\alpha) - \alpha w(1) - (1-\alpha)w(0) + \frac{\mu_1\alpha(1-\alpha)}{2}\|\pi_1 - \pi_0\|^2$$

$$\overset{(b)}{\geq} \frac{(\mu_1 - \mu_2)\alpha(1-\alpha)}{2}\|\pi_1 - \pi_0\|^2$$

$$\overset{(c)}{=} \frac{\mu\alpha(1-\alpha)}{2}\|\pi_1 - \pi_0\|^2, \tag{95}$$

where (a) uses Eq. (91) with $r$ replaced by $r_\alpha$, (b) uses the fact proved above that $w(\alpha)$ is $\mu_2\|\pi_1 - \pi_0\|^2$-Lipschitz smooth, and (c) defines the following constant $\mu$ which is the same as Eq. (13).

$$\mu \overset{\text{def}}{=} \mu_1 - \mu_2$$

$$\overset{(a)}{=} \frac{D\lambda}{1-\gamma} - \frac{6\gamma|\mathcal{S}|(1+\lambda\log|\mathcal{A}|)}{D(1-\gamma)^3}\left[\epsilon_p\left(\sqrt{|\mathcal{A}|} + \gamma\epsilon_p\sqrt{|\mathcal{S}|}\right) + S_p(1-\gamma)\right] - \frac{S_r(1-\gamma) + 4\epsilon_r(\sqrt{|\mathcal{A}|} + \epsilon_p\sqrt{|\mathcal{S}|})}{D^2(1-\gamma)^2},$$

where (a) uses Eqs. (92) and (94). Rearranging Eq. (95), we obtain that

$$\frac{V_{\lambda,\pi_\alpha}^{\pi_\alpha} - V_{\lambda,\pi_0}^{\pi_0}}{\alpha} \geq V_{\lambda,\pi_1}^{\pi_1} - V_{\lambda,\pi_0}^{\pi_0} + \frac{\mu(1-\alpha)}{2}\|\pi_1 - \pi_0\|^2.$$

Letting $\alpha \to +0$ above, we can prove the conclusion as follows.

$$V_{\lambda,\pi_1}^{\pi_1} - V_{\lambda,\pi_0}^{\pi_0} + \frac{\mu}{2}\|\pi_1 - \pi_0\|^2$$

$$\leq \left[\frac{d}{d\alpha}V_{\lambda,\pi_\alpha}^{\pi_\alpha}\right]\Big|_{\alpha=0}$$

$$\leq \sum_{s,a}\frac{\partial V_{\lambda,\pi_0}^{\pi_0}}{\partial\pi_0(s,a)}\left[\frac{d}{d\alpha}\pi_\alpha(a|s)\right]\Big|_{\alpha=0}$$

$$\overset{(a)}{=} \sum_s\frac{d_1(s)}{d_0(s)}\sum_a\frac{\partial V_{\lambda,\pi_0}^{\pi_0}}{\partial\pi_0(s,a)}[\pi_1(a|s) - \pi_0(a|s)]$$

$$\leq \sum_s\frac{d_1(s)}{d_0(s)}\left[\max_{a'}\frac{\partial V_{\lambda,\pi_0}^{\pi_0}}{\partial\pi_0(s,a')} - \sum_a\pi_0(a|s)\frac{\partial V_{\lambda,\pi_0}^{\pi_0}}{\partial\pi_0(s,a)}\right]$$

$$\overset{(b)}{\leq} D^{-1}\sum_{s,a}\frac{\partial V_{\lambda,\pi_0}^{\pi_0}}{\partial\pi_0(s,a)}[\pi_0^*(a|s) - \pi_0(a|s)]$$

$$\leq D^{-1}\max_{\pi\in\Pi}\langle\nabla_{\pi_0}V_{\lambda,\pi_0}^{\pi_0}, \pi - \pi_0\rangle,$$

where (a) uses Eq. (80), and (b) uses Assumption 3 as well as the following Eq. (96) where $\pi_0^* \in \Pi$ is defined as $\pi_0^*(a^*|s) = 1$ for a certain $a^* \in \arg\max_{a'}\frac{\partial V_{\lambda,\pi_0}^{\pi_0}}{\partial\pi_0(s,a')}$ and $\pi_0^*(a'|s) = 0$ for $a' \neq a^*$.

$$\sum_a\pi_0^*(a|s)\frac{\partial V_{\lambda,\pi_0}^{\pi_0}}{\partial\pi_0(s,a)} = \max_{a'}\frac{\partial V_{\lambda,\pi_0}^{\pi_0}}{\partial\pi_0(s,a')} \geq \sum_a\pi_0(a|s)\frac{\partial V_{\lambda,\pi_0}^{\pi_0}}{\partial\pi_0(s,a)}. \tag{96}$$

## E. Proof of Corollary 1

Based on Theorem 1, Eq. (12) holds for any $\pi_0, \pi_1 \in \Pi$ as repeated below.

$$V_{\lambda,\pi_1}^{\pi_1} \leq V_{\lambda,\pi_0}^{\pi_0} + D^{-1}\max_{\pi\in\Pi}\langle\nabla_{\pi_0}V_{\lambda,\pi_0}^{\pi_0}, \pi - \pi_0\rangle - \frac{\mu}{2}\|\pi_1 - \pi_0\|^2, \tag{97}$$

In the above inequality, let $\pi_1 \in \arg\max_{\pi \in \Pi} V_{\lambda,\pi}^\pi$ and $\pi_0 = \pi$ is any a $D\epsilon$-stationary policy of interest. Then the inequality above becomes

$$\max_{\tilde{\pi} \in \Pi} V_{\lambda,\tilde{\pi}}^{\tilde{\pi}} \leq V_{\lambda,\pi}^\pi + D^{-1} \cdot D\epsilon - \frac{\mu}{2}\|\pi_1 - \pi\|^2.$$

If $\mu \geq 0$, the inequality above further implies that $\max_{\tilde{\pi} \in \Pi} V_{\lambda,\tilde{\pi}}^{\tilde{\pi}} - V_{\lambda,\pi}^\pi \leq \epsilon$, that is, the $D\epsilon$-stationary policy $\pi$ is also an $\epsilon$-PO policy.

Furthermore, suppose $\mu > 0$ and there are two PO policies $\pi_0, \pi_1 \in \Pi$, which should satisfy

$$V_{\lambda,\pi_1}^{\pi_1} = V_{\lambda,\pi_0}^{\pi_0} = \max_{\pi \in \Pi} V_{\lambda,\pi}^\pi,$$

$$\max_{\pi \in \Pi} \langle \nabla_{\pi_0} V_{\lambda,\pi_0}^{\pi_0}, \pi - \pi_0 \rangle = 0.$$

Substituting the two equalities above into Eq. (97), we obtain that $\frac{\mu}{2}\|\pi_1 - \pi_0\|^2 \leq 0$, which along with $\mu > 0$ implies $\pi_1 = \pi_0$, that is, the PO policy is unique.

## F. Proof of Theorem 2

For any $\pi \in \Pi$, $p \in \mathcal{P}$, $r \in \mathcal{R}$, we have

$$\frac{\partial J_\lambda(\pi,\pi,p,r)}{\partial \pi(a|s)} \overset{(a)}{=} \frac{d_{\pi,p}(s)[Q_\lambda(\pi,\pi,p,r;s,a) - \lambda]}{1 - \gamma}$$

$$\overset{(b)}{=} \frac{d_{\pi,p}(s)}{1-\gamma}\Big[r(s,a) - \lambda - \lambda \log \pi(a|s) + \gamma \sum_{s'} p(s'|s,a)V_\lambda(\pi,p,r;s')\Big], \tag{98}$$

where (a) uses Eqs. (47), and (b) uses Eq. (41).

Then we have

$$\nabla_\pi J_\lambda(\pi,\pi,p,r)^\top(\pi' - \pi)$$

$$= \sum_s \Big[\frac{\partial J_\lambda(\pi,\pi,p,r)}{\partial \pi[a_{\max}(s)|s]}\big(\pi'[a_{\max}(s)|s] - \pi[a_{\max}(s)|s]\big) + \frac{\partial J_\lambda(\pi,\pi,p,r)}{\partial \pi[a_{\min}(s)|s]}\big(\pi'[a_{\min}(s)|s] - \pi[a_{\min}(s)|s]\big)\Big]$$

$$= \sum_s \Big\{\frac{d_{\pi,p}(s)}{1-\gamma}\big(\pi[a_{\max}(s)|s] - \pi[a_{\min}(s)|s]\big)\Big[r[s,a_{\min}(s)] - r[s,a_{\max}(s)] + \lambda \log \frac{\pi[a_{\max}(s)|s]}{\pi[a_{\min}(s)|s]}$$

$$+ \gamma \sum_{s'}[p(s'|s,a_{\min}(s)) - p(s'|s,a_{\max}(s))]V_\lambda(\pi,p,r;s')\Big]\Big\}$$

$$\overset{(a)}{\geq} \frac{1}{1-\gamma}\max_s\Big\{\big(\pi[a_{\max}(s)|s] - \pi[a_{\min}(s)|s]\big)\Big[\lambda \log \frac{\pi[a_{\max}(s)|s]}{\pi[a_{\min}(s)|s]} - 1 - \frac{\gamma(1 + \lambda \log|\mathcal{A}|)}{1-\gamma}\Big]\Big\}, \tag{99}$$

where (a) uses $\pi[a_{\max}(s)|s] - \pi[a_{\min}(s)|s] \geq 0$, $r(a|s) \in [0,1]$, $p(s'|s,a) \in [0,1]$ for any $s, a, s'$ and Lemma 4.

Consider the following two cases.

(Case I) If $\pi[a_{\min}(s)|s] \geq \frac{1}{2}\pi[a_{\max}(s)|s]$, then as $\pi[a_{\max}(s)|s] \geq \frac{1}{|\mathcal{A}|}$, we have $\pi[a_{\min}(s)|s] \geq \frac{1}{2|\mathcal{A}|}$.

(Case II) $\pi[a_{\min}(s)|s] < \frac{1}{2}\pi[a_{\max}(s)|s]$, then as $\pi[a_{\max}(s)|s] \geq \frac{1}{|\mathcal{A}|}$, Eq. (99) implies that

$$\nabla_\pi J_\lambda(\pi,\pi,p,r)^\top(\pi' - \pi)$$

$$\geq \max_s\Big\{\frac{\pi[a_{\max}(s)|s]}{2(1-\gamma)}\Big[\lambda \log \frac{1}{|\mathcal{A}|\pi[a_{\min}(s)|s]} - \frac{1 + \gamma\lambda \log|\mathcal{A}|}{1-\gamma}\Big]\Big\}$$

$$\geq -\frac{1}{2|\mathcal{A}|(1-\gamma)}\Big[\lambda \log\big(|\mathcal{A}|\min_s \pi[a_{\min}(s)|s]\big) + \frac{1 + \gamma\lambda \log|\mathcal{A}|}{1-\gamma}\Big], \tag{100}$$

which further implies that for any $s \in \mathcal{S}$ and $a \in \mathcal{A}$, we have

$$\pi(a|s) \geq \pi[a_{\min}(s)|s] \geq \frac{1}{|\mathcal{A}|} \exp\left[ -\frac{1/\lambda + \gamma \log|\mathcal{A}|}{1-\gamma} - \frac{2|\mathcal{A}|}{\lambda}(1-\gamma)\nabla_\pi J_\lambda(\pi, \pi, p, r)^\top(\pi'-\pi) \right]$$

$$\geq \frac{1}{2|\mathcal{A}|^{1/(1-\gamma)}} \exp\left[ -\frac{1}{\lambda(1-\gamma)} - \frac{2|\mathcal{A}|}{\lambda}(1-\gamma)\nabla_\pi J_\lambda(\pi, \pi, p, r)^\top(\pi'-\pi) \right], \quad (101)$$

Note that in the two cases above, Eq. (101) always holds.

Furthermore, if Assumption 1 holds and $p_\pi, r_\pi$ are differentiable functions of $\pi$, then we have

$$\left\| \nabla_\pi J_\lambda(\pi, \pi, p_\pi, r_\pi) - \nabla_\pi J_\lambda(\pi, \pi, p_{\tilde\pi}, r_{\tilde\pi})|_{\tilde\pi=\pi} \right\|$$

$$= \left\| \nabla_p J_\lambda(\pi, \pi, p_\pi, r_\pi)\nabla_\pi p_\pi + \nabla_r J_\lambda(\pi, \pi, p_\pi, r_\pi)\nabla_\pi r_\pi \right\|$$

$$\leq \left\| \nabla_p J_\lambda(\pi, \pi, p_\pi, r_\pi) \right\|\left\| \nabla_\pi p_\pi \right\| + \left\| \nabla_r J_\lambda(\pi, \pi, p_\pi, r_\pi) \right\|\left\| \nabla_\pi r_\pi \right\|$$

$$\overset{(a)}{\leq} \frac{\epsilon_p \sqrt{|\mathcal{S}|}(1 + \lambda\log|\mathcal{A}|)}{(1-\gamma)^2} + \frac{\epsilon_r}{1-\gamma}, \quad (102)$$

where (a) uses Assumption 1 as well as Eqs. (49) and (50). Therefore,

$$\left[ \nabla_\pi J_\lambda(\pi, \pi, p_{\tilde\pi}, r_{\tilde\pi})|_{\tilde\pi=\pi} \right]^\top(\pi'-\pi)$$

$$= \nabla_\pi J_\lambda(\pi, \pi, p_\pi, r_\pi)^\top(\pi'-\pi) - \left[ \nabla_\pi J_\lambda(\pi, \pi, p_\pi, r_\pi) - \nabla_\pi J_\lambda(\pi, \pi, p_{\tilde\pi}, r_{\tilde\pi})|_{\tilde\pi=\pi} \right]^\top(\pi'-\pi)$$

$$\leq \nabla_\pi J_\lambda(\pi, \pi, p_\pi, r_\pi)^\top(\pi'-\pi) + \left\| \nabla_\pi J_\lambda(\pi, \pi, p_\pi, r_\pi) - \nabla_\pi J_\lambda(\pi, \pi, p_{\tilde\pi}, r_{\tilde\pi})|_{\tilde\pi=\pi} \right\|\left\|\pi'-\pi\right\|$$

$$\overset{(a)}{\leq} \nabla_\pi J_\lambda(\pi, \pi, p_\pi, r_\pi)^\top(\pi'-\pi) + \sqrt{2|\mathcal{S}|}\left( \frac{\epsilon_p \sqrt{|\mathcal{S}|}(1 + \lambda\log|\mathcal{A}|)}{(1-\gamma)^2} + \frac{\epsilon_r}{1-\gamma} \right), \quad (103)$$

where (a) uses Eq. (102) and Lemma 12. Substituting $p = p_\pi$, $r = r_\pi$ and then Eq. (103) into Eq. (101), we can prove Eq. (17) as follows.

$$\pi(a|s) \geq \frac{1}{2|\mathcal{A}|^{1/(1-\gamma)}} \exp\left\{ -\frac{1}{\lambda(1-\gamma)} \right.$$

$$\left. -\frac{2|\mathcal{A}|}{\lambda}(1-\gamma)\left[ \nabla_\pi J_\lambda(\pi, \pi, p_\pi, r_\pi)^\top(\pi'-\pi) + \sqrt{2|\mathcal{S}|}\left( \frac{\epsilon_p \sqrt{|\mathcal{S}|}(1 + \lambda\log|\mathcal{A}|)}{(1-\gamma)^2} + \frac{\epsilon_r}{1-\gamma} \right) \right] \right\}$$

$$= \pi_{\min} \exp\left[ -\frac{2|\mathcal{A}|}{\lambda}(1-\gamma)\langle \nabla_\pi V_{\lambda,\pi}^\pi, \pi'-\pi \rangle \right],$$

where the $=$ uses $V_{\lambda,\pi}^\pi = J_\lambda(\pi, \pi, p_\pi, r_\pi)$ and $\pi_{\min}$ defined by Eq. (18).

# G. Proof of Theorem 3

For any policies $\pi, \pi'$, we have

$$|V_{\lambda,\pi'}^{\pi'} - V_{\lambda,\pi}^\pi|$$

$$\leq |J_\lambda(\pi', p_{\pi'}, r_{\pi'}) - J_\lambda(\pi, p_\pi, r_\pi)|$$

$$\leq |J_\lambda(\pi', p_{\pi'}, r_{\pi'}) - J_\lambda(\pi', p_{\pi'}, r_\pi)| + |J_\lambda(\pi', p_{\pi'}, r_\pi) - J_\lambda(\pi', p_\pi, r_\pi)| + |J_\lambda(\pi', p_\pi, r_\pi) - J_\lambda(\pi, p_\pi, r_\pi)|$$

$$\overset{(a)}{\leq} \frac{\|r_{\pi'} - r_\pi\|}{1-\gamma} + L_p\|p_{\pi'} - p_\pi\| + L_\pi \max_s \|\log\pi'(\cdot|s) - \log\pi(\cdot|s)\|$$

$$\overset{(b)}{\leq} \left( L_p\epsilon_p + \frac{\epsilon_r}{1-\gamma} \right)\|\pi'-\pi\| + L_\pi\sqrt{\sum_s \|\log\pi'(\cdot|s) - \log\pi(\cdot|s)\|^2}$$

$$\overset{(c)}{\leq} \left( L_p\epsilon_p + \frac{\epsilon_r}{1-\gamma} \right)\|\log\pi' - \log\pi\| + L_\pi\|\log\pi' - \log\pi\|$$

$$\overset{(d)}{=} L_\lambda\|\log\pi' - \log\pi\|, \quad (104)$$

where (a) uses Eqs. (48), (49) and (50), (b) uses Assumption 9, (c) uses $|\log y - \log x| \leq |y - x|$ for any $x, y \in \mathbb{R}$, and (d) defines the following constant.

$$L_\lambda = L_p \epsilon_p + \frac{\epsilon_r}{1-\gamma} + L_\pi = \frac{\sqrt{|\mathcal{A}|}(2 - \gamma + \gamma\lambda\log|\mathcal{A}|) + \epsilon_p\sqrt{|\mathcal{S}|}(1 + \lambda\log|\mathcal{A}|) + \epsilon_r(1-\gamma)}{(1-\gamma)^2}.$$

Note that for any $u, v \geq \Delta > 0$,

$$|\log u - \log v| = \log\max(u,v) - \log\min(u,v) = \int_{\min(u,v)}^{\max(u,v)} \frac{1}{x}dx \leq \frac{1}{\Delta}[\max(u,v) - \min(u,v)] = \frac{|u-v|}{\Delta}.$$

Therefore, for any $\pi, \pi' \in \Pi_\Delta \stackrel{\text{def}}{=} \{\pi \in \Pi : \pi(a|s) \geq \Delta\}$, we have

$$\|\log\pi' - \log\pi\|^2 = \sum_{s,a}|\log\pi'(a|s) - \log\pi(a|s)|^2 \leq \Delta^{-2}\sum_{s,a}|\pi'(a|s) - \pi(a|s)|^2 = \Delta^{-2}\|\pi' - \pi\|^2.$$

Substituting the above inequality into Eq. (104) proves Eq. (20).

Next, we will prove Eq. (21) about the Lipschitz continuity of the following performative policy gradient.

$$\begin{aligned}
\nabla_\pi V_{\lambda,\pi}^\pi &= \nabla_\pi J_\lambda(\pi, \pi, p_\pi, r_\pi) \\
&= \nabla_\pi J_\lambda(\pi, \pi, p_{\tilde{\pi}}, r_{\tilde{\pi}})|_{\tilde{\pi}=\pi} + (\nabla_\pi p_\pi)\nabla_{p_\pi} J_\lambda(\pi, \pi, p_\pi, r_\pi) + (\nabla_\pi r_\pi)\nabla_{r_\pi} J_\lambda(\pi, \pi, p_\pi, r_\pi).
\end{aligned} \qquad (105)$$

For any $\pi, \pi' \in \Pi_\Delta$, we prove Eq. (21) as follows.

$$\begin{aligned}
&\|\nabla_{\pi'} V_{\lambda,\pi'}^{\pi'} - \nabla_\pi V_{\lambda,\pi}^\pi\| \\
&\leq \big\|\nabla_{\pi'} J_\lambda(\pi', \pi', p_{\tilde{\pi}}, r_{\tilde{\pi}})|_{\tilde{\pi}=\pi'} - \nabla_\pi J_\lambda(\pi, \pi, p_{\tilde{\pi}}, r_{\tilde{\pi}})|_{\tilde{\pi}=\pi}\big\| \\
&\quad + \|\nabla_{\pi'} p_{\pi'}\| \cdot \|\nabla_{p_{\pi'}} J_\lambda(\pi', \pi', p_{\pi'}, r_{\pi'}) - \nabla_{p_\pi} J_\lambda(\pi, \pi, p_\pi, r_\pi)\| + \|\nabla_{p_\pi} J_\lambda(\pi, \pi, p_\pi, r_\pi)\| \cdot \|\nabla_{\pi'} p_{\pi'} - \nabla_\pi p_\pi\| \\
&\quad + \|\nabla_{\pi'} r_{\pi'}\| \cdot \|\nabla_{r_{\pi'}} J_\lambda(\pi', \pi', p_{\pi'}, r_{\pi'}) - \nabla_{r_\pi} J_\lambda(\pi, \pi, p_\pi, r_\pi)\| + \|\nabla_{r_\pi} J_\lambda(\pi, \pi, p_\pi, r_\pi)\| \cdot \|\nabla_{\pi'} r_{\pi'} - \nabla_\pi r_\pi\| \\
&\stackrel{(a)}{\leq} \left(\frac{|\mathcal{A}|(1 + 2\lambda\log|\mathcal{A}|)}{(1-\gamma)^2} + \gamma L_\pi\right)\max_s\|\log\pi'(\cdot|s) - \log\pi(\cdot|s)\| + \left[\frac{2(1 + \lambda\log|\mathcal{A}|)}{(1-\gamma)^2} + \gamma L_p\right]\sqrt{|\mathcal{S}||\mathcal{A}|}\|p_{\pi'} - p_\pi\| \\
&\quad + \frac{\sqrt{|\mathcal{A}|}\|r_{\pi'} - r_\pi\|_\infty}{1-\gamma} + \epsilon_p\Big[\ell_\pi\max_s\|\log\pi'(\cdot|s) - \log\pi(\cdot|s)\| + \ell_p\|p_{\pi'} - p_\pi\| + \frac{2-\gamma}{1-\gamma}\sqrt{|\mathcal{S}|}\|r_{\pi'} - r_\pi\|_\infty\Big] \\
&\quad + L_p S_p\|\pi' - \pi\| + \frac{\gamma\epsilon_r}{(1-\gamma)^2}\Big(\max_s\|\pi'(\cdot|s) - \pi(\cdot|s)\|_1 + \max_{s,a}\|p_{\pi'}(\cdot|s,a) - p_\pi(\cdot|s,a)\|_1\Big) + \frac{S_r}{1-\gamma}\|\pi' - \pi\| \\
&\stackrel{(b)}{\leq} \left(\frac{|\mathcal{A}|(1 + 2\lambda\log|\mathcal{A}|)}{\Delta(1-\gamma)^2} + \frac{\gamma L_\pi}{\Delta}\right)\|\pi' - \pi\| + \epsilon_p\sqrt{|\mathcal{S}||\mathcal{A}|}\left[\frac{2(1 + \lambda\log|\mathcal{A}|)}{(1-\gamma)^2} + \gamma L_p\right]\|\pi' - \pi\| \\
&\quad + \frac{\epsilon_r\sqrt{|\mathcal{A}|}\|\pi' - \pi\|}{1-\gamma} + \epsilon_p\Big[\frac{\ell_\pi}{\Delta}\|\pi' - \pi\| + \ell_p\epsilon_p\|\pi' - \pi\| + \frac{2-\gamma}{1-\gamma}\epsilon_r\sqrt{|\mathcal{S}|}\|\pi' - \pi\|\Big] \\
&\quad + L_p S_p\|\pi' - \pi\| + \frac{\gamma\epsilon_r}{(1-\gamma)^2}\big(\sqrt{|\mathcal{S}|}\|\pi' - \pi\| + \epsilon_p\sqrt{|\mathcal{S}|}\|\pi' - \pi\|\big) + \frac{S_r}{1-\gamma}\|\pi' - \pi\| \\
&\stackrel{(c)}{\leq} \left(\frac{|\mathcal{A}|(1 + 2\lambda\log|\mathcal{A}|)}{\Delta(1-\gamma)^2} + \frac{\gamma L_\pi}{\Delta}\right)\|\pi' - \pi\| + \frac{\epsilon_p}{\Delta}\sqrt{\frac{|\mathcal{S}|}{|\mathcal{A}|}}\left[\frac{2(1 + \lambda\log|\mathcal{A}|)}{(1-\gamma)^2} + \gamma L_p\right]\|\pi' - \pi\| \\
&\quad + \frac{\epsilon_r\|\pi' - \pi\|}{\Delta\sqrt{|\mathcal{A}|}(1-\gamma)} + \frac{\epsilon_p}{\Delta}\Big[\ell_\pi + \frac{\ell_p\epsilon_p}{|\mathcal{A}|} + \frac{2-\gamma}{|\mathcal{A}|(1-\gamma)}\epsilon_r\sqrt{|\mathcal{S}|}\Big]\|\pi' - \pi\| \\
&\quad + \frac{\gamma\epsilon_r\sqrt{|\mathcal{S}|}(1 + \epsilon_p)}{\Delta|\mathcal{A}|(1-\gamma)^2}\|\pi' - \pi\| + \frac{L_p S_p + S_r/(1-\gamma)}{\Delta|\mathcal{A}|}\|\pi' - \pi\| \\
&\stackrel{(d)}{\leq} \left(\frac{|\mathcal{A}|(1 + 2\lambda\log|\mathcal{A}|)}{\Delta(1-\gamma)^2} + \frac{\gamma\sqrt{|\mathcal{A}|}(2 - \gamma + \gamma\lambda\log|\mathcal{A}|)}{\Delta(1-\gamma)^2}\right)\|\pi' - \pi\| \\
&\quad + \frac{\epsilon_p}{\Delta}\sqrt{\frac{|\mathcal{S}|}{|\mathcal{A}|}}\left[\frac{2(1 + \lambda\log|\mathcal{A}|)}{(1-\gamma)^2} + \frac{\gamma\sqrt{|\mathcal{S}|}(1 + \lambda\log|\mathcal{A}|)}{(1-\gamma)^2}\right]\|\pi' - \pi\|
\end{aligned}$$

$$+ \frac{\epsilon_p}{\Delta}\left[\frac{\sqrt{|\mathcal{S}||\mathcal{A}|}(2+3\gamma\lambda\log|\mathcal{A}|)}{(1-\gamma)^3} + \frac{2\epsilon_p\gamma|\mathcal{S}|(1+\lambda\log|\mathcal{A}|)}{|\mathcal{A}|(1-\gamma)^3} + \frac{2-\gamma}{|\mathcal{A}|(1-\gamma)}\epsilon_r\sqrt{|\mathcal{S}|}\right]\|\pi'-\pi\|$$

$$+ \frac{\epsilon_r\sqrt{|\mathcal{A}|}(1-\gamma)+\gamma\epsilon_r\sqrt{|\mathcal{S}|}(1+\epsilon_p)}{\Delta|\mathcal{A}|(1-\gamma)^2}\|\pi'-\pi\| + \frac{S_p\sqrt{|\mathcal{S}|}(1+\lambda\log|\mathcal{A}|)+S_r(1-\gamma)}{\Delta|\mathcal{A}|(1-\gamma)^2}\|\pi'-\pi\|$$

$$\leq \frac{3|\mathcal{A}|(1+\lambda\log|\mathcal{A}|)}{\Delta(1-\gamma)^2}\|\pi'-\pi\| + \frac{\epsilon_p\sqrt{|\mathcal{S}||\mathcal{A}|}(5+6\lambda\log|\mathcal{A}|)}{\Delta(1-\gamma)^3}\|\pi'-\pi\|$$

$$+ \frac{\epsilon_r\left[\sqrt{|\mathcal{A}|}(1-\gamma)+\sqrt{|\mathcal{S}|}(\gamma+2\epsilon_p)\right]+S_p\sqrt{|\mathcal{S}|}(1+\lambda\log|\mathcal{A}|)+S_r(1-\gamma)}{\Delta|\mathcal{A}|(1-\gamma)^2}\|\pi'-\pi\|$$

$$\overset{(e)}{=} \frac{\ell_\lambda}{\Delta}\|\pi'-\pi\|,$$

where (a) uses Eqs. (49), (50) and (53)-(55) as well as Assumptions 1-2, and (b) uses the following bounds for any $\pi, \pi' \in \Delta$ in which (d) uses Assumption 1, (c) uses $\Delta \leq |\mathcal{A}|^{-1}$ (since for any $\pi \in \Pi_\Delta$, $1 = \sum_a \pi(a|s) \geq \Delta|\mathcal{A}|$), (d) uses $L_\pi := \frac{\sqrt{|\mathcal{A}|}(2-\gamma+\gamma\lambda\log|\mathcal{A}|)}{(1-\gamma)^2}$, $L_p := \frac{\sqrt{|\mathcal{S}|}(1+\lambda\log|\mathcal{A}|)}{(1-\gamma)^2}$, $\ell_\pi := \frac{\sqrt{|\mathcal{S}||\mathcal{A}|}(2+3\gamma\lambda\log|\mathcal{A}|)}{(1-\gamma)^3}$ and $\ell_p := \frac{2\gamma|\mathcal{S}|(1+\lambda\log|\mathcal{A}|)}{(1-\gamma)^3}$ defined in Lemma 6, (e) uses $\ell_\lambda$ defined by Eq. (23).

$$\max_s \|\log\pi'(\cdot|s) - \log\pi(\cdot|s)\| \leq \Delta^{-1}\max_s\|\pi'(\cdot|s)-\pi(\cdot|s)\| \leq \Delta^{-1}\|\pi'-\pi\|,$$

$$\|p_{\pi'}-p_\pi\| \overset{(d)}{\leq} \epsilon_p\|\pi'-\pi\|,$$

$$\|r_{\pi'}-r_\pi\|_\infty \leq \|r_{\pi'}-r_\pi\| \overset{(d)}{\leq} \epsilon_r\|\pi'-\pi\|,$$

$$\max_s\|\pi'(\cdot|s)-\pi(\cdot|s)\|_1 \leq \sqrt{|\mathcal{S}|}\max_s\|\pi'(\cdot|s)-\pi(\cdot|s)\| \leq \sqrt{|\mathcal{S}|}\|\pi'-\pi\|,$$

$$\max_{s,a}\|p_{\pi'}(\cdot|s,a)-p_\pi(\cdot|s,a)\|_1 \leq \sqrt{|\mathcal{S}|}\max_{s,a}\|p_{\pi'}(\cdot|s,a)-p_\pi(\cdot|s,a)\| \leq \sqrt{|\mathcal{S}|}\|p_{\pi'}-p_\pi\| \overset{(d)}{\leq} \epsilon_p\sqrt{|\mathcal{S}|}\|\pi'-\pi\|.$$

# H. Proof of Proposition 1

We prove the validity of the stochastic gradient (26) first. For any $\pi \in \Pi_\Delta$, $s \in \mathcal{S}$ and $a \in \mathcal{A}$, we have $\pi(a|s) \geq \Delta$, so $\pi(a|s) \leq 1 - \Delta$ (since $\sum_{a'}\pi(a'|s) = 1$). For any $u_i \in U_1$, we have $|u_i(a|s)| \leq 1$. Therefore,

$$(\pi \pm \delta u_i)(a|s) \geq \pi(a|s) - \delta|u_i(a|s)| \geq \Delta - \delta > 0, \tag{106}$$

which means $\pi \pm \delta u_i \in \Pi$. Hence, $V_{\lambda,\pi'}^{\pi'}$ is well defined for $\pi' \in \{\pi + \delta u_i, \pi - \delta u_i\}$.

Then we will prove the estimation error (30). Based on Lemma 10, there exists an orthogonal transformation $\mathcal{T} : \mathbb{R}^{|\mathcal{A}|} \to \mathcal{Z}_{|\mathcal{A}|-1} = \{z = [z_1, \ldots, z_{|\mathcal{A}|}] \in \mathbb{R}^{|\mathcal{A}|} : \sum_i z_i = 0\}$.

Note that any $x \in \mathbb{R}^{|\mathcal{S}|(|\mathcal{A}|-1)}$ can be written as $x = [x_s]_{s\in\mathcal{S}}$, a concatenation of $|\mathcal{S}|$ vectors $x_s \in \mathbb{R}^{|\mathcal{A}|}$. Therefore, we can define the transformation $T : \mathbb{R}^{|\mathcal{S}|(|\mathcal{A}|-1)} \to \mathcal{L}_0 \overset{\text{def}}{=} \{u \in \mathbb{R}^{|\mathcal{S}||\mathcal{A}|} : u(\cdot|s) \in \mathcal{Z}_{|\mathcal{A}|-1}, \forall s \in \mathcal{S}\}$ as follows

$$[T(x)](\cdot|s) = \mathcal{T}(x_s), \forall s \in \mathcal{S} \tag{107}$$

where $x_s \in \mathbb{R}^{|\mathcal{A}|}$ are extracted from $|\mathcal{A}|$ entries of $x = [x_s]_{s\in\mathcal{S}}$. For any $x = [x_s]_{s\in\mathcal{S}}, y = [y_s]_{s\in\mathcal{S}} \in \mathbb{R}^{|\mathcal{S}|(|\mathcal{A}|-1)}$ and $\alpha, \beta \in \mathbb{R}$, we can prove that $T$ is an orthogonal transformation as follows.

$$[T(\alpha x + \beta y)](\cdot|s) = \mathcal{T}(\alpha x_s + \beta y_s) = \alpha\mathcal{T}(x_s) + \beta\mathcal{T}(y_s) = \alpha[T(x)](\cdot|s) + \beta[T(x)](\cdot|s)$$
$$\Rightarrow T(\alpha x + \beta y) = \alpha T(x) + \beta T(y).$$

$$\langle T(x), T(y)\rangle = \sum_s \langle [T(x)](\cdot|s), [T(y)](\cdot|s)\rangle = \sum_s \langle \mathcal{T}(x_s), \mathcal{T}(y_s)\rangle = \sum_s \langle x_s, y_s\rangle = \langle x, y\rangle.$$

Define the following set.

$$T^{-1}(\Pi_\Delta - |\mathcal{A}|^{-1}) \overset{\text{def}}{=} \{\pi \in \Pi_\Delta : T^{-1}(\pi - |\mathcal{A}|^{-1})\}, \tag{108}$$

where $\pi - |\mathcal{A}|^{-1} \in \mathbb{R}^{|\mathcal{S}||\mathcal{A}|}$ has entries $(\pi - |\mathcal{A}|^{-1})(a|s) = \pi(a|s) - |\mathcal{A}|^{-1}$, so $\pi - |\mathcal{A}|^{-1} \in \mathcal{L}_0$. Furthermore, since $\Pi_\Delta$ is a convex and compact set and $T^{-1}$ is an orthogonal transformation, $T^{-1}(\Pi_\Delta - |\mathcal{A}|^{-1})$ is a convex and compact subset of $\mathcal{L}_0$.

Then for any $x \in T^{-1}(\Pi_\Delta - |\mathcal{A}|^{-1})$, we have $T(x) + |\mathcal{A}|^{-1} \in \Pi_\Delta$, so we can define the function $f_\lambda(x) \overset{\text{def}}{=} V_{\lambda,T(x)+|\mathcal{A}|^{-1}}^{T(x)+|\mathcal{A}|^{-1}}$.

Note that as $V_{\lambda,\pi}^\pi$ is a differentiable function of $\pi$, so for any $\pi' \in \Pi$ and fixed $\pi \in \Pi$ we have

$$\frac{V_{\lambda,\pi'}^{\pi'} - V_{\lambda,\pi}^\pi - \langle \nabla_\pi V_{\lambda,\pi}^\pi, \pi' - \pi \rangle}{\|\pi' - \pi\|} = \frac{V_{\lambda,\pi'}^{\pi'} - V_{\lambda,\pi}^\pi - \langle \text{proj}_{\mathcal{L}_0}(\nabla_\pi V_{\lambda,\pi}^\pi), \pi' - \pi \rangle}{\|\pi' - \pi\|} \to 0 \text{ as } \pi' \in \Pi \text{ and } \pi' \to \pi, \quad (109)$$

where the above $=$ uses $\pi' - \pi \in \mathcal{L}_0$. Then, we can prove that $f_\lambda$ is differentiable with gradient $\nabla f_\lambda(x) = T^{-1}\big(\text{proj}_{\mathcal{L}_0} \nabla_\pi V_{\lambda,\pi}^\pi \big|_{\pi=T(x)+|\mathcal{A}|^{-1}}\big)$, since for any $x' \in T^{-1}(\Pi_\Delta - |\mathcal{A}|^{-1})$ and fixed $x \in T^{-1}(\Pi_\Delta - |\mathcal{A}|^{-1})$ we have

$$\frac{f_\lambda(x') - f_\lambda(x) - \big\langle T^{-1}\big[\text{proj}_{\mathcal{L}_0}\big(\nabla_\pi V_{\lambda,\pi}^\pi \big|_{\pi=T(x)+|\mathcal{A}|^{-1}}\big)\big], x' - x\big\rangle}{\|x' - x\|}$$

$$\overset{(a)}{=} \frac{V_{\lambda,T(x')+|\mathcal{A}|^{-1}}^{T(x')+|\mathcal{A}|^{-1}} - V_{\lambda,T(x)+|\mathcal{A}|^{-1}}^{T(x)+|\mathcal{A}|^{-1}} - \big\langle \text{proj}_{\mathcal{L}_0}\big(\nabla_\pi V_{\lambda,\pi}^\pi \big|_{\pi=T(x)+|\mathcal{A}|^{-1}}\big), [T(x')+|\mathcal{A}|^{-1}] - [T(x)+|\mathcal{A}|^{-1}]\big\rangle}{\big\|[T(x')+|\mathcal{A}|^{-1}] - [T(x)+|\mathcal{A}|^{-1}]\big\|}$$

$$\overset{(b)}{\to} 0 \text{ as } x' \in T^{-1}(\Pi_\Delta - |\mathcal{A}|^{-1}) \text{ and } x' \to x, \tag{110}$$

where (a) uses the property of the orthogonal transformation $T$, and (b) uses Eq. (109) and the fact that $x' \to x$ means $\big\|[T(x')+|\mathcal{A}|^{-1}] - [T(x)+|\mathcal{A}|^{-1}]\big\| = \|x' - x\| \to 0$.

Furthermore, we will show that $f_\lambda(x)$ is a Lipscthiz continuous and Lipschitz smooth function of $x \in \Pi_\Delta$. For any $x, x' \in T^{-1}(\Pi_\Delta - |\mathcal{A}|^{-1})$, we have

$$|f_\lambda(x') - f_\lambda(x)| = \big|V_{\lambda,T(x')+|\mathcal{A}|^{-1}}^{T(x')+|\mathcal{A}|^{-1}} - V_{\lambda,T(x)+|\mathcal{A}|^{-1}}^{T(x)+|\mathcal{A}|^{-1}}\big| \overset{(a)}{\leq} \frac{L_\lambda}{\Delta}\|T(x') - T(x)\| \overset{(b)}{=} \frac{L_\lambda}{\Delta}\|x' - x\|,$$

$$\|\nabla f_\lambda(x') - \nabla f_\lambda(x)\| = \big\|T^{-1}\big[\text{proj}_{\mathcal{L}_0}\big(\nabla_\pi V_{\lambda,\pi}^\pi \big|_{\pi=T(x')}\big)\big] - T^{-1}\big[\text{proj}_{\mathcal{L}_0}\big(\nabla_\pi V_{\lambda,\pi}^\pi \big|_{\pi=T(x)}\big)\big]\big\|$$

$$\overset{(b)}{=} \big\|\text{proj}_{\mathcal{L}_0}\big(\nabla_\pi V_{\lambda,\pi}^\pi \big|_{\pi=T(x')+|\mathcal{A}|^{-1}}\big) - \text{proj}_{\mathcal{L}_0}\big(\nabla_\pi V_{\lambda,\pi}^\pi \big|_{\pi=T(x)+|\mathcal{A}|^{-1}}\big)\big\|$$

$$\leq \big\|\big(\nabla_\pi V_{\lambda,\pi}^\pi \big|_{\pi=T(x')+|\mathcal{A}|^{-1}}\big) - \big(\nabla_\pi V_{\lambda,\pi}^\pi \big|_{\pi=T(x)+|\mathcal{A}|^{-1}}\big)\big\|$$

$$\overset{(a)}{\leq} \frac{\ell_\lambda}{\Delta}\|T(x') - T(x)\| \overset{(b)}{=} \frac{\ell_\lambda}{\Delta}\|x' - x\|,$$

In both the inequalities above, (a) applies Theorem 3 to $T(x) + |\mathcal{A}|^{-1}, T(x') + |\mathcal{A}|^{-1} \in \Pi_\Delta$ and (b) uses the property of the orthogonal transformation $T$. The two inequalities above implies that $f_\lambda$ is an $\frac{L_\lambda}{\Delta}$-Lipschitz continuous and $\frac{\ell_\lambda}{\Delta}$-Lipschitz smooth function on $T^{-1}(\Pi_\Delta - |\mathcal{A}|^{-1})$.

Denote

$$g_{\lambda,\delta}(\pi) = \frac{|\mathcal{S}|(|\mathcal{A}|-1)}{2N\delta} \sum_{i=1}^N \big(V_{\lambda,\pi+\delta u_i}^{\pi+\delta u_i} - V_{\lambda,\pi-\delta u_i}^{\pi-\delta u_i}\big)u_i, \tag{111}$$

which replaces $\hat{V}_{\lambda,\pi'}^{\pi'}$ with $V_{\lambda,\pi'}^{\pi'}$ in Eq. (26). The estimation error of the performative policy gradient estimator above can be rewritten as follows for any $\pi \in \Pi_\Delta$.

$$g_{\lambda,\delta}(\pi) - \text{proj}_{\mathcal{L}_0}(\nabla_\pi V_{\lambda,\pi}^\pi)$$

$$\overset{(a)}{=} \Big(\frac{|\mathcal{S}|(|\mathcal{A}|-1)}{2N\delta} \sum_{i=1}^N \big(V_{\lambda,\pi+\delta u_i}^{\pi+\delta u_i} - V_{\lambda,\pi-\delta u_i}^{\pi-\delta u_i}\big)u_i\Big) - \text{proj}_{\mathcal{L}_0}(\nabla_\pi V_{\lambda,\pi}^\pi)$$

$$\overset{(b)}{=} \Big(\frac{|\mathcal{S}|(|\mathcal{A}|-1)}{2N\delta} \sum_{i=1}^N \big(f_\lambda\big[T^{-1}(\pi - |\mathcal{A}|^{-1}) + \delta T^{-1}(u_i)\big] - f_\lambda\big[T^{-1}(\pi - |\mathcal{A}|^{-1}) - \delta T^{-1}(u_i)\big]\big)T^{-1}(u_i)\Big)$$

$$- T^{-1}[\text{proj}_{\mathcal{L}_0}(\nabla_\pi V_{\lambda,\pi}^\pi)]$$

$$\overset{(c)}{=} \Big( \frac{|\mathcal{S}|(|\mathcal{A}|-1)}{2N\delta} \sum_{i=1}^N \big( f_\lambda \big[ T^{-1}(\pi - |\mathcal{A}|^{-1}) + \delta T^{-1}(u_i) \big] - f_\lambda \big[ T^{-1}(\pi - |\mathcal{A}|^{-1}) \big] - \delta T^{-1}(u_i) \big] \big) T^{-1}(u_i) \Big)$$

$$- \nabla f_\lambda \big[ T^{-1}(\pi - |\mathcal{A}|^{-1}) \big], \tag{112}$$

where (a) uses Eq. (26), (b) uses $f_\lambda(x) \overset{\text{def}}{=} V_{\lambda, T(x)+|\mathcal{A}|^{-1}}^{T(x)+|\mathcal{A}|^{-1}}$ and the property of the orthogonal transformation $T^{-1}$, (c) uses $\nabla f_\lambda(x) = T^{-1}\big(\text{proj}_{\mathcal{L}_0} \nabla_\pi V_{\lambda,\pi}^\pi \big|_{\pi = T(x)+|\mathcal{A}|^{-1}}\big)$. Note that in the above Eq. (112), $\pi \in \Pi_\Delta$ and $u_i$ is uniformly distributed on the sphere $U_1 \cap \mathcal{L}_0$ with $U_1$ defined by Eq. (27), as repeated below.

$$U_1 \overset{\text{def}}{=} \{ u \in \mathbb{R}^{|\mathcal{S}||\mathcal{A}|} : \|u\| = 1 \}. \tag{113}$$

Hence, $\pi \pm \delta u_i \in \Pi_{\Delta-\delta}$ which implies $T^{-1}(\pi - |\mathcal{A}|^{-1}) \pm \delta T^{-1}(u_i) = T^{-1}(\pi \pm \delta u_i - |\mathcal{A}|^{-1}) \in T^{-1}(\Pi_{\Delta-\delta} - |\mathcal{A}|^{-1})$. Also, $T^{-1}(u_i)$ is uniformly distributed on the sphere $T^{-1}(U_{1,0}) = \mathbb{S}_{|\mathcal{S}|(|\mathcal{A}|-1)} = \{ u \in \mathbb{R}^{|\mathcal{S}|(|\mathcal{A}|-1)} : \|u\| = 1 \}$. Therefore, we can apply Lemma 9 to the above Eq. (112) where the function $f_\lambda$ is an $\frac{L_\lambda}{\Delta-\delta}$-Lipschitz continuous and $\frac{\ell_\lambda}{\Delta-\delta}$-Lipschitz smooth function on $T^{-1}(\Pi_{\Delta-\delta} - |\mathcal{A}|^{-1})$, and obtain the following bound which holds with probability at least $1-\eta$.

$$\|g_{\lambda,\delta}(\pi) - \text{proj}_{\mathcal{L}_0}(\nabla_\pi V_{\lambda,\pi}^\pi)\|$$

$$\leq \frac{4L_\lambda|\mathcal{S}|(|\mathcal{A}|-1)}{3N(\Delta-\delta)} \log\Big( \frac{|\mathcal{S}|(|\mathcal{A}|-1)+1}{\eta} \Big) + \frac{L_\lambda|\mathcal{S}|(|\mathcal{A}|-1)}{\Delta-\delta} \sqrt{\frac{2}{N} \log\Big( \frac{|\mathcal{S}|(|\mathcal{A}|-1)+1}{\eta} \Big)} + \frac{\delta\ell_\lambda}{\Delta-\delta}$$

$$\leq \frac{4L_\lambda|\mathcal{S}||\mathcal{A}|}{3N(\Delta-\delta)} \log\Big( \frac{|\mathcal{S}||\mathcal{A}|}{\eta} \Big) + \frac{L_\lambda|\mathcal{S}||\mathcal{A}|}{\Delta-\delta} \sqrt{\frac{2}{N} \log\Big( \frac{|\mathcal{S}||\mathcal{A}|}{\eta} \Big)} + \frac{\delta\ell_\lambda}{\Delta-\delta}. \tag{114}$$

Note that Eq. (24) holds for any a certain policy $\pi$ with probability at least $1-\eta$. Therefore, with probability at least $1 - 2N\eta$, we have

$$|\hat{V}_{\lambda,\pi'}^{\pi'} - V_{\lambda,\pi'}^{\pi'}| \leq \epsilon_V, \forall \pi' \in \{\pi \pm \delta u_i\}_{i=1}^N \tag{115}$$

Therefore, with probability at least $1 - (2N+1)\eta$, Eqs. (114) and (115) hold and thus we have

$$\|\hat{g}_{\lambda,\delta}(\pi) - \text{proj}_{\mathcal{L}_0}(\nabla_\pi V_{\lambda,\pi}^\pi)\|$$

$$\leq \|\hat{g}_{\lambda,\delta}(\pi) - g_{\lambda,\delta}(\pi)\| + \|g_{\lambda,\delta}(\pi) - \text{proj}_{\mathcal{L}_0}(\nabla_\pi V_{\lambda,\pi}^\pi)\|$$

$$\overset{(a)}{\leq} \Big\| \frac{|\mathcal{S}|(|\mathcal{A}|-1)}{2N\delta} \sum_{i=1}^N \big( \hat{V}_{\lambda,\pi+\delta u_i}^{\pi+\delta u_i} - V_{\lambda,\pi+\delta u_i}^{\pi+\delta u_i} - \hat{V}_{\lambda,\pi-\delta u_i}^{\pi-\delta u_i} + V_{\lambda,\pi-\delta u_i}^{\pi-\delta u_i} \big) u_i \Big\|$$

$$+ \frac{4L_\lambda|\mathcal{S}||\mathcal{A}|}{3N(\Delta-\delta)} \log\Big( \frac{|\mathcal{S}||\mathcal{A}|}{\eta} \Big) + \frac{L_\lambda|\mathcal{S}||\mathcal{A}|}{\Delta-\delta} \sqrt{\frac{2}{N} \log\Big( \frac{|\mathcal{S}||\mathcal{A}|}{\eta} \Big)} + \frac{\delta\ell_\lambda}{\Delta-\delta}$$

$$\overset{(b)}{\leq} \frac{|\mathcal{S}||\mathcal{A}|}{N\delta} \sum_{i=1}^N \big\| \big( \hat{V}_{\lambda,\pi+\delta u_i}^{\pi+\delta u_i} - V_{\lambda,\pi+\delta u_i}^{\pi+\delta u_i} - \hat{V}_{\lambda,\pi-\delta u_i}^{\pi-\delta u_i} + V_{\lambda,\pi-\delta u_i}^{\pi-\delta u_i} \big) u_i \big\|$$

$$+ \frac{4L_\lambda|\mathcal{S}||\mathcal{A}|}{3N(\Delta-\delta)} \log\Big( \frac{|\mathcal{S}||\mathcal{A}|}{\eta} \Big) + \frac{L_\lambda|\mathcal{S}||\mathcal{A}|}{\Delta-\delta} \sqrt{\frac{2}{N} \log\Big( \frac{|\mathcal{S}||\mathcal{A}|}{\eta} \Big)} + \frac{\delta\ell_\lambda}{\Delta-\delta}$$

$$\leq \frac{|\mathcal{S}||\mathcal{A}|}{N\delta} \sum_{i=1}^N \big( |\hat{V}_{\lambda,\pi+\delta u_i}^{\pi+\delta u_i} - V_{\lambda,\pi+\delta u_i}^{\pi+\delta u_i}| + |\hat{V}_{\lambda,\pi-\delta u_i}^{\pi-\delta u_i} + V_{\lambda,\pi-\delta u_i}^{\pi-\delta u_i}| \big)$$

$$+ \frac{4L_\lambda|\mathcal{S}||\mathcal{A}|}{3N(\Delta-\delta)} \log\Big( \frac{|\mathcal{S}||\mathcal{A}|}{\eta} \Big) + \frac{L_\lambda|\mathcal{S}||\mathcal{A}|}{\Delta-\delta} \sqrt{\frac{2}{N} \log\Big( \frac{|\mathcal{S}||\mathcal{A}|}{\eta} \Big)} + \frac{\delta\ell_\lambda}{\Delta-\delta}$$

$$\overset{(c)}{\leq} \frac{2|\mathcal{S}||\mathcal{A}|\epsilon_V}{\delta} + \frac{4L_\lambda|\mathcal{S}||\mathcal{A}|}{3N(\Delta-\delta)} \log\Big( \frac{|\mathcal{S}||\mathcal{A}|}{\eta} \Big) + \frac{L_\lambda|\mathcal{S}||\mathcal{A}|}{\Delta-\delta} \sqrt{\frac{2}{N} \log\Big( \frac{|\mathcal{S}||\mathcal{A}|}{\eta} \Big)} + \frac{\delta\ell_\lambda}{\Delta-\delta}, \tag{116}$$

where (a) uses Eqs. (26), (64) and (114), (b) uses Jensen's inequality that $\|\frac{1}{N}\sum_{i=1}^{N}x_i\|^2 \le \frac{1}{N}\sum_{i=1}^{N}\|x_i\|^2$ for any vectors $\{x_i\}_{i=1}^{N}$ of the same dimensionality, (c) uses Eq. (24). The conclusion can be proved by replacing $\eta$ with $\frac{\eta}{3N}$ in the inequality above.

## I. Proof of Proposition 2

For any $\pi \in \Pi_\Delta$, it is easily seen that the corresponding $\pi'$ defined by Eq. (19) also belongs to $\Pi_\Delta$. Therefore,

$$\langle \nabla_\pi V_{\lambda,\pi}^\pi, \pi' - \pi \rangle \le \max_{\tilde{\pi}\in\Pi_\Delta} \langle \nabla_\pi V_{\lambda,\pi}^\pi, \tilde{\pi} - \pi \rangle \le \frac{D\lambda}{5|\mathcal{A}|(1-\gamma)}.$$

Substituting the above inequality into Eq. (17), we obtain that

$$\pi(a|s) \ge \pi_{\min}\exp\left[-\frac{2|\mathcal{A}|}{D\lambda}(1-\gamma)\langle \nabla_\pi V_{\lambda,\pi}^\pi, \pi' - \pi \rangle\right] \ge \frac{2\pi_{\min}}{3} \ge 2\Delta.$$

Therefore, for any $\pi_2 \in \Pi$, we can prove that $\frac{\pi_2+\pi}{2} \in \Pi_\Delta$ as follows.

$$\frac{\pi_2(a|s)+\pi(a|s)}{2} \ge \frac{0+2\Delta}{2} = \Delta.$$

Therefore, we can prove Eq. (34) as follows.

$$\max_{\pi_2\in\Pi}\langle \nabla_\pi V_{\lambda,\pi}^\pi, \pi_2 - \pi\rangle = 2\max_{\pi_2\in\Pi}\left\langle \nabla_\pi V_{\lambda,\pi}^\pi, \frac{\pi_2+\pi}{2} - \pi\right\rangle \overset{(a)}{\le} 2\max_{\tilde{\pi}\in\Pi_\Delta}\langle \nabla_\pi V_{\lambda,\pi}^\pi, \tilde{\pi} - \pi\rangle.$$

where (a) uses $\frac{\pi_2+\pi}{2} \in \Pi_\Delta$.

## J. Proof of Theorem 4

If $\pi_t \in \Pi_\Delta$, then $\pi_{t+1} \in \Pi_\Delta$, since $\Pi_\Delta$ is a convex set and $\pi_{t+1}$ obtained by Eq. (32) is a convex combination of $\pi_t, \tilde{\pi}_t \in \Pi_\Delta$. Since $\pi_0 \in \Pi_\Delta$, we have $\pi_t \in \Pi_\Delta$ for all $t$ by induction. Therefore, Proposition 1 implies that the following bound holds simultaneously for all $\{\pi_t\}_{t=1}^{T} \subseteq \Pi_\Delta$ with probability at least $1-\eta$.

$$\|\hat{g}_{\lambda,\delta}(\pi_t) - \text{proj}_{\mathcal{L}_0}(\nabla_\pi V_{\lambda,\pi_t}^{\pi_t})\|$$
$$\le \frac{2|\mathcal{S}||\mathcal{A}|\epsilon_V}{\delta} + \frac{4L_\lambda|\mathcal{S}||\mathcal{A}|}{3TN(\Delta-\delta)}\log\left(\frac{3TN|\mathcal{S}||\mathcal{A}|}{\eta}\right) + \frac{L_\lambda|\mathcal{S}||\mathcal{A}|}{\Delta-\delta}\sqrt{\frac{2}{N}\log\left(\frac{3TN|\mathcal{S}||\mathcal{A}|}{\eta}\right)} + \frac{\delta\ell_\lambda}{\Delta-\delta}. \tag{117}$$

The bound above further implies that for any $\pi \in \Pi$, we have

$$\left|\langle \hat{g}_{\lambda,\delta}(\pi_t) - \nabla_\pi V_{\lambda,\pi_t}^{\pi_t}, \pi - \pi_t\rangle\right|$$
$$\overset{(a)}{=} \left|\langle \hat{g}_{\lambda,\delta}(\pi_t) - \text{proj}_{\mathcal{L}_0}(\nabla_\pi V_{\lambda,\pi_t}^{\pi_t}), \pi - \pi_t\rangle\right|$$
$$\le \|\hat{g}_{\lambda,\delta}(\pi_t) - \text{proj}_{\mathcal{L}_0}(\nabla_\pi V_{\lambda,\pi_t}^{\pi_t})\| \cdot \|\pi - \pi_t\|$$
$$\overset{(b)}{\le} \sqrt{2|\mathcal{S}|}\left[\frac{2|\mathcal{S}||\mathcal{A}|\epsilon_V}{\delta} + \frac{4L_\lambda|\mathcal{S}||\mathcal{A}|}{3TN(\Delta-\delta)}\log\left(\frac{3TN|\mathcal{S}||\mathcal{A}|}{\eta}\right) + \frac{L_\lambda|\mathcal{S}||\mathcal{A}|}{\Delta-\delta}\sqrt{\frac{2}{N}\log\left(\frac{3TN|\mathcal{S}||\mathcal{A}|}{\eta}\right)} + \frac{\delta\ell_\lambda}{\Delta-\delta}\right], \tag{118}$$

where (a) uses $\tilde{\pi}_t - \pi_t, \tilde{\pi} - \pi_t \in \mathcal{L}_0$ for $\tilde{\pi}_t, \tilde{\pi} \in \Pi_\Delta$, and (b) uses Eq. (117) and Lemma 12.

Under the conditions above, we have

$$V_{\lambda,\pi_{t+1}}^{\pi_{t+1}}$$
$$\overset{(a)}{\ge} V_{\lambda,\pi_t}^{\pi_t} + \langle \nabla_\pi V_{\lambda,\pi_t}^{\pi_t}, \pi_{t+1} - \pi_t\rangle - \frac{\ell_\lambda}{2\Delta}\|\pi_{t+1} - \pi_t\|^2$$
$$\overset{(b)}{=} V_{\lambda,\pi_t}^{\pi_t} + \beta\langle \nabla_\pi V_{\lambda,\pi_t}^{\pi_t}, \tilde{\pi}_t - \pi_t\rangle - \frac{\ell_\lambda\beta^2}{2\Delta}\|\tilde{\pi}_t - \pi_t\|^2$$

$$= V_{\lambda,\pi_t}^{\pi_t} + \beta \langle \hat{g}_{\lambda,\delta}(\pi_t), \tilde{\pi}_t - \pi_t \rangle + \beta \langle \nabla_\pi V_{\lambda,\pi_t}^{\pi_t} - \hat{g}_{\lambda,\delta}(\pi_t), \tilde{\pi}_t - \pi_t \rangle - \frac{\ell_\lambda \beta^2}{2\Delta} \|\tilde{\pi}_t - \pi_t\|^2$$

$$\overset{(c)}{\geq} V_{\lambda,\pi_t}^{\pi_t} + \beta \langle \hat{g}_{\lambda,\delta}(\pi_t), \tilde{\pi}_t - \pi_t \rangle - \frac{\ell_\lambda |\mathcal{S}| \beta^2}{\Delta}$$

$$- \beta \sqrt{2|\mathcal{S}|} \Big[ \frac{2|\mathcal{S}||\mathcal{A}|\epsilon_V}{\delta} + \frac{4L_\lambda |\mathcal{S}||\mathcal{A}|}{3TN(\Delta-\delta)} \log\Big(\frac{3TN|\mathcal{S}||\mathcal{A}|}{\eta}\Big) + \frac{L_\lambda |\mathcal{S}||\mathcal{A}|}{\Delta-\delta} \sqrt{\frac{2}{N} \log\Big(\frac{3TN|\mathcal{S}||\mathcal{A}|}{\eta}\Big)} + \frac{\delta \ell_\lambda}{\Delta-\delta} \Big], \quad (119)$$

where (a) uses the $\frac{\ell_\lambda}{\Delta}$-Lipschitz smoothness of $V_{\lambda,\pi}^\pi$ on $\Pi_\Delta$, (b) uses Eq. (32), (c) uses Eq. (118) and Lemma 12.

Rearranging and averaging Eq. (119) over $t = 0, 1, \ldots, T-1$, we obtain that

$$\max_{\tilde{\pi} \in \Pi_\Delta} \langle \hat{g}_{\lambda,\delta}(\pi_{\tilde{T}}), \tilde{\pi} - \pi_{\tilde{T}} \rangle$$

$$\overset{(a)}{=} \langle \hat{g}_{\lambda,\delta}(\pi_{\tilde{T}}), \tilde{\pi}_{\tilde{T}} - \pi_{\tilde{T}} \rangle$$

$$\overset{(b)}{\leq} \frac{1}{T} \sum_{t=0}^{T-1} \langle \hat{g}_{\lambda,\delta}(\pi_t), \tilde{\pi}_t - \pi_t \rangle$$

$$\leq \frac{V_{\lambda,\pi_T}^{\pi_T} - V_{\lambda,\pi_0}^{\pi_0}}{T\beta} + \frac{\ell_\lambda |\mathcal{S}| \beta}{\Delta}$$

$$+ \sqrt{2|\mathcal{S}|} \Big[ \frac{2|\mathcal{S}||\mathcal{A}|\epsilon_V}{\delta} + \frac{4L_\lambda |\mathcal{S}||\mathcal{A}|}{3TN(\Delta-\delta)} \log\Big(\frac{3TN|\mathcal{S}||\mathcal{A}|}{\eta}\Big) + \frac{L_\lambda |\mathcal{S}||\mathcal{A}|}{\Delta-\delta} \sqrt{\frac{2}{N} \log\Big(\frac{3TN|\mathcal{S}||\mathcal{A}|}{\eta}\Big)} + \frac{\delta \ell_\lambda}{\Delta-\delta} \Big]$$

$$\leq \frac{1 + \lambda \log|\mathcal{A}|}{T\beta(1-\gamma)} + \frac{\ell_\lambda |\mathcal{S}| \beta}{\Delta}$$

$$+ \sqrt{2|\mathcal{S}|} \Big[ \frac{2|\mathcal{S}||\mathcal{A}|\epsilon_V}{\delta} + \frac{4L_\lambda |\mathcal{S}||\mathcal{A}|}{3TN(\Delta-\delta)} \log\Big(\frac{3TN|\mathcal{S}||\mathcal{A}|}{\eta}\Big) + \frac{L_\lambda |\mathcal{S}||\mathcal{A}|}{\Delta-\delta} \sqrt{\frac{2}{N} \log\Big(\frac{3TN|\mathcal{S}||\mathcal{A}|}{\eta}\Big)} + \frac{\delta \ell_\lambda}{\Delta-\delta} \Big], \quad (120)$$

where (a) uses Lemma 1 which means $\tilde{\pi}_t$ satisfies Eq. (31) and (b) uses the output rule of Algorithm 1 that $\tilde{T} \in \arg\min_{0 \leq t \leq T-1} \langle \hat{g}_{\lambda,\delta}(\pi_t), \tilde{\pi}_t - \pi_t \rangle$. Therefore,

$$\max_{\tilde{\pi} \in \Pi_\Delta} \langle \nabla_\pi V_{\lambda,\pi_{\tilde{T}}}^{\pi_{\tilde{T}}}, \tilde{\pi} - \pi_{\tilde{T}} \rangle$$

$$= \max_{\tilde{\pi} \in \Pi_\Delta} \big[ \langle \nabla_\pi V_{\lambda,\pi_{\tilde{T}}}^{\pi_{\tilde{T}}} - \hat{g}_{\lambda,\delta}(\pi_{\pi_{\tilde{T}}}), \tilde{\pi} - \pi_{\tilde{T}} \rangle + \langle \hat{g}_{\lambda,\delta}(\pi_{\pi_{\tilde{T}}}), \tilde{\pi} - \pi_{\tilde{T}} \rangle \big]$$

$$\overset{(a)}{\leq} \frac{1 + \lambda \log|\mathcal{A}|}{T\beta(1-\gamma)} + \frac{\ell_\lambda |\mathcal{S}| \beta}{\Delta}$$

$$+ 2\sqrt{2|\mathcal{S}|} \Big[ \frac{2|\mathcal{S}||\mathcal{A}|\epsilon_V}{\delta} + \frac{4L_\lambda |\mathcal{S}||\mathcal{A}|}{3TN(\Delta-\delta)} \log\Big(\frac{3TN|\mathcal{S}||\mathcal{A}|}{\eta}\Big) + \frac{L_\lambda |\mathcal{S}||\mathcal{A}|}{\Delta-\delta} \sqrt{\frac{2}{N} \log\Big(\frac{3TN|\mathcal{S}||\mathcal{A}|}{\eta}\Big)} + \frac{\delta \ell_\lambda}{\Delta-\delta} \Big], \quad (121)$$

where (a) uses Eqs. (118) and (120).

Use the following hyperparameter choices for Algorithm 1.

$$\Delta = \frac{\pi_{\min}}{3}, \tag{122}$$

$$\beta = \frac{D\Delta\epsilon}{12\ell_\lambda |\mathcal{S}|} = \frac{D\pi_{\min}\epsilon}{36\ell_\lambda |\mathcal{S}|} = \mathcal{O}(\epsilon), \tag{123}$$

$$T = \frac{12(1 + \lambda \log|\mathcal{A}|)}{D\epsilon\beta(1-\gamma)} = \frac{432\ell_\lambda |\mathcal{S}|(1 + \lambda \log|\mathcal{A}|)}{\pi_{\min}D^2(1-\gamma)\epsilon^2} = \mathcal{O}(\epsilon^{-2}) \tag{124}$$

$$\delta = \frac{D\Delta\epsilon}{48\sqrt{2|\mathcal{S}|}\ell_\lambda} = \frac{D\pi_{\min}\epsilon}{144\sqrt{2|\mathcal{S}|}\ell_\lambda} = \mathcal{O}(\epsilon) \overset{(a)}{\leq} \frac{\Delta}{2}, \tag{125}$$

$$\epsilon_V = \frac{D\delta\epsilon}{48|\mathcal{S}||\mathcal{A}|\sqrt{2|\mathcal{S}|}} = \frac{\pi_{\min}D^2\epsilon^2}{13824\ell_\lambda |\mathcal{S}|^2|\mathcal{A}|} = \mathcal{O}(\epsilon^2) \tag{126}$$

$$N = \frac{663552 L_\lambda^2 |\mathcal{S}|^3 |\mathcal{A}|^2}{D^2 \pi_{\min}^2 \epsilon^2} \log \max \left( \frac{165888 L_\lambda^2 |\mathcal{S}|^3 |\mathcal{A}|^2}{D^2 \pi_{\min}^2 \epsilon^2}, \frac{1296 \ell_\lambda |\mathcal{S}|^2 |\mathcal{A}| (1 + \lambda \log |\mathcal{A}|)}{D^2 \eta \pi_{\min} (1 - \gamma) \epsilon^2} \right)$$

$$+ 2 \log \left( \frac{3 |\mathcal{S}||\mathcal{A}|}{\eta} \right) + 3$$

$$= \mathcal{O}[\epsilon^{-2} \log(\eta^{-1} \epsilon^{-1})] \tag{127}$$

where (a) uses $\epsilon \leq 24 \sqrt{2|\mathcal{S}|} \ell_\lambda / D$. With the hyperparameter choices above, we obtain the following inequalities (128)-(130).

$$2\sqrt{2|\mathcal{S}|} \cdot \frac{L_\lambda |\mathcal{S}||\mathcal{A}|}{\Delta - \delta} \sqrt{\frac{2}{N} \log \left( \frac{3TN|\mathcal{S}||\mathcal{A}|}{\eta} \right)}$$

$$\overset{(a)}{\leq} \frac{24 L_\lambda |\mathcal{S}|^{1.5} |\mathcal{A}|}{\pi_{\min}} \sqrt{\frac{\log N}{N} + \frac{1}{N} \log \left( \frac{1296 \ell_\lambda |\mathcal{S}|^2 |\mathcal{A}| (1 + \lambda \log |\mathcal{A}|)}{\eta \pi_{\min} D^2 (1 - \gamma) \epsilon^2} \right)}$$

$$\overset{(b)}{\leq} \frac{24 L_\lambda |\mathcal{S}|^{1.5} |\mathcal{A}|}{\pi_{\min}} \sqrt{\tilde{\epsilon} + \frac{\tilde{\epsilon}}{4}}$$

$$= \frac{12\sqrt{5} L_\lambda |\mathcal{S}|^{1.5} |\mathcal{A}|}{\pi_{\min}} \cdot \frac{D \pi_{\min} \epsilon}{\sqrt{165888} L_\lambda |\mathcal{S}|^{1.5} |\mathcal{A}|}$$

$$\leq \frac{D\epsilon}{12}, \tag{128}$$

where (a) uses Eq. (124) and $\delta \leq \Delta/2 = \pi_{\min}/6$ implied by Eqs. (122) and (125), (b) uses Eq. (127) and its implication that $N \geq 4\tilde{\epsilon}^{-1} \log(\tilde{\epsilon}^{-1})$ with $\tilde{\epsilon} = \frac{\pi_{\min}^2 \epsilon^2}{165888 D^2 L_\lambda^2 |\mathcal{S}|^3 |\mathcal{A}|^2} \leq 0.5$ (since $\epsilon \leq \frac{288 D L_\lambda |\mathcal{S}|^{1.5} |\mathcal{A}|}{\pi_{\min}}$), which implies $\frac{\log N}{N} \leq \tilde{\epsilon}$ based on Lemma 11.

$$\frac{1}{TN} \log \left( \frac{3TN|\mathcal{S}||\mathcal{A}|}{\eta} \right) = \frac{\log(TN)}{TN} + \frac{1}{TN} \log \left( \frac{3|\mathcal{S}||\mathcal{A}|}{\eta} \right) \overset{(a)}{\leq} \frac{1}{2} + \frac{1}{2} = 1, \tag{129}$$

where (a) uses $NT \geq N \geq \max \left[ 3, 2 \log \left( \frac{3|\mathcal{S}||\mathcal{A}|}{\eta} \right) \right]$ and Lemma 11.

$$2\sqrt{2|\mathcal{S}|} \cdot \frac{4 L_\lambda |\mathcal{S}||\mathcal{A}|}{3TN(\Delta - \delta)} \log \left( \frac{3TN|\mathcal{S}||\mathcal{A}|}{\eta} \right) \overset{(a)}{\leq} 2\sqrt{2|\mathcal{S}|} \cdot \frac{\sqrt{2} L_\lambda |\mathcal{S}||\mathcal{A}|}{\Delta - \delta} \sqrt{\frac{1}{TN} \log \left( \frac{3TN|\mathcal{S}||\mathcal{A}|}{\eta} \right)} \overset{(b)}{\leq} \frac{D\epsilon}{12} \tag{130}$$

where (a) uses $\frac{4}{3} < \sqrt{2}$ and $y \leq \sqrt{y}$ for $y = \frac{1}{TN} \log \left( \frac{3TN|\mathcal{S}||\mathcal{A}|}{\eta} \right) \leq 1$ (Eq. (129)), and (b) uses $T \geq 1$ and Eq. (128). By substituting the hyperparameter choices (122)-(127) as well as Eqs. (128) and (130) into Eq. (121), we have

$$\max_{\tilde{\pi} \in \Pi_\Delta} \left\langle \nabla_\pi V_{\lambda, \pi_{\widetilde{T}}}^{\pi_{\widetilde{T}}}, \tilde{\pi} - \pi_{\widetilde{T}} \right\rangle$$

$$\leq \frac{1 + \lambda \log |\mathcal{A}|}{T\beta(1 - \gamma)} + \frac{\ell_\lambda |\mathcal{S}| \beta}{\Delta}$$

$$+ 2\sqrt{2|\mathcal{S}|} \left[ \frac{2|\mathcal{S}||\mathcal{A}| \epsilon_V}{\delta} + \frac{4 L_\lambda |\mathcal{S}||\mathcal{A}|}{3TN(\Delta - \delta)} \log \left( \frac{3TN|\mathcal{S}||\mathcal{A}|}{\eta} \right) + \frac{L_\lambda |\mathcal{S}||\mathcal{A}|}{\Delta - \delta} \sqrt{\frac{2}{N} \log \left( \frac{3TN|\mathcal{S}||\mathcal{A}|}{\eta} \right)} + \frac{\delta \ell_\lambda}{\Delta - \delta} \right]$$

$$\leq \frac{1 + \lambda \log |\mathcal{A}|}{\beta(1 - \gamma)} \frac{\epsilon \beta (1 - \gamma)}{12 D (1 + \lambda \log |\mathcal{A}|)} + \frac{\ell_\lambda |\mathcal{S}|}{\Delta} \cdot \frac{\Delta \epsilon}{12 D \ell_\lambda |\mathcal{S}|}$$

$$+ \frac{4\sqrt{2|\mathcal{S}|} |\mathcal{S}||\mathcal{A}|}{\delta} \cdot \frac{\delta \epsilon}{48 D |\mathcal{S}||\mathcal{A}| \sqrt{2|\mathcal{S}|}} + \frac{\epsilon}{12 D} + \frac{\epsilon}{12 D} + \frac{2\sqrt{2|\mathcal{S}|} \ell_\lambda}{\Delta/2} \cdot \frac{\Delta \epsilon}{48 \sqrt{2|\mathcal{S}|} D \ell_\lambda}$$

$$= \frac{D\epsilon}{2} \overset{(a)}{\leq} \frac{D\lambda}{5 |\mathcal{A}| (1 - \gamma)},$$

where (a) uses $\epsilon \leq \frac{2\lambda D^2}{5 |\mathcal{A}| (1 - \gamma)}$. Then based on Proposition 2, the inequality above implies that

$$\max_{\tilde{\pi} \in \Pi} \left\langle \nabla_\pi V_{\lambda, \pi_{\widetilde{T}}}^{\pi_{\widetilde{T}}}, \tilde{\pi} - \pi_{\widetilde{T}} \right\rangle \leq D\epsilon,$$

which means $\pi_{\widetilde{T}}$ is a $D\epsilon$-stationary policy. Then if $\mu \geq 0$, Corollary 1 implies that $\pi_{\widetilde{T}}$ is also an $\epsilon$-PO policy.

# K. Adjusting Our Results to the Existing Quadratic Regularizer

In Section 4, we have proposed a 0-PPG algorithm and obtain its finite-time convergence result to the desired PO policy for our entropy-regularized value function (8). We will briefly show that 0-PPG algorithm can also converge to PO for the existing performative reinforcement learning defined by the value function (1) with quadratic regularizer $\mathcal{H}_{\pi'}(\pi) = \frac{1}{2}\|d_{\pi,p_{\pi'}}\|^2$ (Mandal et al., 2023; Rank et al., 2024). The *performative value function* can be rewritten as the following $\lambda$-strongly concave function of $d_{\pi,p_\pi}$.

$$V^\pi_{\lambda,\pi} = \langle d_{\pi,p_\pi}, r_\pi \rangle - \lambda \|d_{\pi,p_\pi}\|^2. \tag{131}$$

We can prove the *performative value function* above also satisfies Theorem 1 (gradient dominance) with a different $\mu$, following the same proof logic, since both regularizers $\mathcal{H}_\pi(\pi)$ are strongly convex functions of $d_{\pi,p_\pi}$ which implies that $V^{\pi_\alpha}_{\lambda,\pi_\alpha}$ is a $\mu$-strongly concave function of $\alpha$ as shown in the proof of Theorem 1 in Appendix D. By direct calculation, we can also show that $V^\pi_{\lambda,\pi}$ above is a Lipschitz continuous and Lipschitz smooth function of $\pi \in \Pi$. With these two properties, we can follow the proof logic of Theorem 4 to show that the 0-PPG algorithm (with the same procedure as that of Algorithm 1 except the different values of $V^{\pi_\alpha}_{\lambda,\pi_\alpha}$ in the policy evaluation step) converges to a stationary policy of the *performative value function* (131), which by gradient dominance is a PO policy when the new value of $\mu$ satisfies $\mu \geq 0$.

