# OpenReview forum: "Achieve Performatively Optimal Policy for Performative Reinforcement Learning"
_ICML.cc/2025/Conference — Submitted to ICML 2025_

### Official Review · Reviewer_uoEY · 2025-03-02

**Overall Recommendation:** 2

**Summary:**

- **Proposed Algorithm:** This work introduces a zeroth-order performative policy gradient (0-PPG) algorithm that converges to the PO policy with polynomial computational complexity under mild conditions.
- **Key Theoretical Properties:**
  - When the policy regularizer dominates the environmental shift, the value function exhibits a gradient dominance property, meaning any stationary point is a PO.
  - Although the value function may have unbounded gradients, all sufficiently stationary points lie within a convex and compact policy subspace $\Pi_\Delta$, where the policy value is bounded below by $\Delta > 0$, ensuring the gradient is both bounded and Lipschitz continuous.

**Claims And Evidence:**

Evidence well supports the claims

**Essential References Not Discussed:**

No.

**Experimental Designs Or Analyses:**

There are no experiments.

**Methods And Evaluation Criteria:**

Proposed methods make sense, and there are no experiments.

**Other Comments Or Suggestions:**

Please refer to the [Questions For Authors]

**Other Strengths And Weaknesses:**

Strengths:
 - I like how authors provide a key takeaway of Theorems (Like Remark after Theorem 1 or Implications after Theorem 2, and remark after Proposition 1). This makes the reading more comfortable.
Weakness:
 - Please refer to the [Questions For Authors]

**Questions For Authors:**

My current score is approximately 2.5 (I cannot choose between 2 and 3, but I am currently just set at 2), primarily due to following questions, especially 2 and 4. I am willing to increase my score if the following issues are addressed:

1. **Clarification on Existing Work:** Could the authors specify why previous research in performative RL has focused solely on the PS policy?

2. **Interpretation of Regularized Value Function:** In both the abstract and the remark following Theorem 1, the analysis suggests that when the policy regularizer dominates the environmental shift, the value function exhibits a gradient dominance property, which is intuitively appealing. However, I am concerned about the practical significance of the optimal policy derived from this regularized value function. Since the policy regularizer may impede convergence to the true optimal policy (thereby affecting generalization), if it dominates the environmental shift, does this imply that the optimal policy is biased towards a more uniform distribution? If so, this might render the primary contribution somewhat trivial.

3. **Insights from Theorem 3:** Could the authors elaborate on the key takeaways of Theorem 3? The Lipschitz continuity property appears to be a direct consequence of Assumptions 1 through 3. It would be helpful to understand how the upper bound is affected by the parameters $L$ and $l$.

4. **Experimental Validation:** Has the proposed approach been tested empirically? Given that the paper introduces several convergence theorems, including experimental results—perhaps on a simple environment like a grid world—would strengthen the manuscript by demonstrating the convergence behavior of the PO policy.

**Relation To Broader Scientific Literature:**

This paper looks promising since it has done a theoretical analysis on PO convergence. However, the second concern that I have written down on  [Questions For Authors]  may challenge the novelty of this paper.

**Theoretical Claims:**

I have not taken a close look at all proof,s but the takeaways and intuitions, remarks after the Theorem all make sense (at least to me)

---

> ### Author Rebuttal · Authors · 2025-04-01
>
> **Clarification on Existing Work:** Could you specify why previous research in performative RL has focused solely on the PS policy?
>
> **A:** Great question. There are two reasons. First, the method to obtain a performatively stable (PS) policy is more straightforward to think of than to obtain a performatively optimal (PO) policy. To elaborate, since a PS policy $\pi_{PS}$ is only required to be optimal in its corresponding fixed environment $(p_{\pi_{PS}}, r_{\pi_{PS}})$, so we can obtain a PS policy by repeated training, i.e., applying **traditional policy optimization methods** to a fixed environment. In contrast, a PO policy $\pi_{PO}$ is required to have larger value in the environment $(p_{\pi_{PO}}, r_{\pi_{PO}})$ than the value of any policy $\pi$ in its own environment $(p_{\pi}, r_{\pi})$, so we cannot use **traditional policy optimization methods**. Second, the distance between a PS policy and a PO policy is $\mathcal{O}(\epsilon_p+\epsilon_r)$ where $\epsilon_p$ and $\epsilon_r$ are the Lipschitz constants of the Lipschitz continuous $p_{\pi}$ (transition kernel) and $r_{\pi}$ (reward), so PS approximates PO well in a slowly changing environment with small $\epsilon_p$ and $\epsilon_r$ (Mandal et. al. 2023).
>
> **Interpretation of Regularized Value Function:** I am concerned about the practical significance of the optimal policy from this regularized value function. Since the policy regularizer may impede convergence to the true optimal policy (thereby affecting generalization), if it dominates the environmental shift, does this imply that the optimal policy is biased towards a more uniform distribution? If so, this might render the primary contribution somewhat trivial.
>
> **A:** Great question. The answer is partially yes. The optimal policy for regularized objective is closer to the uniform policy than the optimal policy for unregularized setting. However, we do not think it as a bias, because the optimal policy for entropy regularized setting is also an important target. To elaborate, entropy regularization has been demonstrated to make the policy robust against perturbation to the environment (transition kernel and reward), thereby improving generalization [1], and to encourage the agent to explore unknown environment, yielding better exploration-exploitation trade-off (Mnih et al., 2016; Mankowitz et al., 2019; Cen et al., 2022; Chen and Huang, 2024). As our algorithm converges to this important target policy, the regularized optimal solution, we do not think it as a bias.
>
> [1] Eysenbach, Benjamin, and Sergey Levine. "Maximum Entropy RL (Provably) Solves Some Robust RL Problems." International Conference on Learning Representations (2022).
>
> **Insights from Theorem 3:** Could you elaborate on the key takeaways of Theorem 3? The Lipschitz continuity property appears to be a direct consequence of Assumptions 1 through 3. It would be helpful to understand how the upper bound is affected by the parameters $L$ and $\ell$.
>
> **A:** The key takeaway of Theorem 3 is that the objective function $V _ {\lambda,\pi}^{\pi}$ is Lipschitz continuous and Lipschitz smooth in the domain $\pi\in\Pi_{\Delta}=\{\pi\in\Pi:\pi(a|s)\ge\Delta\}$. You may misunderstood Theorem 3. First, the Lipschitz property comes from Assumption 1-2 but the proof is not very straightforward. Second, the upper bounds for Lipschitz continuity and Lipschitz smoothness are proportional to $L_{\lambda}$ and $\ell_{\lambda}$ (not $L$ and $\ell$) respectively, defined by Eqs. (22) and (23) respectively. $L_{\lambda}$ and $\ell_{\lambda}$ depend on problem-related constants like $|\mathcal{S}|$, $|\mathcal{A}|$, $\gamma$, $\lambda$, $\epsilon_p$, $\epsilon_r$,  not tunable parameters.
>
> **Experimental Validation:** Has the proposed approach been tested empirically?
>
> **A:** Good question. We compared our Algorithm 1 with the existing repeated training algorithm in a simulation environment with 5 states, 4 actions, discount factor $\gamma=0.95$, entropy regularizer coefficient $\lambda=0.5$, transition kernel $p_{\pi}(s'|s,a)=\frac{\pi(a|s)+\pi(a|s')+1}{\sum_{s''}[\pi(a|s)+\pi(a|s'')+1]}$, and reward $r_{\pi}(s,a)=\pi(a|s)$. We implement our Algorithm 1 for 400 iterations with $N=1000$, $\beta=0.01$, $\Delta=10^{-3}$, $\delta=10^{-4}$ and value functions evaluated by value iteration. The repeated training algorithm obtains the next policy $\pi_{t+1}$ by applying the natural policy gradient algorithm [1] with 100 steps and stepsize 0.01 to the entropy-regularized reinforcement learning with transition kernel $p_{\pi_t}$ and reward $r_{\pi_t}$. Both algorithms start from the uniform policy (i.e. $\pi(a|s)\equiv 1/4$). Our experimental results in the anonymous website https://docs.google.com/document/d/1bH3eEoGhfDwq1NBNW7_zjCSLvvmcUyDusaINivK5bdo/edit?tab=t.0 shows that the existing repeated training algorithm stucks at the initial policy which is performatively optimal, while our Algorithm 1 converges to a much larger objective function value.

---

### Official Review · Reviewer_MUNV · 2025-03-13

**Overall Recommendation:** 4

**Summary:**

The paper studies the problem of performative reinforcement learning, where the choice of policy actively influences the dynamics in the environments (transitions) as well as the rewards.

The authors introduce the first algorithm which provably converges to the performatively optimal (not stable policy) under standard regularity conditions.

**Claims And Evidence:**

Yes, all the claims are well supported.

**Essential References Not Discussed:**

The relevant literature is appropriately cited. It might be nice to tell a bit of this story above around how their results contribute to the broader literature on performative prediction, but this is really up to the authors.

**Experimental Designs Or Analyses:**

NA

**Methods And Evaluation Criteria:**

The main contributions of the paper are theoretical. Their analysis makes sense.

**Other Comments Or Suggestions:**

NA

**Other Strengths And Weaknesses:**

Convergence to optimality, not stability, is a real strength of the paper. The analysis is substantial and involved but the authors do a good job of providing intuition. I think the paper would be even better if they give a broader overview of performativity and spend a bit more time delving into the intuition for their proofs. For instance, readers may not be familiar with these kinds of gradient dominance conditions and a gentler review of why these conditions are useful and where they have been previously studied in the literature (e.g. LQR) could be very nice.

**Questions For Authors:**

NA

**Relation To Broader Scientific Literature:**

The paper makes an excellent contribution to the growing area of performative prediction and performative reinforcement learning. To date, there was no known algorithm that one could show convergence to the performatively optimal solution.

Their results mirror a similar story developed in the classical performative prediction literature over the last few years where initially people only knew of algorithms that would converge to stable points. Then, in 2021, Miller et al introduced the first set of conditions under which the performative risk was convex, and designed algorithms which converged to the performatively optimal solution.

This result completes a similar arc for the performative reinforcement learning setting which is substantially more complicated than that initially considered by Perdomo et al in their paper on performative prediction. This is a very nice result that will be of interest to the community. Here, gradient dominance is somehow the analogous structural condition to convexity in the standard setup.

**Theoretical Claims:**

I did not.

---

> ### Author Rebuttal · Authors · 2025-04-01
>
> **Essential References Not Discussed:** The relevant literature is appropriately cited. It might be nice to tell a bit of this story above around how their results contribute to the broader literature on performative prediction, but this is really up to the authors.
>
> **A:** Thank you very much for telling the story showing our contribution to the broader literature on performative prediction. We are glad to add this story to our revision.
>
> **Other Strengths And Weaknesses:** I think the paper would be even better if they give a broader overview of performativity and spend a bit more time delving into the intuition for their proofs. For instance, readers may not be familiar with these kinds of gradient dominance conditions and a gentler review of why these conditions are useful and where they have been previously studied in the literature (e.g. LQR) could be very nice.
>
> **A:** Thanks for your suggestion. We will add a discussion of related works including those on performative prediction. We will stress that the major idea of both performative prediction and performative reinforcement learning is performativity which means the data distribution can be affected by the decision, as observed in many applications.
>
> We will elaborate more on gradient dominance right after our Theorem 1 as you suggested. Specifically, when $\mu\ge 0$, our Theorem 1 implies the following gradient dominance result widely used in reinforcement learning [1,2].
> $$f(\pi^*)-f(\pi)\le C_1\max _ {\pi'\in\Pi}\big\langle \nabla f(\pi),\pi'-\pi\big\rangle,\quad{\rm(G1)}$$
> where we use the constant $C_1=D^{-1}$, the objective function $f(\pi)=V _ {\lambda,\pi}^{\pi}$, the performatively optimal solution $\pi^*\in{\arg\max} _ {\pi}f(\pi)$, $C_1=D^{-1}>0$. This further implies the following weaker gradient dominance result widely used in optimization [3,4] and linear quadratic regulator (LQR) [5,6].
> $$f(\pi^*)-f(\pi)\le C_2||\nabla f(\pi)||^{\alpha},$$
> where we use the constant $C_2=2D^{-1}>0$ (since $||\pi'-\pi||\le 2$ in Eq. (G1) above), and the power $\alpha=1$.
>
> Both the gradient dominance conditions above are useful for global convergence to the optimal solution $\pi^*$, since under either of these conditions, $||\nabla f(\pi_t)||\to 0$ can imply $f(\pi_t)\to f(\pi^*)$.
>
> [1] Agarwal, A., Kakade, S. M., Lee, J. D., \& Mahajan, G. (2021). On the theory of policy gradient methods: Optimality, approximation, and distribution shift. Journal of Machine Learning Research, 22(98), 1-76.
>
> [2] Chen, Z., Wen, Y., Hu, Z., \& Huang, H. (2024). Robust Reinforcement Learning with General Utility. Advances in Neural Information Processing Systems, 37, 11290-11344.
>
> [3] Masiha, S., Salehkaleybar, S., He, N., Kiyavash, N., \& Thiran, P. (2022). Stochastic second-order methods improve best-known sample complexity of SGD for gradient-dominated functions. Advances in Neural Information Processing Systems, 35, 10862-10875.
>
> [4] Nesterov, Y., \& Polyak, B. T. (2006). Cubic regularization of Newton method and its global performance. Mathematical programming, 108(1), 177-205.
>
> [5] Mohammadi, H., Zare, A., Soltanolkotabi, M., \& Jovanović, M. R. (2021). Convergence and sample complexity of gradient methods for the model-free linear–quadratic regulator problem. IEEE Transactions on Automatic Control, 67(5), 2435-2450.
>
> [6] Ye, L., Mitra, A., \& Gupta, V. (2024, December). On the Convergence of Policy Gradient for Designing a Linear Quadratic Regulator by Leveraging a Proxy System. In 2024 IEEE 63rd Conference on Decision and Control (CDC) (pp. 6016-6021). IEEE.

---

### Official Review · Reviewer_qtXf · 2025-03-14

**Overall Recommendation:** 1

**Summary:**

This paper proposes an algorithm to compute performatively optimal policies, i.e. policies maximizing the expected sum of rewards in an MDP-like environment where the transition and reward functions are dependent on the policy that is executed. The algorithm consists in iteratively building an ascent direction from samples in the decision process and using this direction in the Frank-Wolfe algorithm to update the policy. Convergence is guaranteed, as the ascent direction is "valid" and the objective function is gradient dominated.

**Claims And Evidence:**

The claim that it is possible to find the performatively optimal policy with the proposed algorithm is theoretically supported.

There are several other claims that are incorrect or inefficiently detailed:
1. Authors claim that there is no analytical form to the performative policy gradient [line 320 right column]. To my understanding this has not been shown, and, intuitively, It is unclear for me why there would not be an analytical form to the gradient.
2. Authors claim in the abstract (and through the paper) that it is a "zeroth-order policy gradient method". This is insufficient to well-understand how the policy is effectively optimized and misleading to my understanding of what a zero-order method, first-order method, and policy gradient method is. On the one hand, a zero-order method optimizes a function without computing gradients but solely estimating the function. A first-order method, on the other hand, uses gradient. Policy-gradient methods fall into the second type of methods as the point is to estimate the gradient of the return (and computing the gradient of the policy) to do stochastic gradient ascent steps. If one where to use finite difference to compute an ascent direction to optimize the return, I am not sure it can still be considered a policy gradient method. The abstract should be clearer about the how the ascent direction is computed and used to update the policy.
3. Authors highlight that the performative optimal policy cannot be computed with previous algorithms from the literature. It nevertheless seems that the problem at hand is a particular case of some stochastic game where the objective to compute policies against adversarial opponents, e.g. [1, 2, 3]. Does this part of the literature provides algorithms that would compute an optimal performative policy?


[1] Sessa, P. G., Bogunovic, I., Kamgarpour, M., & Krause, A. (2020). Learning to play sequential games versus unknown opponents. Advances in neural information processing systems, 33, 8971-8981.

[2] Ramponi, G., Metelli, A. M., Concetti, A., & Restelli, M. (2021). Learning in non-cooperative configurable markov decision processes. Advances in Neural Information Processing Systems, 34, 22808-22821.

[3] Jackson, M. T., Jiang, M., Parker-Holder, J., Vuorio, R., Lu, C., Farquhar, G., ... & Foerster, J. (2023). Discovering general reinforcement learning algorithms with adversarial environment design. Advances in Neural Information Processing Systems, 36, 79980-79998.

**Essential References Not Discussed:**

See previous remarks on non-stationary or adversarial RL.

**Experimental Designs Or Analyses:**

There is no empirical evaluation, which is to me problematic. The paper should include experiments to validate the final algorithm, and compare to algorithms from the literature dealing with non-stationary or adversarial settings.

**Methods And Evaluation Criteria:**

There is no evaluation of the final algorithm.

**Other Comments Or Suggestions:**

I would have clearly stated that distributions are represented by vectors at the beginning of section 2.1. In other words, sentence line 90 should come earlier for clarity.

**Other Strengths And Weaknesses:**

Authors should formally define the n-step transition distribution in equation (2).

In section 2.1, when defining $\mathcal{P}$, the sum should be over $s'$ and not $s$ I beleive.

I think equation (5) might be wrong, is it $r_{d'}$ or $r_d$? In (Mandal et al., 2023), they use the measure $d$ in their equation (3).

Authors should be mathematically clear about what a "valid approximation" is in Proposition 1.

**Questions For Authors:**

Does convergence require the batch size $N$ to grow unbounded?

**Relation To Broader Scientific Literature:**

The contribution should be related to the literature dealing with non-stationary or adversarial settings. Do there exist algorithms that could be applied to compute performatively optimal policy?

**Theoretical Claims:**

Theoretical claims seem correct, but I haven't checked proofs in appendices.

---

> ### Author Rebuttal · Authors · 2025-04-01
>
> **Claims And Evidence (1):** Why there would not be an analytical form to the gradient?
>
> **A:** Good question. I later found this gradient can be computed by chain rule, but involves the unknown $\nabla_{\pi}p_{\pi}(s'|s,a)$ and $\nabla_{\pi}r_{\pi}(s,a)$. We will revise this claim.
>
> **Claims And Evidence (2):** The abstract should be clearer about the how the ascent direction is computed and used to update the policy. Can we use the name "zeroth-order policy gradient method"?
>
> **A:** Thanks for your suggestions. Our algorithm uses a Frank-Wolfe update to find the ascent direction, where the policy gradient is approximated by its zeroth order estimation (will be added to the revised abstract), so the name "zeroth-order policy gradient method" is valid, as also been used in [1,2]. We may also use "zeroth-order Frank-Wolfe algorithm" to reveal more optimization details.
>
> [1] Wang, Z., et al. Policy evaluation in distributional LQR. In Learning for Dynamics and Control Conference 2023.
>
> [2] Han, Y., Razaviyayn, M., \& Xu, R. Policy gradient finds global optimum of nearly linear-quadratic control systems. NeurIPS 2022 Workshop.
>
> **Claims And Evidence (3):** It seems that the problem at hand is a particular case of some stochastic game where the objective to compute policies against adversarial opponents, e.g. [1-3]. Do their algorithms compute an optimal performative policy?
>
> **A:** No, these adversarial settings are very different from our performative reinforcement learning problem without adversarial environment.
>
> **Experimental Designs Or Analyses:** The paper should include experiments.
>
> **A:** Thanks for your suggestion. Due to limited space, see the experimental details in my final response to Reviewr uoEY. Our result in https://docs.google.com/document/d/1bH3eEoGhfDwq1NBNW7_zjCSLvvmcUyDusaINivK5bdo/edit?tab=t.0 shows that our Algorithm 1 outperforms the existing repeated training algorithm.
>
> **Relation To Broader Scientific Literature:** The contribution should be related to the literature dealing with non-stationary or adversarial settings. Do there exist algorithms that could be applied to compute performatively optimal policy?
>
> **A:** No, since to our knowledge, performative reinforcement learning is not a special case of any other problems.
>
> We will add a discussion of related works including those on non-stationary MDP (e.g. [1,2]) that are weakly related to our work. To elaborate, during the training of performative reinforcement learning, the policy $\pi$ and thus the environment $(p_{\pi}, r_{\pi})$ change with iterations. In non-stationary MDP, the environment $(p_t, r_t)$ changes with MDP time scale $t$ not the iteration.
>
> [1] Chandak, Yash, et al. Optimizing for the future in non-stationary MDPs. ICML 2020.
>
> [2] Chandak, Yash, et al. Towards safe policy improvement for non-stationary MDPs. Neurips 2020.
>
> **Other Strengths And Weaknesses (1):** Authors should formally define the n-step transition distribution in Eq. (2).
>
> **A:** Thanks for your suggestion. Since $s_{t+1}\sim p_{\pi}(\cdot|s_t,a_t)$, $a_t\sim\pi(\cdot|s_t)$ and $s_0\sim\rho$, the n-step transition can be computed below, which will be added to the revision.
> $$\mathbb{P} _ {\pi,p,\rho}(s_n=s,a_n=a)=\sum_{s_0,...,s_{n-1}\in\mathcal{S}}\sum_{a_0,...,a_{n-1}\in\mathcal{A}}\rho(s_0)\pi(a|s)p_{\pi}(s|s_{n-1},a_{n-1})\pi(a_{n-1}|s_{n-1})\prod_{t=0}^{n-2}[\pi(a_t|s_t)p_{\pi}(s_{t+1}|s_t,a_t)].$$
>
> **Other Strengths And Weaknesses (2):** In Section 2.1, when defining $\mathcal{P}$, the sum should be over $s'$ and not $s$ I believe.
>
> **A:** Thanks. We have corrected that.
>
> **Other Strengths And Weaknesses (3):** I think equation (5) might be wrong, is it $r_{d'}$ or $r_d$? In (Mandal et al., 2023)? they use the measure $d$ in their equation (3).
>
> **A:** We use $r_{d'}$ for two reasons. First, since performatively stable policy is defined as $\pi_S\in{\arg\max} _ {\pi'}V _ {\pi_S}^{\pi'}(\rho)$, their Eq. (3) that defines the corresponding performatively stable occupancy measure $d_S$ should have used $r_{d_S}$, corresponding to our Eq. (5) with $d'=d_S$. Second, their Eq. (5) about their repeated training algorithm corresponds to our Eq. (5) with $d'=d_t$ at iteration $t$.
>
> **Other Strengths And Weaknesses (4):** Authors should be mathematically clear about what a "valid approximation" is in Proposition 1.
>
> **A:** Thanks for your suggestion. "Valid approximation" means the $\pi+\delta u_i$ and $\pi-\delta u_i$ in Eq. (26) are valid policies, i.e., $\pi'(a|s)\ge0$ and $\sum_a\pi'(a|s)=1$ for $\pi'\in\{\pi\pm\delta u_i\}$. We will add that explanation to the revision.
>
> **Other Comments Or Suggestions:** Sentence line 90 should be moved to the beginning of section 2.1 for clarity.
>
> **A:** Thanks. We have done that.
>
> **Questions For Authors:** Does convergence require the batch size to grow unbounded?
>
> **A:** No. Usually we fix $\epsilon,\eta$, so the batch size $N=O[\epsilon^{-2}\log(\eta^{-1}\epsilon^{-1})]$ is also fixed.

---

> > ### Comment · Reviewer_qtXf · 2025-04-02
> >
> > Thank you for your response. I would advise updating the paper so that these elements are made clear.
> >
> > I think that the paper is still incomplete without an experimental validation, which I cannot review solely based on the elements you provided in the response to reviewers.

---

> > > ### Author Response · Authors · 2025-04-02
> > >
> > > Did you see our experimental results in
> > > https://docs.google.com/document/d/1bH3eEoGhfDwq1NBNW7_zjCSLvvmcUyDusaINivK5bdo/edit?tab=t.0
> > >
> > > The experimental details are in our final response to the Reviewer uoEY. We **have added the above experimental details and results to our paper, but ICML 2025 does not allow us to upload the updated paper.**
> > >
> > > In addition, how do you think about our responses to your other concerns?
> > >
> > > Thanks.
> > > Authors

---

### Decision · Program_Chairs · 2025-05-01

**Decision:**

Reject

**Comment:**

The submission received three reviews with all three reviewers acknowledging the rebuttal. There was some divergence in the opinions of the reviewers, however, after the discussion, the panel feels that the paper is not ready yet for publication. While the main strength is a credible proof of convergence in polynomial time to a perfomatively optimal policy for entropy regularised objective value functions, the two main limitations that prevent a more positive recommendation at the moment are 1) convergence holds only for large values of the regularizer and 2) there is a lack of proper experimental evaluation. In more detail:

- The convergence holds when the regulariser is large enough, which essentially cancels out the performative effect rendering the result somewhat standard (convergence towards an approximately uniform distribution when regularisation is high). It is also indicative that the main proof leverages/adapts standard convergence theory to show this result. The paper does not accurately present that limitation resulting in the impression that it (slightly) overclaims its contributions. A proper account of this would benefit the paper.
- Lack of proper experimental evaluation. While this is a theoretical paper, and could be appreciated solely as such, the introduction of the entropy regulariser in the objective function and the ensuing uncertainty (see previous comment) would benefit from experimental study to assess how much this can depart from the original (unregularised) objective function and affect actual optimisation in different settings. The experiments offered in the rebuttal are welcome but are still insufficient to provide an adequate experimental evaluation.

The relation to the literature was lifted as a concern and some other comments were satisfactorily discussed in the rebuttal leading to improvements in a potentially revised submission. In sum, the main result remains important, but the limitations mentioned above prevent a stronger positive recommendation at the moment.